# MacroH2A restricts inflammatory gene expression in melanoma cancer-associated fibroblasts by coordinating chromatin looping

Dan Filipescu [1,2] ✉, Saul Carcamo[3,4], Aman Agarwal[3,4], Navpreet Tung[1,5], Étienne Humblin [1,5], Matthew S. Goldberg [2], Nikki S. Vyas[2,6], Kristin G. Beaumont [7], Deniz Demircioglu [1,3,4], Subhasree Sridhar [1,8], Flavia G. Ghiraldini[1,3], Claudia Capparelli [9], Andrew E. Aplin [9], Hélène Salmon[1,5,10], Robert Sebra [8,7,11], Alice O. Kamphorst [1,3,5,8], Miriam Merad [1,3,5,8], Dan Hasson[8,4,3,2,1,11] & Emily Bernstein [8,3,2,1,11] ✉

MacroH2A has established tumour suppressive functions in melanoma and other cancers, but an unappreciated role in the tumour microenvironment. Using an autochthonous, immunocompetent mouse model of melanoma, we demonstrate that mice devoid of macroH2A variants exhibit increased tumour burden compared with wild-type counterparts. MacroH2A-deficient tumours accumulate immunosuppressive monocytes and are depleted of functional cytotoxic T cells, characteristics consistent with a compromised anti-tumour response. Single cell and spatial transcriptomics identify increased dedifferentiation along the neural crest lineage of the tumour compartment and increased frequency and activation of cancer-associated fibroblasts following macroH2A loss. Mechanistically, macroH2A-deficient cancer-associated fibroblasts display increased myeloid chemoattractant activity as a consequence of hyperinducible expression of inflammatory genes, which is enforced by increased chromatin looping of their promoters to enhancers that gain H3K27ac. In summary, we reveal a tumour suppressive role for macroH2A variants through the regulation of chromatin architecture in the tumour stroma with potential implications for human melanoma.

Histone variant chromatin incorporation has profound consequences within the cell, and its deregulation plays an important role in cancer[1]. Among the H2A variants, macroH2A has a distinct carboxy-terminal non-histone macro domain[2]. Vertebrate genomes contain two macroH2A genes: *H2AFY* and *H2AFY2*. *H2AFY* (also known as *MACROH2A1*) produces alternative splice isoforms, macroH2A1.1 and macroH2A1.2, whereas *H2AFY2* (also known as *MACROH2A2*) produces macroH2A2 (refs. 3,4). Although macroH2A generally associates with condensed chromatin and transcriptionally inactive genes (see ref. 5 for review), its role in distinct cell types, tissues or disease states is complex[1,6–9]. Notably, the function of macroH2A variants during tumorigenesis in vivo remains poorly understood.

Mice deficient for both *H2afy* and *H2afy2*, referred to as double KO (dKO) herein, lack manifest developmental abnormalities or cancer development during ageing[10]. However, macroH2A regulates cellular states. For example, macroH2A variants impede reprogramming of somatic cells towards pluripotency[11–13] and mediate gene expression in response to stimuli, such as pro-inflammatory signals[7,14–16] or oncogene-induced senescence[8].

Melanoma incidence is rising and remains clinically challenging to treat[17]. We have previously shown that macroH2A expression is lost in advanced human melanoma and, functionally, macroH2A depletion from melanoma cells promoted enhanced tumour growth and metastatic colonization[18]. Accordingly, overexpression of macroH2A2 induced tumour cell dormancy and suppressed the growth of disseminated cancer cells into overt metastasis[19]. Furthermore, high macroH2A levels correlate with favourable prognosis in lung cancer and in colon cancer[20,21]. Collectively, these data provide support for a tumour suppressive role for macroH2A.

Epigenetic regulation of the melanoma tumour microenvironment (TME) remains poorly characterized. Here we investigate the consequences of macroH2A deficiency on autochthonous BRAF^V600E/PTEN-deficient melanomas. We report an unappreciated level of intratumoral heterogeneity and an impaired anti-tumour immune response in dKO animals. This phenotype is driven by cancer-associated fibroblasts (CAFs), which accumulate in the TME and express high levels of inflammatory genes through increased promoter–enhancer interactions. Our study highlights a unique role for macroH2A histones in the TME by limiting the pro-inflammatory properties of the tumour stroma.

## Results

### MacroH2A suppresses autochthonous melanoma growth

We crossed mice constitutively deficient for macroH2A histones (*H2afy* and *H2afy2* dKO strain)[10] to the *Braf^CA Pten^fl Tyr-CreERT2* triallelic melanoma strain[22] (Extended Data Fig. 1a) and initiated tumours through the topical application of 4-hydroxytamoxifen in wild-type (WT) mice and in dKO mice. Although the tumour area was similar at 25 days post-induction (DPI), by 50 DPI, dKO tumours acquired a significant >40% increase in area and a twofold increase in weight versus WT tumours. The increase was independent of sex (Fig. 1a–c and Extended Data Fig. 1b) and involved increased vertical growth (Fig. 1d). The dKO tumours displayed accelerated development (Fig. 1e) and progressed beyond 50 mm² significantly earlier than WT tumours (Extended Data Fig. 1c).

Histology analyses revealed melanocytic lesions with a pigmented epicentre, loss of pigmentation on the margins and depth, and invasion through subcutaneous muscle as determined by S100 staining (Fig. 1f). MelanA staining was present in pigmented foci within tumours and normal melanocytes in the hair follicle, which suggested that most transformed cells lost melanocytic antigens (Fig. 1f). Differences were not observed in proliferation markers, melanocytic spread into the epidermis (pagetoid scatter) or pigmentation between genotypes (Extended Data Fig. 1d–f). Besides melanophages or isolated disseminated tumour cells in lymph nodes (Extended Data Fig. 1g), or rare non-pigmented S100-negative lung lesions (Extended Data Fig. 1h), we did not detect overt metastases at 50 DPI. Furthermore, *Pten^WT* dKO mice developed nevi following BRAF^V600E induction[22], but did not proliferate or transform during the lifetime of the animals (Extended Data Fig. 1i). MacroH2A1, present throughout normal skin, was retained in melanomas, whereas macroH2A2, detected primarily at low levels in the hair follicle in normal skin, was focally present in the tumour (Fig. 1g and Extended Data Fig. 1j). Altogether, the results indicate that macroH2A loss promotes tumour growth, which may occur through mechanisms distinct from melanocytic hyperproliferation.

### dKO melanomas deregulate anti-tumour immunity genes

Transcriptomic profiling of bulk tumours at 50 DPI highlighted 170 upregulated genes and 218 downregulated genes (Extended Data Fig. 2a and Supplementary Table 1). Gene set enrichment analysis (GSEA) terms that were up in dKO samples were primarily associated with immune function (Fig. 2a and Extended Data Fig. 2b). The pro-inflammatory cytokines *Ccl2*, *Cxcl1*, *Ccl9* and *Il6* (Fig. 2b), produced by multiple cell types to attract monocytes and myeloid progenitors into tumours[23–26], were upregulated. Monocytes and neutrophils recruited by these cytokines and signalling though the G-CSF receptor (*Csf3r*) may limit the anti-tumour immune response[23–25,27–29]. Also upregulated were matrix metalloproteinases (Fig. 2b), which have numerous functions, including proteolysis-mediated activation of secreted chemokines[30,31].

Among the downregulated genes, those associated with muscle and keratinization (Fig. 2b and Extended Data Fig. 2b) indicated displacement of normal epidermal and subcutaneous muscle structures in the dKO samples. Also downregulated were gasdermins, mediators of pyroptosis, an immunogenic form of cell death[32], together with the CD200 axis, which inhibits myeloid cell function[33,34]. Importantly, dKO tumours downregulated markers of effector CD8⁺ cells, including *Cd8a*, *Ifng* and *Gzmc*. Overall, the transcriptomics data suggest that there is an impaired anti-tumour immune response in dKO animals.

### Tumour-infiltrating immune cell dysfunction in dKO mice

We immunophenotyped tumours at 50 DPI (Extended Data Fig. 2c) and observed expansion of classical (CCR2⁺) and non-classical (CCR2⁻CX3CR1⁺) monocytes in dKO samples (Fig. 2c and Extended Data Fig. 2d,e). These cells, often termed mononuclear phagocyte-like myeloid-derived suppressor cells, inhibit T cell function and have negative prognostic value in cancer[35]. Accordingly, there was a significant reduction in the relative abundance of CD8⁺ T cells (Fig. 2c) and their proliferation (Fig. 2d). Other CD8 cell markers and their counterparts in CD4⁺ cells were not affected (Extended Data Fig. 2f). Increased PD-1 ligands in dKO immune cells (Fig. 2e and Extended Data Fig. 2g), probably stemming from the myeloid compartment to which PD-L2 is mainly restricted[36], further indicated immunosuppression[37].

By sorting CD8⁺ T cells and profiling them using RNA sequencing (RNA-seq) (Supplementary Table 2), we found downregulation of the G2/M checkpoint and E2F targets, as well as genes associated with response to interferons and viral infection (Fig. 2f,g). Upregulated genes highlighted a pathological CD8⁺ T cell phenotype, including the RORγt receptor (*Rorc*), IL-17F and IL-23R, which are markers of T_C17 cell polarization. This response can be driven by monocyte-derived IL-1β (the receptors of which are upregulated in the dKO CD8⁺ population; Fig. 2g), IL-6 and IL-23 (which signal though the upregulated JAK–STAT pathway)[38] (Fig. 2f), and is associated with impaired anti-tumour activity[39–41]. Finally, TNF signalling (upregulated in dKO mice; Fig. 2g) in activated melanoma-infiltrating CD8⁺ T cells triggers their death[42].

We did not observe significant changes in the relative proportions of immune cells in peripheral blood, regardless of tumour-bearing status (Extended Data Fig. 2h,i), which suggested that immunophenotypic changes were TME-specific. The increase in monocytes and decrease in CD8⁺ T cell abundance and functionality was recapitulated at 35 DPI (Extended Data Fig. 2j,k), which signified that the dKO immunophenotype is not a consequence of tumour size. Together, our data indicate that without macroH2A, increased pro-tumour inflammatory signals in the TME inhibit immune-mediated tumour cell killing, which facilitates tumour growth.

### Melanocyte dedifferentiation in dKO tumours

This melanoma model lacks macroH2A in a constitutive manner; thus, the dKO phenotype could stem from several cell types. We performed single cell RNA-seq (scRNA-seq) of three WT and three dKO melanomas, generating a dataset of ~24,000 high-quality cells. We identified

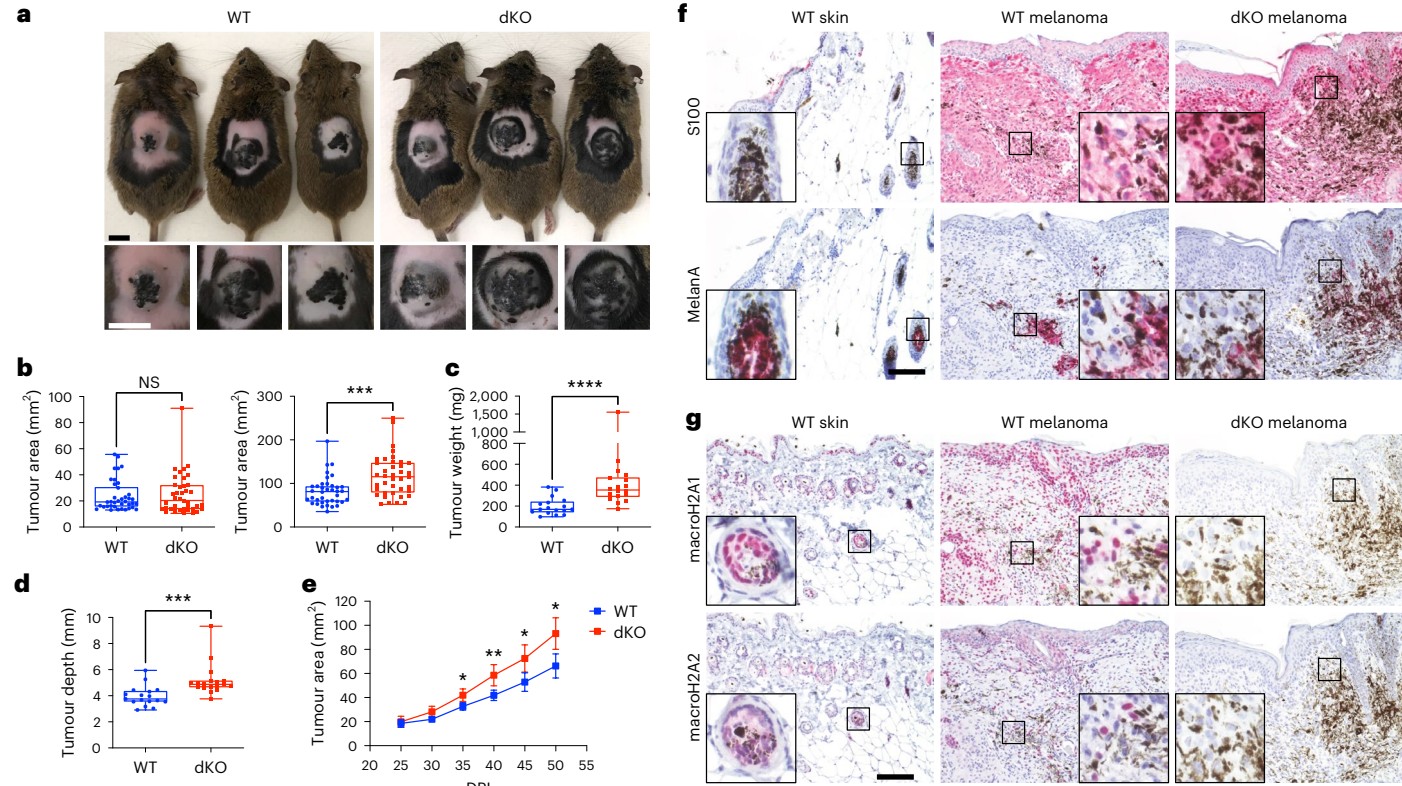

**Fig. 1 | MacroH2A loss in the melanoma TME promotes primary tumour growth. a**, Macroscopic appearance of BRAF^V600E^/PTEN-deficient autochthonous melanomas in macroH2A WT mice and in dKO mice at 50 DPI. **b**, Comparison of tumour area across genotypes at the indicated time points. $n_{WT}$ = 38, $n_{dKO}$ = 39. $P$ = 0.7962 at 25 DPI, $P$ = 0.0002 at 50 DPI. **c**, Measured weight of resected tumours at the end point (50 DPI). $n_{WT}$ = 17, $n_{dKO}$ = 18. $P$ < 0.0001. **d**, Average tumour depth calculated from the tumour area and volume at the end point (50 DPI). $n_{WT}$ = 17, $n_{dKO}$ = 18. For **b**–**d**, significance was determined using Mann–Whitney two-tailed test. Box plot whiskers represent the minimum to maximum range, the box plot limits the 25th to 75th percentiles, and the centre line the median. $P$ = 0.0001.

**e**, Tumour growth kinetics between 25 and 50 DPI. $n_{WT}$ = 22, $n_{dKO}$ = 22. Mean and 95% confidence interval error bars are shown. $P$ values adjusted for multiple comparisons: *$P$ < 0.05, **$P$ < 0.01, Mann–Whitney two-tailed test. Exact $P$ values are provided as numerical source data. **f**, Immunohistochemical characterization of normal dorsal skin and representative tumours in **a**. Antigens indicated are stained pink (Vector Red substrate). **g**, Immunohistochemical analysis as in **f**, but demonstrating macroH2A1 and macroH2A2 staining in normal skin in WT and dKO melanoma. For **f** and **g**, insets are shown at additional ×4 magnification. Staining was repeated on $n_{WT\,skin}$ = 2, $n_{WT\,melanoma}$ = 7, $n_{dKO\,melanoma}$ = 6 mice with similar results. Scale bars, 100 μm (**f**,**g**) or 1 cm (**a**). NS, not significant.

33 cell clusters, including melanocytes, immune cells and CAFs, as well as rarer cell populations (Fig. 3a). Cell types and states represented by each cluster were annotated on the basis of expression of known lineage markers (Extended Data Fig. 3a), similarity (Fig. 3b) or the most significant cluster-specific genes[43] (Supplementary Table 3). We performed spatial transcriptomics (ST) to visualize the distribution of tumour populations (spot clusters T1–T6) compared with normal regions (for example, epidermis) in their native tissue context (Fig. 3c and Supplementary Table 4). ST confirmed the identities we ascribed using scRNA-seq (Extended Data Fig. 3b) and the increased tumour area and invasiveness of the dKO tumours (Fig. 3c) as observed above (Fig. 1).

In the tumour compartment, melanocytes represented only ~5% of cells (Fig. 3d); however, ~25% of cells expressed the neural crest (NC) cell marker *Sox10* while lacking melanocyte markers or a *MITF* gene signature derived from a human melanoma scRNA-seq analysis[44] (Extended Data Fig. 3c–e). These 'NC' clusters (Fig. 3a) expressed genes associated with developmental precursors of mouse melanocytes and Schwann cells[45], including *Ngfr* (NC), *Foxd3* (migratory NC and Schwann cell precursors (SCPs)) and *Dhh* (SCPs) (Extended Data Fig. 3c), and occupied distinct tissue niches in the ST dataset (Extended Data Fig. 3b). The NC arrested cluster displayed a transcriptional profile (Extended Data Fig. 3c) and cell cycle distribution (Extended Data Fig. 3f) that was consistent with growth arrest. The NC clusters *Hapln1* and *Aqp1* expressed melanoma dissemination and NC cell migration

genes. Finally, the NC *Zeb2* and NC Ki-67 clusters were characterized by the transcription factor (TF) *Zeb2*, which is involved in the proliferation of melanoma cells before and following dissemination[46], along with *Gfra3*, which is associated with a NC stem cell signature and residual disease in human melanoma[47]. The NC arrested and *Aqp1* clusters expressed high levels of an AXL-driven programme associated with melanoma invasion[44], whereas the *Zeb2* and Ki-67 clusters expressed the highest levels of a melanocyte stem cell signature[48] (Extended Data Fig. 3d,e).

To dissect the relationship between NC cells and melanocytes, we performed cell trajectory analysis (Extended Data Fig. 3g,h). One branch incurred a growth suppressive programme in the NC arrested cluster, whereas a second branch dedifferentiated towards a state resembling the migratory and stem-like stages of NC development (NC *Aqp1* and *Zeb2*/Ki-67 clusters; Extended Data Fig. 3h), which are associated with poor prognosis in human melanoma[49]. This trajectory was supported by the ST data. As expected, melanocytes were present in the dermis and in T1 spots replacing normal dermis (Fig. 3c and Extended Data Fig. 3b). NC arrested cells were most abundant in T2 spots, whereas migratory NC *Aqp1* and *Hapln1* cells defined T3 spots. Meanwhile, T4–T6 spots harboured the bulk of NC *Zeb2* cells that invaded subcutaneous structures (Fig. 3c and Extended Data Fig. 3b). Relative to the total, dKO tumours were enriched in NC *Zeb2* cells to the detriment of NC arrested and *Hapln1* cells (Fig. 3d).

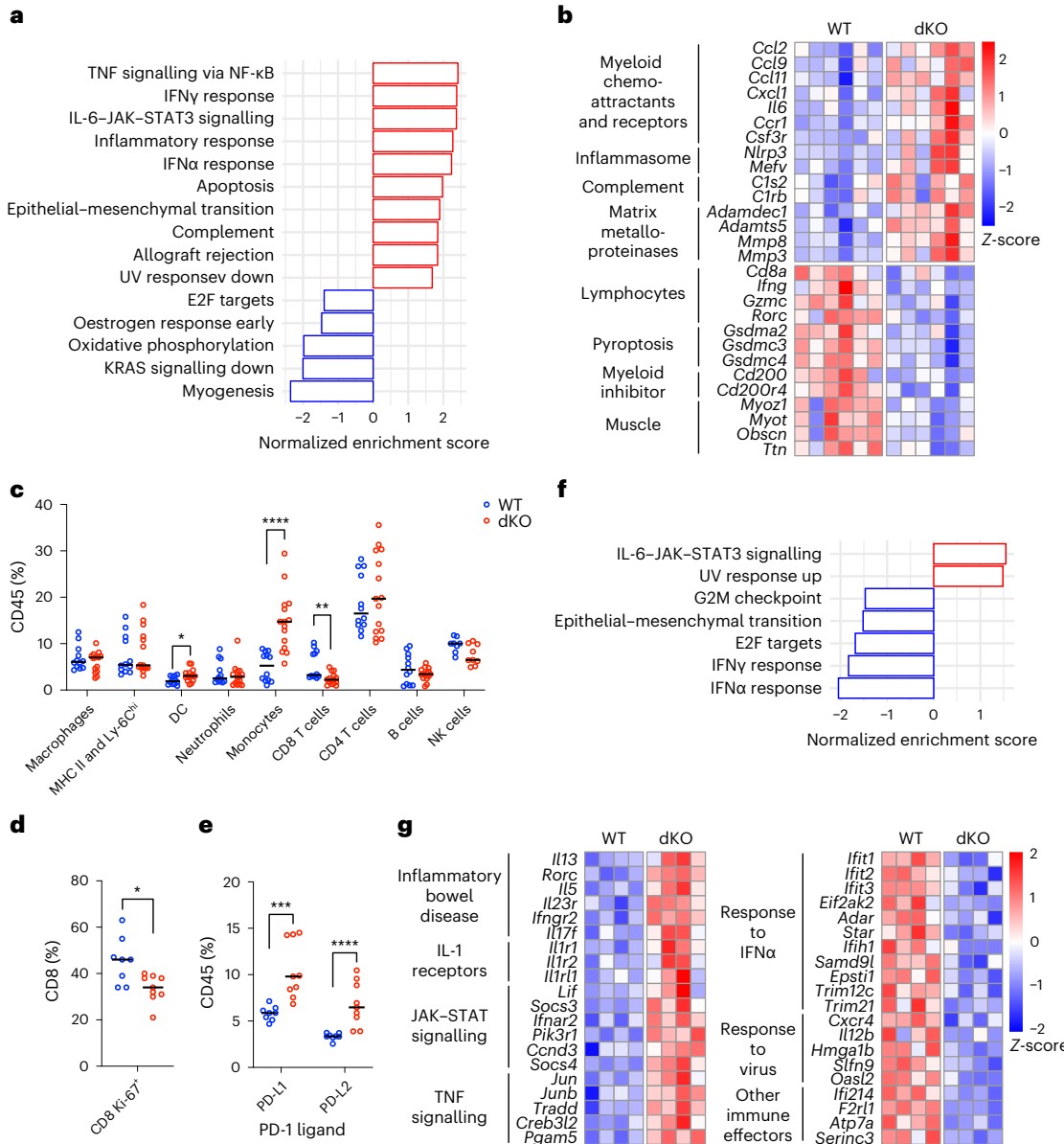

**Fig. 2 | MacroH2A-deficient melanomas deregulate genes associated with an immune anti-tumour response, and accumulate monocytes and dysfunctional CD8+ T cells. a**, GSEA of hallmark pathways performed on bulk RNA-seq of triallelic melanomas at 50 DPI. dKO versus WT comparison. The top ten significant (Benjamini–Hochberg adjusted $P < 0.05$) pathways are shown. Exact $P$ values are provided in Supplementary Table 1. **b**, Heatmap of DEGs in WT and dKO melanomas (bulk tumour), grouped under selected top gene enrichment terms defined using Homer analysis. Each column represents an independent tumour. Expression values are normalized row-wise as $Z$-scores. **c**, Quantification of indicated non-overlapping tumour-infiltrating immune cell populations at 50 DPI, determined by flow cytometry. $n_{WT} = 12$, $n_{dKO} = 15$ except for the natural killer (NK) cell population, for which $n_{WT} = 8$ and $n_{dKO} = 11$. DC, dendritic cells; MHC, major histocompatibility complex. **d**, Proliferative status

of CD8+ T cells in **c** assessed as a percentage of Ki-67 positivity by flow cytometry. **e**, Expression of PD-1 ligands on immune cells assessed as a percentage of PD-L1 or PD-L2 positivity by flow cytometry. For **d** and **e**, $n_{WT} = 8$, $n_{dKO} = 9$. For **c–e**, Mann–Whitney two-tailed test $P$ values shown: *$P < 0.05$, **$P < 0.01$, ***$P < 0.001$, ****$P < 0.0001$, with exact $P$ values provided as numerical source data. The centre line represents the median. Non-significant differences are not labelled. **f**, GSEA of hallmark pathways performed on RNA-seq of CD8+ T cells sorted by flow cytometry from melanomas at 50 DPI. dKO cells versus WT comparison. **g**, Heatmap of DEGs in WT and dKO melanoma-infiltrating sorted CD8+ T cells, grouped under selected top gene enrichment terms defined using Homer analysis. Each column represents target cells from an independent tumour. Expression values are normalized row-wise as $Z$-scores.

This result was confirmed using an unbiased approach that leveraged local cell abundance changes across conditions (Fig. 3e,f). Together, these data suggest that macroH2A deficiency promotes dedifferentiation in the NC compartment.

## Pro-tumour features of myeloid cells in the dKO TME

We identified multiple clusters of mononuclear phagocytes including macrophages (Mac), several of which expressed genes associated with

pro-tumour subtypes, including CD206 (*Mrc1*), *Arg1*, *Retnla*, *Fn1* and *C1qa*[50,51] (Fig. 3a,b and Extended Data Fig. 3i). By annotating against two mouse tumour model scRNA-seq datasets, we found that Mac *Mrc1*, which accumulated in the dKO (Fig. 3d), was fully encompassed by Mac_s2, tumour-enriched macrophages that become depleted following immune checkpoint blockade[52], and by Mac1, the human counterpart of which is associated with poor prognosis in patients with lung adenocarcinoma[53] (Extended Data Fig. 3j,k). These similarities

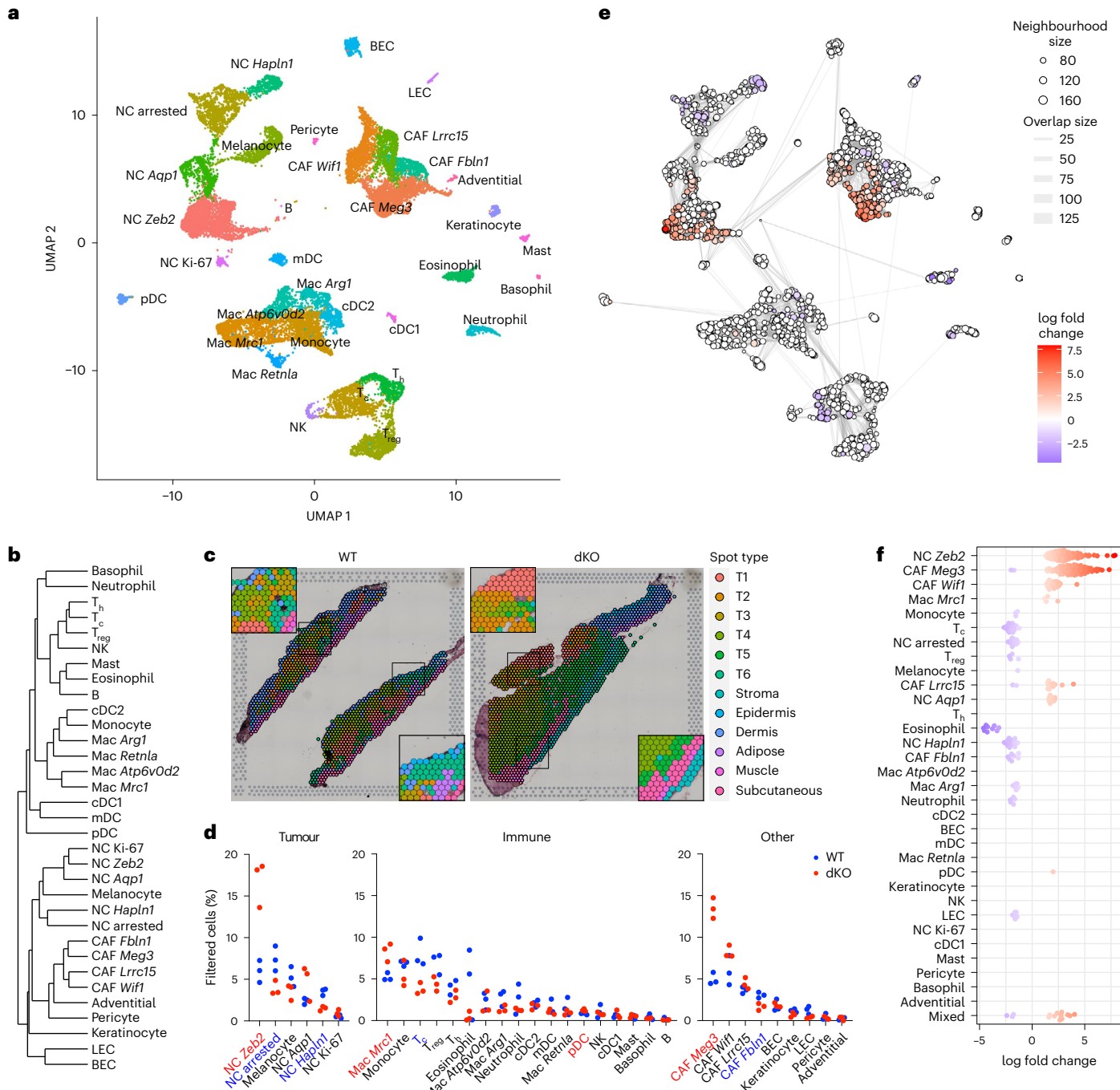

**Fig. 3 | scRNA-seq identifies dKO-associated remodelling of the NC-derived and immune compartments. a**, Dimension-reduced representation using uniform manifold approximation and projection (UMAP) of cell clusters in WT and dKO melanomas profiled by scRNA-seq at 35 DPI. Dots correspond to single cells from three independently processed tumours per genotype, coloured by cluster identity. **b**, Phylogenetic tree showing the degree of cell type and state similarity based on distances between clusters in principal component analysis space. See Supplementary Table 3 for a description of the cell-type acronyms used. **c**, Distribution of annotated spot types derived from ST analysis, overlaid on WT and dKO tumour histology. Insets are shown at ×2 magnification. **d**, Relative cell frequencies across clusters in individual melanomas profiled by scRNA-seq. Values shown are normalized to the total number of high-quality cells per sample included in the analysis. Names in colour represent clusters with

significant differences between WT and dKO frequencies (two-tailed unpaired *t*-test < 0.05); red indicates more abundant in dKO, whereas blue indicates more abundant in WT. *P* values are provided as numerical source data. **e**, UMAP representation of differential abundance analysis performed using Milo. Cells are grouped into overlapping neighbourhoods based on their *k*-nearest neighbour graph position, depicted as circles proportional in size to the number of cells contained, coloured by the log fold change of abundance between genotypes. The graph edge thickness is proportional to the number of cells shared between adjacent neighbourhoods. **f**, Bee swarm plot of significant differences in **e** showing distributions of abundance log fold changes between dKO and WT samples in neighbourhoods belonging to the indicated clusters as in **a**. For **e and f**, neighbourhoods with significant differential abundance at a 5% false discovery rate are coloured. In **f**, non-significant neighbourhoods are not shown.

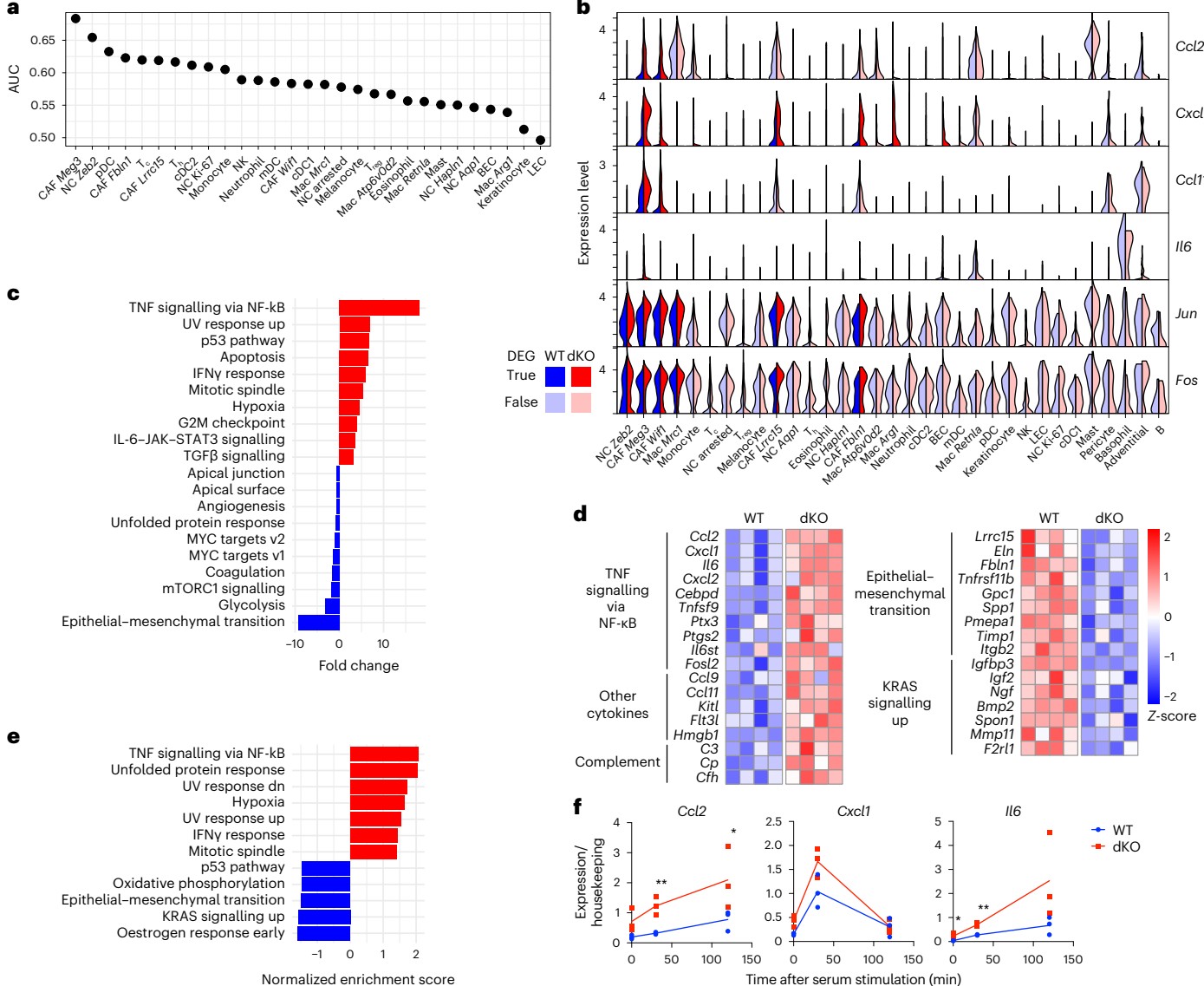

**Fig. 4 | Pro-inflammatory signals in dKO tumours originate from CAFs.**
**a**, Prioritization of the contribution of each cell cluster to gene expression changes in dKO versus WT samples using Augur, a method that measures the separation in gene expression space between cells in each cluster as a function of genotype. AUC, area under the curve. **b**, Genes of interest with significant upregulation in dKO samples in clusters highlighted in bold colours (Wilcoxon rank-sum test adjusted $P < 0.05$). $P$ values are provided in Supplementary Table 3. **c**, Significant hallmark pathways in a GSEA of dKO versus WT samples performed in the CAF *Meg3* cluster. **d**, Heatmap of DEGs in CAFs sorted by flow cytometry from WT and dKO melanomas at 50 DPI, grouped under selected top gene

enrichment terms defined using Homer analysis. Each column represents CAFs from an independent tumour. Expression values are normalized row-wise as $Z$-scores. **e**, Significant hallmark pathways in GSEA of sorted CAFs as in **d**. **f**, Expression normalized to housekeeping controls of indicated cytokine genes determined by reverse transcription-qPCR in cultured CAFs isolated from WT and dKO tumours at the indicated times following serum stimulation. Line represents the mean of three independently performed experiments shown. Ratio paired two-tailed $t$-test $P$ values shown: *$P < 0.05$, **$P < 0.01$. Exact $P$ values are provided as numerical source data. Non-significant differences are not labelled.

reinforce the finding that immunosuppressive myeloid cells accumulate in dKO tumours.

Lymphoid clusters recapitulated the decrease in cytotoxic T ($T_c$) cells in dKO tumours (Fig. 3d), as observed by flow cytometry (Fig. 2c). Through reclustering, we refined lymphoid cells such that the $T_c$ cell cluster split into a *Cd4*-positive CD4 circulating cluster (Extended Data Fig. 3l,m) and a bona fide CD8 cluster negative for *Cd4*, which was locally depleted in dKO tumours (Extended Data Fig. 3n). Overall, scRNA-seq corroborated our bulk RNA-seq and immunophenotyping data, providing support for the presence of increased immunosuppressive myeloid and decreased $T_c$ cell infiltration (Extended Data Fig. 3o) in dKO melanomas.

## CAFs produce pro-inflammatory signals in the dKO TME
We identified four CAF clusters (Extended Data Fig. 4a), which lacked distinction between inflammatory (iCAF), myofibroblastic (myCAF) and antigen-presenting CAFs (apCAF) reported in pancreatic cancer[54] (Extended Data Fig. 4b,c). This result suggests that there is distinct CAF origin or functional specialization across tumours. The CAF *Meg3* cluster exhibited an almost threefold increase in dKO tumours (Fig. 3d), and with NC *Zeb2*, this cluster was the most relatively enriched cell type (Fig. 3e,f). By computationally assessing the weight of each cluster, CAF *Meg3* was highlighted as the top driver of the dKO transcriptional profile (Fig. 4a). Moreover, we found a significant upregulation of the 'dKO tumour up' signature, which consisted of all upregulated genes in

the bulk RNA-seq dataset (Supplementary Table 2), across all dKO CAF clusters, as well as a subset of myeloid clusters (Extended Data Fig. 4d). Importantly, the upregulated cytokines in the bulk RNA-seq data (for example, *Ccl2*, *Cxcl1*, *Ccl11* and *Il6*) were significantly overexpressed in dKO CAF clusters (Fig. 4b). We additionally found upregulation of other immediate-early genes (for example, *Jun* and *Fos*), which was indicative of signal response pathway activation. This finding was confirmed by comparing our data to a signature comprising 139 immediate-early genes[55] (Extended Data Fig. 4e). GSEA revealed upregulation of inflammatory pathways, led by 'TNFα signalling via NF-κB' across all CAF clusters (Fig. 4c and Extended Data Fig. 4f), whereas NC *Zeb2*, the next highest cluster (Fig. 4a) did not (Extended Data Fig. 4g). Together with the increased CAF *Meg3* prevalence (Fig. 3d), these data suggest that CAFs are the primary source of the abovementioned cytokines in the TME and promote the dKO immunophenotype. Of note, whereas *H2afy* expression was readily detected across clusters, *H2afy2* was limited to CAFs (Extended Data Fig. 4h), which suggested that its loss contributes to a CAF-specific phenotype.

Next, we sorted CAFs from WT and dKO tumours by flow cytometry on the basis of CD140a (*Pdgfra*) expression (Extended Data Fig. 4i). RNA-seq analyses confirmed the upregulation of cytokines, among others, and activation of the TNF–NF-κB pathway (Fig. 4d,e and Supplementary Table 5). Differentially expressed genes (DEGs) in the sorted CAFs significantly overlapped with those identified by scRNA-seq in the combined mesenchymal populations (Extended Data Fig. 4j). We established pure cultures of sorted WT and dKO CAFs (Extended Data Fig. 4k–m). Given the inducible nature of cytokines, we stimulated primary CAF cultures with serum[56–58], followed by quantitative PCR (qPCR) and Luminex-based quantitation (Fig. 4f and Extended Data Fig. 4n). dKO CAFs expressed higher levels of *Ccl2*, *Cxcl1* and *Il6* at baseline, which were further induced after serum stimulation. In immortalized dermal fibroblasts (iDFs) derived from non-tumour bearing WT skin and dKO skin[11], similar results were observed (Extended Data Fig. 4o), which highlighted a conserved cell-intrinsic mechanism.

## Conserved macroH2A-dependent cytokine regulation in CAFs

CAFs can recruit immunosuppressive myeloid cells by secreting CCL2, IL-6 and CXCL1 (reviewed in ref. 59). Thus, we performed ligand–receptor analysis of the scRNA-seq dataset. CAFs had the most prolific outgoing interactions with other cell types both in the WT and dKO samples (Extended Data Fig. 5a), as previously described[60]. Differential analysis showed a generalized decrease in the number of interactions in the dKO samples, but an increase in the strength of communication from CAFs to NC cells (Extended Data Fig. 5a). Importantly, dKO CAFs increased signalling interactions to the mononuclear phagocyte lineage through

the CCL2 and IL-6 pathways, and to neutrophils and basophils through CXCL1 (Fig. 5a). Our ST data revealed that CAF *Meg3* cells formed a distinct layer (Fig. 5b), partially overlapping with subcutaneous spot types (Extended Data Fig. 3b). Mac *Mrc1*-enriched spots were enriched at the tumour periphery (Fig. 5b), and correlation analysis revealed a significant positive association between Mac *Mrc1* and *Meg3*, *Fbln1* and *Lrrc15* CAF subtypes (Fig. 5c), which suggested proximity. Importantly, these cell types shared a significant negative correlation with $T_c$ cells, a characteristic consistent with local T cell exclusion (Fig. 5b,c), which was probably driven by *Mrc1*+ myeloid cells. We confirmed the chemoattractant properties of CAFs in vitro by measuring the migration of WT bone-marrow-derived monocytes towards WT or dKO CAFs through Transwell assays. MacroH2A dKO CAFs displayed significantly higher monocyte recruitment at later time points (Fig. 5d and Extended Data Fig. 5c), which bolstered our finding of increased monocyte-derived cells in the dKO TME.

Next, we predicted immune cell abundance in macroH2A high and low tumours from the melanoma cohort of The Cancer Genome Atlas (TCGA SKCM). MacroH2A1[low] and macroH2A2[low] primary and metastatic samples were associated with significantly reduced CD8 T cell scores (Fig. 5e and Extended Data Fig. 5d). In primary tumours, M2 (pro-tumour) macrophages were significantly associated with macroH2A2[low] and trended for macroH2A1[low] tumours (Fig. 5e). Some myeloid subtypes were negatively correlated with macroH2A2 in metastatic tumours (Extended Data Fig. 5d), although macroH2A[low] tumours appeared overall depleted of immune cells (Extended Data Fig. 5e), which affected our ability to detect relative increases in immune subtypes. We also identified key $T_c$ cell marker genes, *CD8A* and *IFNG*, and components of the tumour cytolytic activity score, *GZMA* and *PRF1* (ref. 61), as significantly downregulated in macroH2A[low] tumours (Extended Data Fig. 5f). These correlations highlight anti-tumour immunity dysfunction in human macroH2A[low] melanomas.

Examining human melanoma-derived primary CAF cultures (Extended Data Fig. 5g) revealed homogenous levels of macroH2A1 protein, whereas macroH2A2 spanned almost an order of magnitude (Fig. 5f and Extended Data Fig. 5h). MacroH2A2[low] CAFs secreted more CCL2, CXCL1 and IL-6 when stimulated (Fig. 5g). Although not significant owing to the low sample size (Extended Data Fig. 5h), we analysed a large pan-cancer scRNA-seq dataset[60] comprising over 56,000 CAFs across 98 samples. The majority of CAFs had no detectable *MACROH2A2* counts (Extended Data Fig. 5i), so pseudobulk expression values per tumour were calculated. Whereas *CCL2*, *CXCL1* and *IL6* were positively correlated to each other (Extended Data Fig. 5j), the negative correlation between *IL6* and *MACROH2A2* was significant (Extended Data Fig. 5j). This result suggests that the relationship between macroH2A2 and inflammatory signalling is conserved in human CAFs.

**Fig. 5 | CAFs are the source of pro-inflammatory signals in the dKO TME.**
**a**, Comparison of signalling probability along CCL, CXCL and IL-6 pathways leveraging scRNA-seq data from CAF *Meg3* cells to myeloid cell clusters. Dots represent significant ligand–receptor interaction pairs with increased signalling in the dKO. The dot colour represents communication probability, the dot size represents the *P* value of one-sided permutation test, and the absence of a dot signifies a null probability of signalling. Exact *P* values are provided as numerical source data. **b**, Prediction of the spatial localization of indicated scRNA-seq cell clusters in WT and dKO melanoma by label transfer onto ST data. Insets are shown at ×2 magnification. **c**, Correlation analysis of cell-type scores for all scRNA-seq clusters detected in SC data, based on the combined set of WT and dKO spots. Dots shown correspond to significant correlations (two-tailed *t*-test adjusted for multiple comparisons, adjusted *P* < 0.05), heatmap colour corresponds to Pearson's correlation coefficient. Exact *P* values are provided as numerical source data. Black squares represent hierarchical clusters of cell types based on correlation coefficients. **d**, Transwell assay measuring the migration of CMFDA-labelled WT bone-marrow-derived monocytes towards unlabelled WT or dKO CAFs. Monocyte counts are normalized to the CCL2 condition at 24 h.

Summary of three independent experiments using different monocyte donors shown. Error bars represent s.e.m. Two-tailed *t*-test *P* values shown: *$P$ < 0.05, **$P$ < 0.01. Exact *P* values are provided as numerical source data. Non-significant differences are not labelled. **e**, Comparison of deconvoluted immune cell type scores between TCGA primary melanoma tumours with *MACROH2A1* and *MACROH2A2* high and low expression levels. Wilcoxon rank-sum test *P* values adjusted for multiple comparisons shown: *$P$ < 0.05, **$P$ < 0.01. Exact *P* values are provided as numerical source data. *N* = 35 biologically independent samples per category. The box plot centre line represents the median, the box plot limits indicate the 25th to 75th percentiles, the whiskers extend from the box limit to the most extreme value no further than 1.5× the inter-quartile range from the box limit, any data beyond whiskers are plotted as individual points. **f**, MacroH2A1 and macroH2A2 levels in a panel of 11 human melanoma CAF lines. Histone H3 was used as a control for total histone content. **g**, Indicated cytokine levels in CAF lines in **f**, stratified according to macroH2A2 levels along the median. $N_{high}$ = 6, $n_{low}$ = 5 biologically independent samples. The western blot was repeated three times.

## MacroH2A-regulated genes are enriched in super-MCDs

CUT&RUN analyses of macroH2A1 was performed in WT cultured CAFs (Extended Data Fig. 6a). As previously reported[62], macroH2A was excluded from the bodies of highly expressed genes and retained at lowly expressed ones (Extended Data Fig. 6b). We identified macroH2A1 chromatin domains (MCDs)[62], which we partitioned into 'super' and 'standard' classes on the basis of enrichment and size (Extended Data Fig. 6c–e). Genome-wide, macroH2A1 was enriched at significant DEGs compared with a control set of static genes with matched expression (Fig. 6a). These DEGs also significantly overlapped MCDs (Fig. 6b). Notably, the 39 significantly upregulated inflammatory genes

(Supplementary Table 5) showed higher average macroH2A1 occupancy compared with static genes (Fig. 6c).

Intergenic enrichment of macroH2A suggested that it may regulate *cis*-regulatory elements. Traditional enhancers (TEs) and superenhancers (SEs) are emerging regulators of NF-κB-driven inflammatory gene transcription[63–67]. Moreover, macroH2A regulates specific promoter–enhancer contacts[68] and suppresses a subset of enhancers[69]. Therefore, we performed chromatin immunoprecipitation with sequencing (ChIP–seq) for H3K27ac, which marks active promoters (Extended Data Fig. 7a) and enhancers, in serum stimulated, cultured CAFs. The dKO CAFs revealed increased H3K27ac levels at 4,043 TEs and 73 SEs and

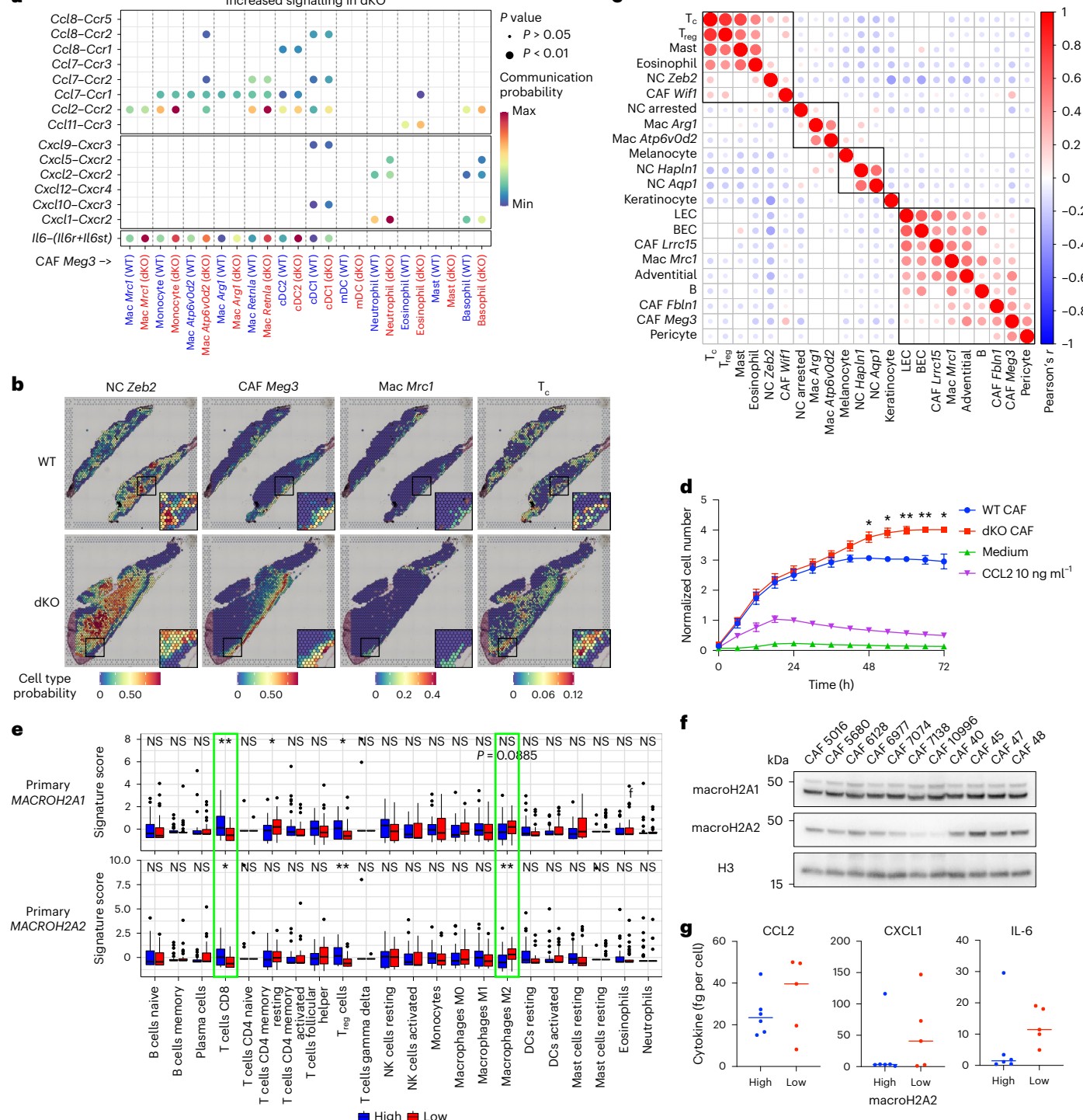

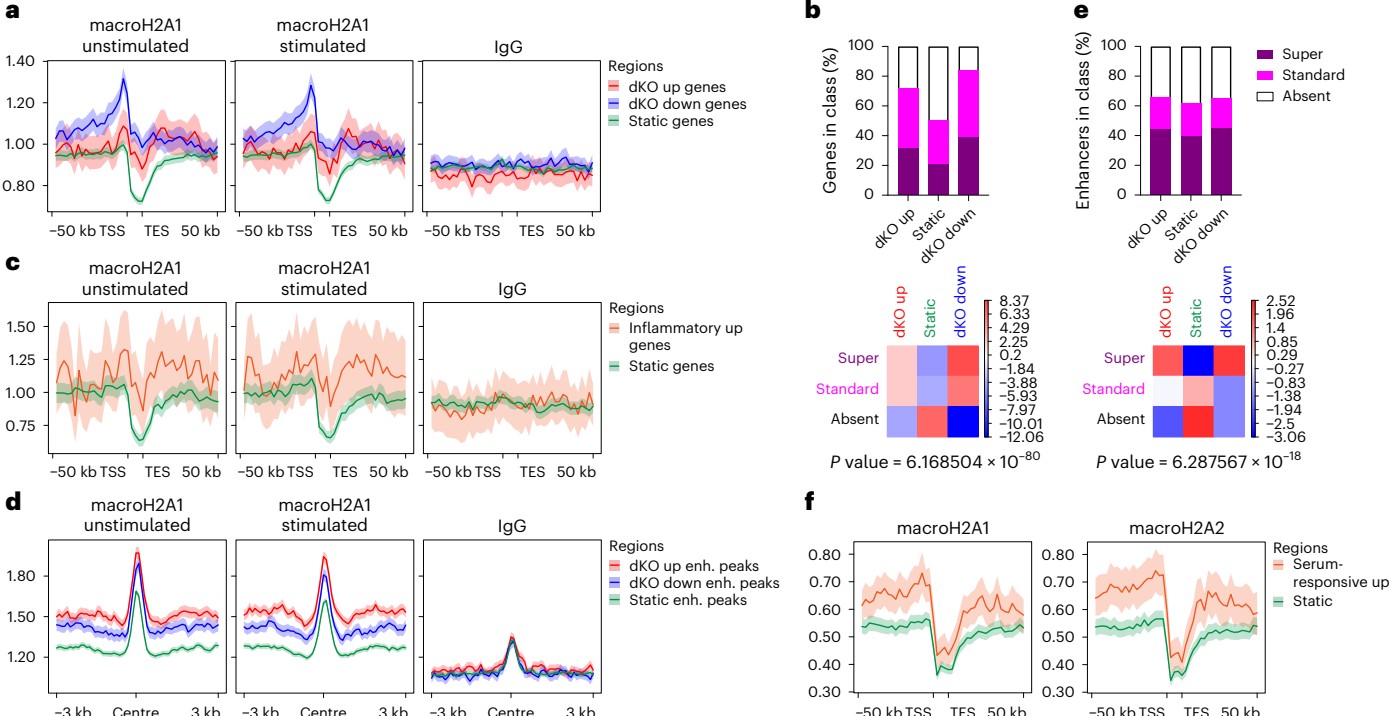

**Fig. 6 | MacroH2A-sensitive genes and enhancers occupy highly enriched macroH2A chromatin domains. a**, Metagene profile of macroH2A1 CUT&RUN signals in cultured WT CAFs before and after serum stimulation at genes differentially up or down or static genes of matched expression levels in dKO versus WT sorted CAFs. $n_{\text{dKO up}} = 357$, $n_{\text{dKO down}} = 884$, $n_{\text{Static}} = 3{,}708$. TES, transcription end site; TSS, transcription start site. **b**, Top, percentage of overlap between DEGs and MCDs. Bottom, deviation from random distribution shown as a heatmap of Chi-square test residuals, together with the associated $P$ value. **c**, As in **a**, but at inflammatory genes upregulated in dKO sorted CAFs and static genes of matched expression levels. $n_{\text{Inflammatory up}} = 39$, $n_{\text{Static}} = 385$. **d**, Average profile of macroH2A CUT&RUN signals in cultured WT CAFs before and after serum

stimulation centred around ATAC peaks located in enhancers (enh.) that gain, lose or maintain static H3K27ac levels in dKO versus WT tumours. $n_{\text{dKO up}} = 6{,}659$, $n_{\text{dKO down}} = 5{,}211$, $n_{\text{Static}} = 18{,}961$. Note the signal pattern at the centre of the ATAC peak, which is probably due to a bias of CUT&RUN for accessible chromatin sites. **e**, As in **b**, but for overlap between enhancers with peaks shown in **d** and MCDs. **f**, Average profile of macroH2A1 and macroH2A2 signals in dermal fibroblasts[62] analysed by ChIP-seq at genes hyperinduced by serum stimulation in the absence of macroH2A and static genes of matched expression levels. $n_{\text{Serum-responsive up}} = 139$, $n_{\text{Static}} = 695$. For **a**, **c**, **d** and **f**, mean signal value and 95% confidence interval as determined by bootstrap analysis are shown.

decreased H3K27ac at 2,574 TEs and 78 SEs (Extended Data Fig. 7b–d). Similar to DEGs, differentially activated enhancers (DAEs) were significantly enriched in macroH2A and preferentially located in super-MCDs (Fig. 6d,e). Moreover, several significantly upregulated inflammatory genes were located within 50 kb of TEs and SEs that gained H3K27ac (Extended Data Fig. 7e).

Chromatin accessibility, however, was only minimally affected in the dKO CAFs (667 up and 668 down of >140,000 detected peaks; Extended Data Fig. 7f). Nonetheless, motif analysis of the dKO up peaks detected using assay for transposase-accessible chromatin with sequencing (ATAC–seq) highlighted significant enrichment of TFs involved in inflammatory signalling, including NF-κB (Extended Data Fig. 7g). The differentially accessible regions displayed concordant changes in H3K27ac (Extended Data Fig. 7h) but remained small compared with DAEs (Extended Data Fig. 7c).

We next utilized iDFs to query whether macroH2A regulates inducible genes beyond CAFs. We performed RNA-seq before and after serum stimulation and found that TNF signalling through the NF-κB pathway was significantly upregulated in the dKO samples following stimulation (Extended Data Fig. 8a and Supplementary Table 6). The upregulated serum-responsive genes ($n = 139$; Extended Data Fig. 8b,c) were preferentially enriched in macroH2A1 and macroH2A2, as measured in WT DFs[62], relative to static genes (Fig. 6f). Generally, DEGs and DAEs were enriched in macroH2A variants in WT fibroblasts compared with static regions, regardless of the change in direction (Extended Data

Fig. 8d–g). Thus, macroH2A variants may act as both repressors and activators, as previously reported[7,8,68,70–72].

**MacroH2A loss leads to rewired chromatin looping**

Emerging evidence suggests that macroH2A regulates 3D genome organization[68,73–76]. To assess this possibility, and to annotate functional promoter–enhancer pairs, we performed Micro-C[77] coupled with promoter capture (pcMicro-C)[78] in CAFs (Extended Data Fig. 9a). At 10 kb resolution, we identified a similar number of promoter-originating loops in WT and dKO CAFs, with similar size and score distributions (Fig. 7a and Extended Data Fig. 9b). Distal loop ends were enriched for open chromatin and active enhancers (Extended Data Fig. 9c), which validated the functional nature of the contacts.

A comparison of loop coordinates revealed moderate overlap between WT and dKO CAFs (Fig. 7a), which suggested that there was genome-wide reorganization of promoter contacts. However, these changes were also driven by small shifts in the distal ends of loops to an adjacent 10 kb bin (Extended Data Fig. 9d). By examining the activity of enhancers located at distal loop ends, we found that gain of H3K27ac in the dKO samples was more frequently associated with unique dKO loops (and loss of H3K27ac with unique WT loops) compared with static enhancers (Fig. 7b and Extended Data Fig. 9e). This association in the dKO samples was highly significant for enhancers within super-MCDs (Fig. 7b), which suggested that macroH2A instructs functional looping properties of the chromatin fibres it is highly enriched in. Similarly, by

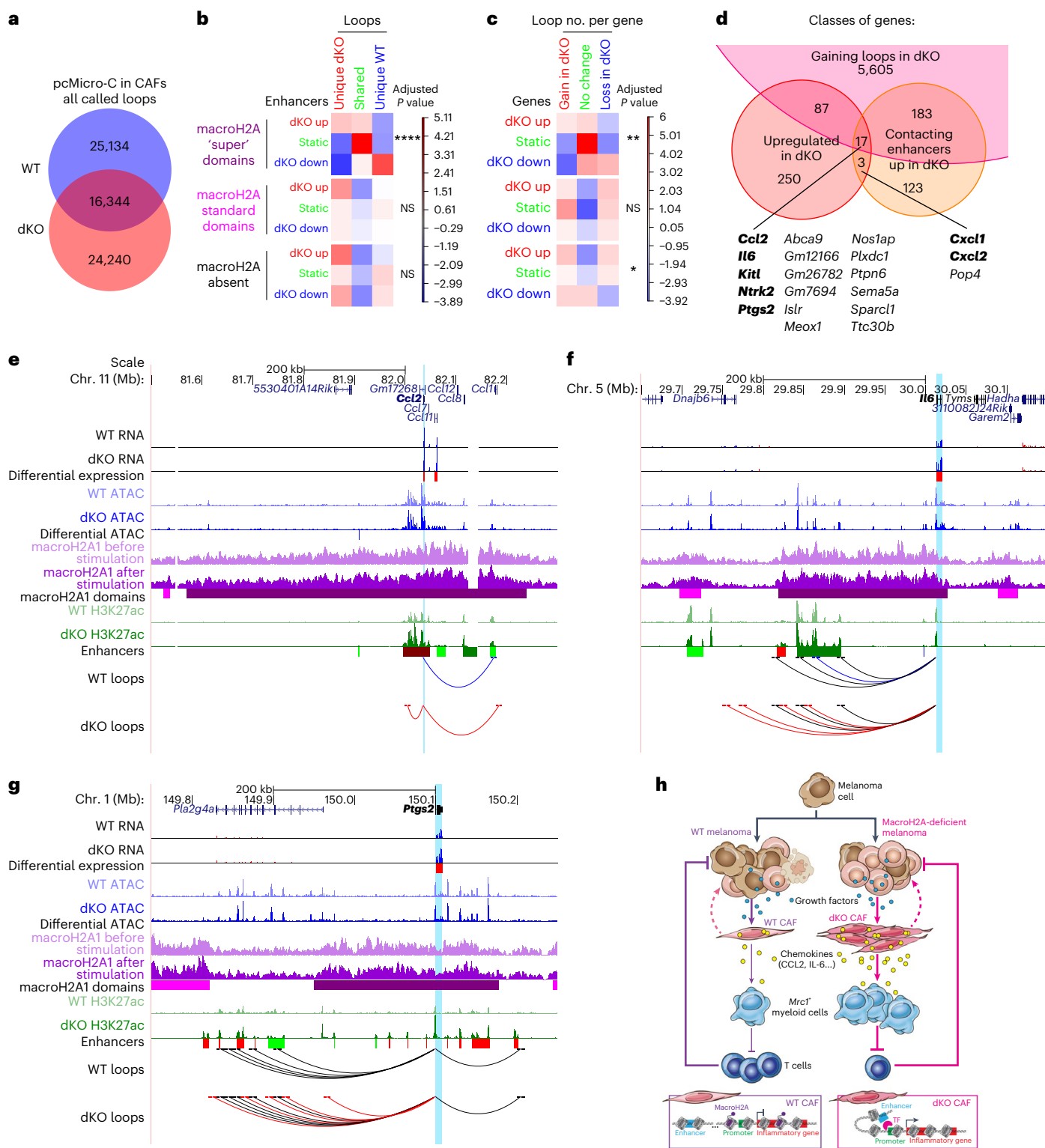

associating changes in the total number of loops originating in each gene with DEGs, the overlap reached the highest significance and level of directional correlation in macroH2A super-MCDs (Fig. 7c and Extended Data Fig. 9f).

Because loop changes often did not match those in gene or enhancer activity (Fig. 7b,c), we intersected genes upregulated in dKO CAFs with bona fide target genes of enhancers gaining H3K27ac, as determined by looping data. Although only 20 genes fit these stringent criteria, 17 had a net increase in the number of loops in dKO CAFs, and

7 were part of inflammatory signalling (Fig. 7d), with significant enrichment for NF-κB targets (Supplementary Table 7). Loci such as *Ccl2, Il6, Cxcl1, Ptgs2* (the predominant prostaglandin-endoperoxide synthase in CAFs) and *Kitl* (which encodes Kit ligand/stem cell factor) were located within super-MCDs (Fig. 7e–g and Extended Data Fig. 9g,h). *Ccl2* and *Il6* gained loops to a hyperacetylated enhancer within their respective super-MCDs, whereas *Ptgs2* and *Kitl* gained loops to more distal hyperacetylated enhancers. *Cxcl1* displayed a net loss of loops in dKO CAFs but maintained contact with an enhancer-gaining H3K27ac.

**Fig. 7 | DAEs and DEGs acquire changes in chromatin looping in dKO tumours. a**, Extent of overlap between chromatin loops detected in WT and dKO CAFs after serum stimulation at 10 kb resolution. **b**, Chi-square test of independence evaluating the association between changes in H3K27ac level at enhancers and gains or losses of loops to these enhancers in the absence of macroH2A. Combinations of loop and enhancer status are stratified according to the position of enhancers with respect to MCDs. **c**, As in **b**, but for changes in gene expression and in the total number of loops per gene. For **b** and **c**, $P$ values adjusted for multiple comparisons shown: *$P < 0.05$, **$P < 0.01$, ****$P < 0.0001$. Exact $P$ values are provided as numerical source data. **d**, Overlap between genes upregulated, in contact with enhancers gaining H3K27ac, and with net loop gains in dKO tumours. Genes in bold are associated with inflammatory signalling pathways according to HOMER analysis. **e**, University of California Santa Cruz (UCSC) genome browser screenshots of the *Ccl2* locus and its chromatin environment showing indicated transcriptomic and epigenomic features. Bars under RNA-seq and ATAC–seq tracks indicate significantly upregulated (red) or downregulated (blue) genes or accessible regions in dKO versus WT sorted CAFs.

Bars under macroH2A CUT&RUN tracks indicate 'super' (purple) and 'standard' (magenta) macroH2A chromatin domains. Below H3K27ac tracks, bright and dark bars indicate TEs and SEs, respectively; red, blue and green denote gain, loss and no change, respectively of H3K27ac level in dKO versus WT CAFs. Chromatin loops, originating at the promoter of the highlighted gene, are shown in red if specific for the dKO, blue for the WT, and black if shared. **f**, As in **e**, but for the *Il6* locus. **g**, As in **e**, but for the *Ptgs2* locus. **h**, Model of the impact of macroH2A loss on the melanoma TME. In the absence of macroH2A, inflammatory genes in CAFs become intrinsically hyperinducible owing to increased enhancer activity and promoter–enhancer looping. This leads to an increased production of pro-inflammatory cytokines by CAFs, which in turn attract *Mrc1*+ myeloid cells with a pro-tumour phenotype. Accumulating myeloid cells inhibit CD8+ T-cell-mediated tumour cell killing, which results in increased tumour size in dKO animals. CAF-driven inflammatory signalling could also promote melanoma dedifferentiation through mechanisms that are yet to be determined (dashed lines). Illustration in **h** by Jill K. Gregory, reproduced with permission from © Mount Sinai Health System.

Altogether, these results demonstrate that macroH2A represses inflammatory genes in CAFs by restricting enhancer contacts and/or activity.

## Discussion

Our study of a macroH2A-deficient melanoma mouse model revealed an unappreciated role for this histone variant in the TME. By profiling macroH2A-dependent chromatin looping, we identified widespread changes in the promoter–enhancer interaction landscape. Together with changes in enhancer activity, this finding suggests that macroH2A may enforce the position of enhancers relative to nuclear compartments or the ability of enhancers to interact with their cognate promoters. Accordingly, previous studies have suggested that macroH2A affects 3D chromatin organization at several scales, including stabilizing nucleosome–DNA contacts to limit the mobility of the chromatin fibre[73,76], associating with the boundaries of lamina-associated domains and promoting heterochromatin[74,75], blocking BRD4 binding at macroH2A-bound enhancers[69], and changing contact frequency of promoter–enhancer pairs, the activity of which is affected by macroH2A1 or macroH2A2 depletion[68]. Interestingly, however, changes in chromatin accessibility were minimal, which suggests that macroH2A loss does not affect chromatin remodelling, as previously reported[79].

In our model, macroH2A-dependent regulatory mechanisms converged on a small set of inflammatory genes, which underscores a role for macroH2A in limiting inflammatory signalling in vivo. Notably, CAFs hijack these inflammatory genes as a mechanism of tumour immune escape[80–83], and our studies suggest that dKO CAFs promote an immunosuppressive environment that leads to increased melanoma burden (Fig. 7h).

MacroH2A deficiency in cancer promotes tumour growth through multiple mechanisms[1]. Our previous report that macroH2A blocks the proliferative and metastatic capacity of melanoma cells[18] aligns with the increased size of primary tumours observed in dKO mice. Here, we revealed that macroH2A suppresses dedifferentiation along the NC lineage towards a state associated with advanced disease and poor prognosis[44,47,49], immune evasion and immunotherapy resistance[84–87]. We uncovered a conserved role for macroH2A in mouse and human melanoma CAFs, which produce increased cytokines when macroH2A levels are low. This phenotype appears CAF intrinsic, although we cannot exclude the possibility that a hyperinducible response to stimuli occurs in other macroH2A-deficient cells of the TME. Furthermore, given that macroH2A loss decreases the frequency of the CAF *Fbln1* cluster (Fig. 3d), which expresses myofibroblast markers (Extended Data Fig. 4a), and downregulates the myofibroblast-associated genes *Lrrc15* and *Fbln1* (refs. 88,89) in sorted CAFs (Fig. 4d), we propose that the increased inflammation observed is a consequence of skewed dermal fibroblast polarization towards iCAFs at the expense of myCAFs[54].

Inflammatory signalling was among the first identified microenvironmental cues that induce melanoma dedifferentiation[84,90,91], which raised the possibility that CAFs not only recruit immunosuppressive myeloid cells but may also promote melanocyte dedifferentiation. Such crosstalk occurs in colorectal cancer, in which *Ptgs2* expression in CAFs drives the expansion of tumour-initiating stem cells in a paracrine manner[92]. We speculate that the convergence of these mechanisms would predict poor response of macroH2A^low tumours to immunotherapy, with potential to stratify patients.

## Online content

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

[1]Department of Oncological Sciences, Icahn School of Medicine at Mount Sinai, New York, NY, USA. [2]Department of Dermatology, Icahn School of Medicine at Mount Sinai, New York, NY, USA. [3]Tisch Cancer Institute, Icahn School of Medicine at Mount Sinai, New York, NY, USA. [4]Bioinformatics for Next Generation Sequencing Facility, Icahn School of Medicine at Mount Sinai, New York, NY, USA. [5]Precision Immunology Institute, Icahn School of Medicine at Mount Sinai, New York, NY, USA. [6]Department of Pathology, Icahn School of Medicine at Mount Sinai, New York, NY, USA. [7]Department of Genetics and Genomic Sciences, Icahn School of Medicine at Mount Sinai, New York, NY, USA. [8]Graduate School of Biomedical Sciences, Icahn School of Medicine at Mount Sinai, New York, NY, USA. [9]Department of Pharmacology, Physiology and Cancer Biology, Sidney Kimmel Cancer Center, Thomas Jefferson University, Philadelphia, PA, USA. [10]Institut Curie, INSERM, U932, and PSL Research University, Paris, France. [11]Black Family Stem Cell Institute, Icahn School of Medicine at Mount Sinai, New York, NY, USA. ✉e-mail: dan.filipescu@mssm.edu; emily.bernstein@mssm.edu

## Methods

### Mouse melanoma model

All animal experiments received previous ethical and technical approval from the Icahn School of Medicine at Mount Sinai (ISMMS) Institutional Animal Care and Use Committee (protocol number LA11-00122). Humane end points for tumour induction studies were inability to breathe, eat, drink or move normally, behavioural abnormalities, body condition score <3, tumour size >1,000mm³, tumour ulceration and loss of body weight greater than 20%. Maximal tumour size was not exceeded before the end point of experiments in this study. *Mus musculus* macroH2A WT and dKO inducible melanoma strains were generated by breeding B6;FVB-*Tg(Tyr-cre/ERT2)13Bos Braf^tm1Mmcm Pten^tm1Rdp*(ref. 22) and 129S6.Cg-*Macroh2a2^tm1.1Peh Macroh2a1^tm1Peh* (ref. 10) or 129S6/SvEvTac (Taconic) mice. Besides decreased body mass and reduced reproductive efficiency, the 129S dKO strain lacks spontaneous disease phenotypes[10]. The *Braf^CA Pten^fl Tyr-CreERT2* strain (C57BL/6-FVB background) develops spatially and temporally inducible[22], polyclonal melanomas arising in quiescent amelanotic precursors and subsequent expansion of pigmented melanocytes[93,94]. Mice were housed in a facility with specific pathogen-free health status, in individually ventilated cages at 21–22 °C and 39–50% relative humidity, on a 7:00–19:00 light cycle, with free access to food and water. Genotyping was performed as previously described[10,95]. Mouse lines were maintained on a mixed background: on average, WT and dKO mice are predicted to have a 75% 129S background, with the remainder a mix of B6 and FVB present in the original triallelic melanoma strain. Cre-positive *Braf^CA/CA Pten^fl/fl* and Cre-positive *Braf^CA/CA Pten^wt/wt* mice of both sexes were used for melanoma and nevus induction, respectively. Differences in tumour growth between sexes were tested for and found to be non-significant (Extended Data Fig. 1b). For tumour growth and histology experiments, Cre-mediated recombination of *Braf* and *Pten* alleles was induced in 7–11-week-old mice through the application of 1 μl of 5 mM 4-hydroxytamoxifen (70% Z-isomer, Sigma H6278, dissolved in ethanol) on the medial dorsal skin 24 h after hair removal with depilatory cream. Tumour length and width were measured from 25 DPI onwards using calipers, and the area was calculated assuming an elliptical tumour shape. Tumour thickness was approximated from the end point tumour area and weight, assuming an elliptic cylinder shape and tissue density = 1. Tumours were collected no later than 50 DPI before reaching a humane end point (tumour length over 1 cm, presence of ulceration). Five WT and 10 dKO male and 7 WT and 5 dKO female mice aged 9–10 weeks at induction were used for tumour immunophenotyping at 50 DPI. Three WT and 5 dKO male and 5 WT and 4 dKO female mice aged 9–10 weeks at induction were used for peripheral blood immune cell counts at 50 DPI. Two WT and 1 dKO male and 4 WT and 5 dKO female mice aged 10–11 weeks at induction were used for tumour immunophenotyping at 35 DPI. Two WT and 3 dKO male and 4 WT and 3 dKO female mice aged 7 weeks were used for peripheral blood immune cell counts in the tumour-naive setting. Two females and 2 males each for WT and dKO mice aged 9 weeks at induction were used for CD8 T cell sorting at 50 DPI for RNA-seq. Two females and 2 males each for WT and dKO mice aged 10 weeks at induction were used for CAF sorting at 50 DPI for RNA-seq. Two WT and 2 dKO females aged 10 weeks at induction were used for CAF sorting at 50 DPI for ATAC–seq. One WT and 1 dKO male aged 13 weeks at induction were used for CAF sorting at 50 DPI for culturing. Females were used for scRNA-seq to avoid the potential impact of sex-specific transcripts on integration. Pairs of WT and dKO age-matched females were induced at 10, 12 and 9 weeks in 3 independent experiments. One WT and 1 dKO female were induced at 9 weeks and processed at 35 DPI for ST analyses.

### Histology

Resected tissue was fixed in neutral buffered formalin for 24 h, paraffin embedded and sectioned at the ISMMS Biorepository and Pathology Core. Antigens were retrieved in citrate buffer (pH 6) in a domestic pressure cooker for 5 min. Immunodetection was performed using ImmPRESS polymer, ImmPACT Vector Red and NovaRed kits (Vector). Primary antibodies are listed in Supplementary Table 8. Ten random ×40 objective fields within the tumour were scored by board-certified dermatopathologists at ISMMS (M.S.G. and N.S.V.) for mitotic cells (H3S10ph) and Ki-67. Pagetoid scatter was measured as the number of melanoma cells within the epidermis across ten ×40 fields. The degree of pigmentation was estimated on sections stained with haematoxylin and eosin as the fraction of tissue area containing pigment using a ×2 objective.

### Tumour dissociation

Resected BRAF^V600E/PTEN-deficient melanomas were cut into 1–2 mm fragments. For immunophenotyping and CD8⁺ T cell sorting, tumour fragments were digested in RPMI containing 400 U ml⁻¹ collagenase IV (Gibco), 100 U ml⁻¹ hyaluronidase (Sigma) and 100 μg ml⁻¹ DNAse (Roche) at 37 °C for 1 h[96], aspirated 5 times through a 14 G needle, and filtered through a 70 μm cell strainer. Immune cells were enriched by centrifugation through a discontinuous 40/90 Percoll (GE Healthcare Life Sciences) gradient[97]. For scRNA-seq and CAF sorting, tumour fragments were digested using a Tumour Dissociation kit, mouse (Miltenyi) in DMEM using the soft/medium protocol according to the manufacturer's instructions. For scRNA-seq specifically, red blood cell lysis was performed in ACK (ammonium–chloride–potassium) buffer for 5 min on ice, followed by a wash in 1× DPBS containing 0.04% BSA.

### Flow cytometry

Before intracellular staining for IFNγ, TNF and FOXP3, cells were stimulated with 100 ng ml⁻¹ PMA (Sigma-Aldrich) and 0.5 μg ml⁻¹ ionomycin (Sigma-Aldrich) in the presence of 10 μg ml⁻¹ brefeldin A (Sigma-Aldrich) for 4 h. Staining was performed as previously described[97] using the fluorophore-conjugated antibodies listed in Supplementary Table 8. Samples were acquired and sorted on LSRFortessa and FACSAria SORP cytometers (BD Biosciences), respectively, running FacsDiVa (v.8.0.2; BD Biosciences) at the ISMMS Flow Cytometry Core. Data were analysed using FCS Express (v.7.12; De Novo Software). Cell population frequencies were compared using a Mann–Whitney test in Prism (v.9.5.1) software (GraphPad).

### RNA extraction

Snap-frozen tumours were disrupted in QIAzol reagent (Qiagen) by milling with zirconia beads (Fisher). Sorted cells were centrifuged and resuspended in QIAzol. Cultured cells were lysed directly in their culture vessel. After the addition of chloroform (Sigma) and centrifugation, RNA was isolated from the aqueous phase using an RNEasy Mini (tumours and cultured cells) or Micro (sorted cells) kit (Qiagen).

### RNA-seq library preparation

For tumours (*n* = 6 WT, 6 dKO), poly-A enrichment was carried out on 1 μg total RNA with a NEBNext Poly(A) mRNA Magnetic Isolation Module (NEB) followed by library preparation with a NEXTFLEX Rapid Directional RNA-seq kit (PerkinElmer). For sorted CAFs (*n* = 4 WT, 4 dKO), NEXTFLEX poly(A) beads 2.0 and a NEXTFLEX Rapid Directional RNA-seq kit 2.0 (PerkinElmer) were used, starting from 80 ng RNA isolated from 67,000–200,000 cells. For iDFs (two experimental replicates each of WT and dKO before and after stimulation), the same protocol was applied starting from 1 μg RNA. For CD8⁺ T cells (*n* = 4 WT, 4 dKO), 5 ng RNA from 5,000 sorted cells was used as input for a NEBNext Single Cell/Low Input RNA Library Prep kit for Illumina (NEB). Libraries were sequenced in 75 bp single-end mode on NextSeq 500 systems (Illumina).

### RNA-seq analysis

Reads were quasi-mapped to the Gencode M25 (GRCm38.p6) gene set using salmon (v.1.2.1)[98], and DEGs were called using DESeq2 (v.1.36.0)[99]

by filtering for an independent hypothesis weighting[100] adjusted *P* value of <0.05 and a log$_2$ fold change of >0.75 or < −0.75. GSEA was performed with the fgsea (v.1.22.0) package[101] on the entire expressed gene set pre-ranked based on the Wald test statistic computed using DESeq2. Significant pathways were reported using an adjusted *P* value cutoff of 0.05 and ordered on normalized enrichment scores. Gene ontology enrichment was performed using HOMER (v.4.10)[102]. Heatmaps were generated for visualization purposes using counts normalized with the variance stabilization transformation of DESeq2 and corrected for library preparation batch or sex-specific effects when applicable using limma (v.3.54.0)[103].

## Reverse transcription-qPCR

A First Strand cDNA Synthesis kit (OriGene), FastStart Universal SYBR Green Master (Rox) mix (Roche) and primers listed in Supplementary Table 7 were used for RT-qPCR. Amplification was performed on a CFX384 instrument (Bio-Rad), and target genes were quantified relative to *Hprt* and *Sdha* housekeeping genes using CFX Manager (v.3.1) software (Bio-Rad).

## scRNA-seq library preparation

Droplet-based scRNA-seq (Chromium, 10x Genomics) was applied to single-cell suspensions of BRAF$^{V600E}$/PTEN-deficient melanomas from 3 WT and 3 dKO mice; $6 × 10^3$ cells were loaded per well for each of the 6 samples, and partitioning and library preparation were performed according to the manufacturer's protocol for the 3′ v.3 chemistry.

## scRNA-seq analysis

Filtered gene–barcode matrices generated using cellranger (v.7.0.1) with the mm10-2020-A transcriptome (10x Genomics) were further analysed using Seurat (v.4.0)[104]. Low-quality cells (fewer than 500 unique molecular identifiers or 250 detected genes, or at least 20% mitochondrial transcripts) were removed, and data were normalized using SCTransform (v.2), regressing out cell cycle scores (determined using the CellCycleScoring function) and mitochondrial ratio. Datasets were integrated using reciprocal principal component analysis on the first 50 principal components with an alignment strength of ten. Within this combined dataset, clusters were determined using a variety of resolutions and annotated using top cluster-specific genes conserved between WT and dKO cells, as well as known lineage genes for expected cell types. At final resolution (0.6), selected to avoid overclustering while distinguishing rare cell types, five clusters with a low number of significant genes, also displaying a low nuclear transcriptome complexity, were considered to represent low-quality cells and were discarded. The least abundant cluster, characterized by genes of multiple lineages, probably contained doublets and was also discarded. dKO versus WT DEGs were identified within each cluster on the RNA slot using the FindMarkers function in Seurat. GSEA was performed using SCPA (v.1.5.1)[105]. Reclustering of related cell types was performed by subsetting the Seurat object to the relevant clusters, then performing normalization, integration and clustering as described above. Pseudotime trajectories for NC lineage clusters were calculated using Monocle 3 (v.1.3.1)[106], ordering cells in pseudotime under the assumption that they shared a common precursor and transformation initiated in the highest *Tyr*-expressing cluster. Unbiased cell identity mapping to existing scRNA-seq datasets was performed by subsetting Seurat objects to myeloid cell clusters, then determining the highest similarity in terms of gene expression to the reference cell types using singleR (v.1.10.0)[107]. Local changes in cell abundance were profiled using miloR (v.1.5.0)[108] with the following parameters for the entire dataset: 50 principal components, 40 nearest neighbours, sampling proportion 0.1 and sampling refinement algorithm. Thirty nearest neighbours and a sampling proportion of 0.2 were used for the lower number of cells in the lymphoid reclustered dataset. Cell-type prioritization to evaluate the contribution to changes in gene expression between WT

and dKO samples was performed using Augur (v.1.0.3)[109] using default parameters except for a minimum cell number of 50. Cell signalling through ligand–receptor interaction analysis was performed using CellChat (v.1.6.1)[110] using the comparison workflow on a merged object.

## Detailed cluster annotation

In addition to those described in the text, the following genes were used as markers during cluster annotation. Melanocytes were identified through the expression of genes associated with pigment production, such as *Mitf*, *Mlana* and *Dct*[45]. The NC arrested cluster expressed high levels of the cell cycle inhibitors p21(CIP1) (encoded by *Cdkn1a*) and p19(ARF) (encoded by *Cdkn2a*), the EGF-like ligands amphiregulin (*Areg*) and epiregulin (*Ereg*), and the histone variant H2A.J (*H2afj*). Amphiregulin expression is associated with BRAF$^{V600E}$-induced senescence in melanocytes[111] and H2A.J accumulates in senescent fibroblasts[112]. Another NC cluster expressed *Hapln1*, an ECM crosslinker the downregulation of which in aged fibroblasts promotes melanoma dissemination[113,114]. The NC *Aqp* cluster was characterized by aquaporin 1 (*Aqp1*) and *Tfap2b*, two factors that orchestrate NC cell migration[115,116]. The monocyte cluster was characterized by *Ccr2*, *Ms4a4c* and *Plac8* expression, markers of tissue monocytes shown to transcriptionally resemble peripheral blood monocytes[117]. Mac *Atp6v0d2* selectively expressed *Atp6v0d2* and *Gpnmb*, which are associated with lysosomal function and phagocytosis[118,119], and probably corresponds to melanophages[118]. We identified conventional type 2 dendritic cells (cDC2) through CD301b (*Mgl2*) and *Plet1* expression[120,121], their type 1 counterparts (cDC1) through CD103 (*Itgae*) and *Xcr1* (ref. [122]), and a mature, migratory subtype (mDC) expressing *Ccl22* and *Ccr7* (ref. [122]). Mast cells and basophils were distinguished by mutually exclusive expression of *Mcpt4* and *Mcpt8* (ref. [123]). After reclustering of lymphoid cells, the CD4 circulating cluster expressed genes consistent with a non-effector phenotype, including *Gramd3*, a marker of circulating T cells[124], *S1pr1*, a positive regulator of T cell emigration from peripheral organs[125], and TCF1 (*Tcf7*), expressed in naive and memory but not effector T cells[126]. The T helper 2 (Th2) cell population comprised *Cd4*-positive cells expressing *Il5* and *Il13*, Tγδ cells expressed γδ T cell receptor genes *Tcrg-C1* and *Trdc*, and innate lymphoid cells in the ILC cluster expressed *Gata3* and *Hlf* while being negative for *Cd3e* and *Ncr1* (ref. [127]). We identified a prototypical CAF cluster expressing the highest levels of *Pdgfra* and *Fap*[128] and CAF-specific lincRNA *Meg3* (refs. [129,130]). A second fibroblast cluster expressed *Wif1*, an emerging marker in papillary dermis[131–133]. Two clusters with high levels of the myofibroblast marker *Acta2* expressed *Lrrc15* and *Fbln1*, respectively, both genes associated with immunoregulatory myofibroblast populations in pancreatic cancer and in breast cancer[88,89].

## ST analysis

Tissue was prepared according to 10x Genomics Visium V1 Slide–3′ Spatial guidelines. Tumours were frozen in a bath of isopentane and liquid nitrogen, stored at −80 °C in a sealed container, then embedded in OCT. Tissue was sectioned at temperatures of −20 °C for the cryostat chamber and −10 °C for the specimen head at 10 μm thickness. An optimal permeabilization time of 18 min was determined using a Visium Spatial Tissue Optimization kit (10x Genomics). Sequencing data were mapped using spaceranger (v.2.0.0) using the mm10-2020-A transcriptome (10x Genomics) to generate spatial gene expression matrices, which were processed according to the Seurat spatial vignette (https://satijalab.org/seurat/articles/spatial_vignette.html). We integrated data normalized with SCTransform (v.2) across the two ST samples. For label transfer, we then integrated this dataset on the first 30 principal components with scRNA-seq data normalized using SCTransform (v.2). The probabilistic distribution of cell types identified by scRNA-seq in each ~50 μm spot of the spatial array was calculated using the TransferData function. To assess cell-type colocalization within the tissue[134], we determined significant Pearson's correlations between cell-type scores

across each array spot with the corr.test function of the psych (v.2.3.3) package (https://CRAN.R-project.org/package=psych), using the Holm method for multiple comparison $P$ value adjustment.

## CAF cultures

Primary mouse CAFs were sorted from tumours at 50 DPI based on PDGFRα (also known as CD140a) expression and cultured in DMEM supplemented with 10% FBS and penicillin–streptomycin at 37 °C and 5% $CO_2$. Cells were split at a ratio of 1:2–1:5 when fully confluent for no more than 15 total passages from the time of sorting. For cytokine gene or protein induction studies, cells were grown to confluency then deprived of serum in DMEM with 0.5% FBS for 24 h, followed by stimulation with 10% FBS for the indicated amount of time. Primary human melanoma CAFs were obtained from the NCI Patient-Derived Models Repository (PDMR[135], NCI-Frederick, Frederick National Laboratory for Cancer Research, Frederick, MD, https://pdmr.cancer.gov/) and the Aplin laboratory[136,137]. All human-derived cell cultures were stripped of any patient identifiers before they were provided to our laboratory, and their use is not subjected to Institutional Review Board approval. Ethics information relevant to their collection is provided in the noted references. Cells were maintained on plates coated with Cultrex Basement Membrane Extract, PathClear (Bio-Techne) in Advanced DMEM/F12 (Gibco) supplemented with 5% heat-inactivated FBS, 2 mM L-glutamine (Gibco), 10 ng ml$^{-1}$ recombinant human EGF (Gibco), 400 ng ml$^{-1}$ hydrocortisone (Sigma), 24 µg ml$^{-1}$ adenine (Sigma), 100 µg ml$^{-1}$ Primocin and 25 µg ml$^{-1}$ Plasmocin (Invivogen); 10 µM Y-27632 dihydrochloride (Tocris) was added to the medium during the initial expansion of cultures. For serum starvation before cytokine induction, hydrocortisone, adenine and EGF were omitted, and the serum concentration was reduced to 0.25%. After 24 h, starvation medium was replaced with complete human CAF medium without Y-27632 dihydrochloride. Supernatant was collected, centrifuged to remove cells and debris, and flash-frozen. Cells were counted using a Guava Muse Cell Analyzer Count & Viability kit (Luminex).

## Protein quantitation

Western blotting was performed as previously described[62] using the antibodies listed in Supplementary Table 8. For cytokine quantitation in mouse samples, total protein was extracted using RIPA buffer and normalized to the lowest sample concentration. Analytes of interest were quantified using a Mouse Cytokine Array/Chemokine Array 31-Plex by Eve Technologies. For cytokine quantitation in human CAFs, supernatant was diluted 1:5, 1:10 and 1:20 with assay diluent and subjected to ELISA in duplicate against human CCL2, IL-6 (BioLegend) and CXCL1 (R&D Systems) according to the manufacturers' instructions. Data points within range of the standard curve were normalized by cell number and averaged for each sample.

## Monocyte Transwell migration assay

Monocyte migration was performed as previously described[138] but with modifications. In brief, $7.5 \times 10^4$ WT or dKO CAFs were seeded per well of a 24-well cell culture insert companion plate (Corning), in triplicate, allowed to grow for 24 h reaching confluency, then starved for 24 h as described above. Monocytes were isolated from femur and tibia bone marrow of tumour-naive WT 129S mice using a MojoSort Mouse Monocyte Isolation kit (BioLegend) and labelled with 3 µM CMFDA green dye in serum-free RPMI 1640 medium for 30 min at 37 °C. After washing with R10 medium (RPMI 1640 supplemented with 10% heat-inactivated FBS, 20 mM HEPES, 0.5 mM sodium pyruvate, 1% penicillin–streptomycin, 1× MEM amino acids without L-glutamine, 1× MEM non-essential amino acids, Gibco), unbound dye was allowed to diffuse out by another incubation for 30 min at 37 °C in R10. Next, $1 \times 10^5$ monocytes in 300 µl R10 were added to a FluoroBlok insert with 3 µm pore size (Corning), which functioned as the upper chamber. Concomitantly, starvation medium on CAFs in the companion plate, which functioned as a lower

chamber, was replaced with 800 µl R10. One well each of R10 and R10 supplemented with 10 ng ml$^{-1}$ recombinant mouse CCL2 (BioLegend) were used as negative and positive controls, respectively. Plates were imaged for green widefield fluorescence at the $Z$ position corresponding to the CAF layer, every 6 h for a 72 h time course, in a Cytation 7 automated microscope (Agilent). Image stitching, background subtraction, segmentation and cell count were performed using the onboard Gen5 (v.3.12) software (Agilent). To account for differences in monocyte reactivity among donor mice, cell numbers were normalized to the number of monocytes migrated at 24 h in the CCL2-positive control.

## TCGA data analysis

TCGA melanoma cohort (SKCM)[139] RNA-seq raw counts and sample annotations were downloaded using the TCGAbiolinks (v.2.25.3) package[140]. Gene counts were normalized using the TMM method from the edgeR (v.3.40.1) package[141] to account for biases arising from library size and gene length. Primary and metastatic lesions were separately analysed, and comparisons for TME composition and gene expression were performed between the first and third tercile of *MACROH2A1* and *MACROH2A2* expression. Deconvolution of immune populations[142] and tumour/stroma/immune microenvironmental composition[143] was performed using the IOBR (v.0.99.9) package[144]. Differential gene expression analysis was performed using DESeq2 (ref. 99) with raw counts as input.

## Human CAF scRNA-seq reanalysis

scRNA count matrix of 855,271 high-quality cells and associated metadata annotations from a previous study[60] (GSE210347) were reanalysed. Counts were scaled using the Seurat LogNormalize function with a scale factor of 10,000. The top 2,000 most highly variable genes were identified using the vst selection method of FindVariableFeatures. The RunFastMNN function from the SeuratWrappers (v.0.3.0) package was utilized to perform batch correction using the 'SampleID' metadata column. The top 30 principal components were used in the downstream analysis. To investigate the correlation between *MACROH2A1*–*MACROH2A2* and *IL6*–*CXCL1*–*CCL2* expression, we generated pseudo-bulk counts from fibroblasts. In brief, the filtered Seurat object containing annotated fibroblasts was converted into a SingleCellExperiment (v.1.12) object using the raw counts per cell. The function aggregate.Matrix was used to sum and collapse the raw counts of each cell by their 'SampleID' annotation. Samples with <100 cells and non-tumour tissue were filtered out. Collapsed raw counts were normalized using the vst function from DESeq2 and the Pearson's correlation between genes was calculated using the chart.Correlation function from the PerformanceAnalytics (v.2.0.4) package.

## ATAC–seq library preparation

$5 \times 10^4$ sorted CAFs per sample were processed for ATAC–seq as previously described[145]. The optimal number of library amplification cycles was determined[146]. Libraries were subjected to double-sided size selection using SPRIselect beads (Beckman Coulter) at ratios of 0.55 and 1.2× before sequencing on a NextSeq 500 (Illumina) in 75 bp paired-end mode.

## ATAC–seq analysis

Read pairs were merged and adapters removed using NGmerge (v.0.3)[147], followed by alignment to the mm10 assembly with bowtie2 (v.2.4.1)[148]. Low-quality (MAPQ < 30), mitochondrial genome and duplicated reads were removed using samtools (v.1.9)[149], and genome coverage calculation for visualization purposes on the UCSC genome browser was performed using deepTools (v.3.5.1)[150] excluding black-listed regions[151]. WT and dKO replicates were concatenated using samtools merge to generate a master bam file. MACS2 (v.2.1.0)[152] was used to identify significant peaks in the master bam file using the parameters –nomodel –nolambada –keepdup all –slocal 10,000. Quantification of

reads in significant peaks for all samples was preformed using bedtools (v.2.29.2)[153] multicov. Differential peak analysis was preformed using DEseq2 (1.30.1)[99] (adjusted *P* value of <0.05 using the Benjamini and Hochberg procedure). For plotting static peaks, a set of 3,000 peaks with *P* adjusted of >0.05 and absolute $\log_2$ fold change of ≤0.2 were randomly selected.

### ChIP–seq
Approximately $8 \times 10^6$ cultured CAFs or iDFs after 30 min of serum stimulation were single-crosslinked and processed for ChIP[154] for H3K27ac (antibody 13-0045, Epicypher, lot number 20120001-28, 4 μg per reaction). Sequencing was performed on a NextSeq 500 in 75 bp single-end mode.

### ChIP–seq analysis
Adapters were trimmed using Trimmomatic (v.0.36)[155], and alignment, read filtering and genome coverage calculation were performed as for ATAC–seq. The bam files of WT and dKO samples were concatenated into a master bam file, which was used to call significant peaks with matching input controls using MACS2 (v.2.1.0)[152]. Cutoff values for *q*-value significance were determined post-hoc, testing several *q*-values on the basis of the signal-to-background ratio. Quantification of reads in significant peaks for all samples was performed using bedtools (v.2.29.2)[153] multicov. TEs and SEs were called on the basis of H3K27ac enrichment using the ROSE algorithm (rank ordering of super-enhancers) (v.0.11)[156,157] with a stitching distance of 12.5 kb and a TSS exclusion zone size of 2.5 kb. The ROSE algorithm was also used to extract H3K27ac levels at TEs and SEs for WT and dKO samples individually. The average H3K27ac signal value across all elements was calculated for each sample and further used for normalization between samples. The $\log_2$ fold change ratio of the normalized signal ($-0.75 < \log_2$ fold change $< 0.75$) was used to call differential TEs and SEs. All other enhancers were considered static.

### CUT&RUN
$2 \times 10^5$ short-term cultured CAFs before or after 30 min of serum stimulation were processed for CUT&RUN[158] for macroH2A1 (antibody ab37264, Abcam, lot number GR278020-1, 1 μg per reaction). Cells were permeabilized with 0.0085% digitonin and incubated with antibody. DNA was cleaved with the CUTANA pAG-MNase (Epicypher). Released DNA was processed using a NEBNext Ultra II Library Preparation kit for Illumina (NEB), including a 14-cycle amplification step. Libraries were sequenced and reads were processed as for ATAC–seq.

### CUT&RUN analysis
MacroH2A1 enrichment determined by CUT&RUN was benchmarked by correlation analysis with published macroH2A1 ChIP–seq data in dermal fibroblasts[62]. Enrichment at the level of read pileups had a correlation coefficient of 0.79 (Extended Data Fig. 6a). MacroH2A1 chromatin domains were called separately for stimulated and unstimulated samples using SEACR (v.1.3)[159] without a control in stringent mode with a threshold of 0.05, and significant peaks were merged using bedtools (v.2.29.2)[153] if within 25 kb distance. Merged peak and alignment files for unstimulated and stimulated cells were then concatenated to generate master files for the ROSE algorithm[156,157]. A stitching distance of 12.5 kb, no TSS exclusion and normalization to element size were used in ROSE to rank macroH2A1 domains. The equivalent of SEs called by ROSE based on the macroH2A1 signal were termed super-MCDs. The equivalent of TEs was then divided into groups of standard and low macroH2A1 signals using a cutoff above 3,780 (signal density × length units) determined post-hoc on the basis of the macroH2A1 enrichment-to-background ratio. Most domains with low macroH2A1 signal were also below 1.5 kb in length, represented less than 0.5% of the genome, showed higher enrichment of IgG over the macroH2A signal, and were therefore excluded from further analysis.

For comparing macroH2A1 enrichment levels, static genes were selected as follows: for each differentially expressed gene, all genes within 25% of its averaged transcripts per million (TPM) level in WT samples were identified, and 10 (inflammatory up genes comparison) or 3 (all DEG comparison) genes that were neither differentially expressed nor less than 1 kb in length were randomly selected, added to a running list, and duplicates were removed.

### pcMicro-C
Micro-C was performed using a Dovetail Micro-C kit (Cantata Bio) according to the manufacturer's protocol. Chromatin from $1 \times 10^6$ WT and dKO cultured CAFs, crosslinked after serum stimulation, was digested with 3 μl MNase to obtain a mononucleosomal fraction within specifications. Each chromatin preparation was subjected to proximity ligation and subsequent library preparation in duplicate. Next, 400 ng of each library was pooled and subjected to promoter capture using a Dovetail Target Enrichment kit and a Mouse Pan Promoter Panel (Cantata Bio). The manufacturer's protocol was adjusted by halving the streptavidin bead elution volume to enable the use of the entire post-capture material for the amplification reaction and performing only seven PCR cycles. The resulting 4-plex library was sequenced on a NextSeq 500 high output lane in a 75 bp paired-end configuration.

### Chromatin looping analysis
Interaction calling and significance thresholding was performed based on the workflow developed by Dovetail Genomics (https://micro-c.readthedocs.io/en/latest/) and the CHiCAGO tool recommendations[160]. Sequenced reads were aligned using BWA (v.0.7.17-r1188)[161] and filtered using PairTools (v.1.0.2)[162] and SAMTools (v.1.9)[149] to identify non-duplicated read pairs. The technical replicates were merged, and significant interactions (chromatin loops) were called using CHiCAGO (v.1.2.0)[163] with default parameters at 10 kb resolution[160]. Significant Interactions were defined as those with a CHiCAGO score of ≥5 per condition. For visualization on the UCSC genome browser, output loop files were converted from the WashU EpiGenome Browser interactBED format to the bigInteract format using BEDTools (v.2.30.0)[153,164] and UCSC-Utils (v.2.9)[165,166]. Bait map plots depicting all called interactions per bait were generated using the plotBaits function native to CHiCAGO. Histograms showing the enrichment of functional elements (ATAC–seq peaks and H3K27ac peaks) at the distal end of loops were generated as a default output of the CHiCAGO pipeline. Interactions were defined as shared if both ends overlapped between conditions, and unique otherwise. Promoters and enhancers at proximal and distal ends of loops, respectively, were divided into classes on the basis of the type of macroH2A1 domain they overlapped with—super, standard and absent—and the nature of the loops underlying them—shared, WT-specific or dKO-specific. For the enhancer-specific association, enhancers were further divided on the basis of the direction of their change in H3K27ac levels by macro level to generate a count matrix of changes in enhancer activity versus changes in interactions, stratified by macroH2A1 level. We performed a Chi-square test of independence to determine associations between enhancer and looping deregulation in the absence of macroH2A. The mouse promoter panel uses multiple baits to capture alternative promoters of the same gene; therefore, we summed loop counts for all baits associated with the same gene name for overlap with gene expression changes. We compared the loop counts per gene in dKO versus WT samples and further stratified genes on the basis of their differential expression status and macroH2A1 occupancy. We performed a Chi-square test of independence to determine associations between gene expression and looping deregulation in the absence of macroH2A.

### Statistics and reproducibility
Information on the number of biologically independent samples analysed and the number of times experiments were performed is included

in the figure legends. No assumption of data normality was made, and non-parametric statistical tests were performed except when $n$ = 3 replicates (comparison of cell-type frequency in scRNA-seq data, CAF RT-qPCR) for which data distribution was assumed to be normal but this was not formally tested. All statistical tests performed were two-sided except when noted. No statistical method was used to predetermine sample sizes, but were similar to those reported in previous publications[22,97,167]. No data were excluded from the analyses, and biological samples were excluded from the study only if sample preparation or data acquisition failed. The experiments were not randomized, and the investigators were not blinded to allocation during experiments and outcome assessment.

### Reporting summary
Further information on research design is available in the Nature Portfolio Reporting Summary linked to this article.

### Data availability
The transcriptomics and epigenomic datasets, including raw and processed sequencing data generated and analysed during the current study, are available in the Gene Expression Omnibus (GEO) repository under accession number GSE200751 (https://www.ncbi.nlm.nih.gov/geo/query/acc.cgi?acc=GSE200751) and in this article's table files. TCGA melanoma data are publicly available through the NCI Genomic Data Commons (GDC) data portal under project ID TCGA SKCM (https://portal.gdc.cancer.gov/projects/TCGA-SKCM). The human pan-cancer scRNA-seq dataset mined in this study is available in the GEO repository under accession number GSE210347 (https://www.ncbi.nlm.nih.gov/geo/query/acc.cgi?acc=GSE210347). The mouse M25 (GRCm38.p6) genome assembly and gene set used for transcriptomics and epigenomic analyses are available at Gencode (https://www.gencodegenes.org/mouse/release_M25.html). All other data supporting the findings of this study is available from the corresponding authors upon request. Source data are provided with this paper.

### Code availability
All packages used for data analysis are publicly available. No custom code was generated for this study. All scripts used for bulk RNA-seq, scRNA-seq, ATAC–seq, ChIP–seq and CUT&RUN data analyses in this study are available from the corresponding authors upon request.

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

## Acknowledgements

The authors thank the laboratories of J.-C. Marine, J. Pehrson, J. Celebi and M. Bosenberg for reagents and advice; R. Singh for pathology support; I. Russell, L. Munding and M. Elkhawand (Cantata Bio) for technical support of pcMicro-C; M. Spivakov for assistance with Micro-C analysis; the Bernstein and Merad laboratories for discussions and feedback; and J. Gregory for illustrations. This study was supported by an American Skin Association Ping Y. Tai Melanoma Research Grant to D.F.; a Mount Sinai Skin Biology and Diseases Resource-based Center Transition to Independence Minigrant to D.F., funded through NIAMS/NIH SBDRC P30 AR079200; a Melanoma Research Alliance Pilot Award (https://doi.org/10.48050/pc.gr.91577), Department of Defense (W81XWH2010803) and NIH/NCI R01 CA154683 and CA218024 to E.B. A.E.A. is supported by NIH/NCI R01 CA182635. A.O.K. is supported by NIH/NCI R01 AI153363-01A1. This work was supported in part through the Oncological Sciences Sequencing Core supported by the Tisch Cancer Institute of the ISMMS Cancer Center Support Grant P30 (CA196521), and Scientific Computing is supported by the Office of Research Infrastructure of the NIH under award number S10OD026880 to ISMMS. The funders had no role in the study design, data collection and analysis, decision to publish or preparation of the manuscript. The authors acknowledge the Center for Advanced Genomics Technology, the Dean's Flow Cytometry Core, the Biorepository and Pathology Core, The Bioinformatics for Next Generation Sequencing Core, the Center for Comparative Medicine and Surgery, and the Microscopy Core at ISMMS.

## Author contributions

Conceptualization: D.F. and E.B. Investigation: D.F., D.H., N.T., É.H., S.S. and F.G.G. Formal analysis: D.F., D.H., S.C., A.A., M.S.G. and N.S.V. Writing original draft: D.F. and E.B. Writing, review and editing: D.F., D.H., S.C., A.A., D.D., M.S.G., H.S., R.S., A.O.K., M.M. and E.B. Visualization: D.F., S.C., A.A. and D.H. Resources, expertise and methods: É.H., D.D., M.S.G., K.G.B., C.C., A.E.A., H.S., R.S., A.O.K. and M.M. Data curation: D.F. Supervision: E.B. Funding acquisition: D.F. and E.B.

## Competing interests

The authors declare no competing interests.

## Additional information

**Extended data** is available for this paper at https://doi.org/10.1038/s41556-023-01208-7.

**Correspondence and requests for materials** should be addressed to Dan Filipescu or Emily Bernstein.

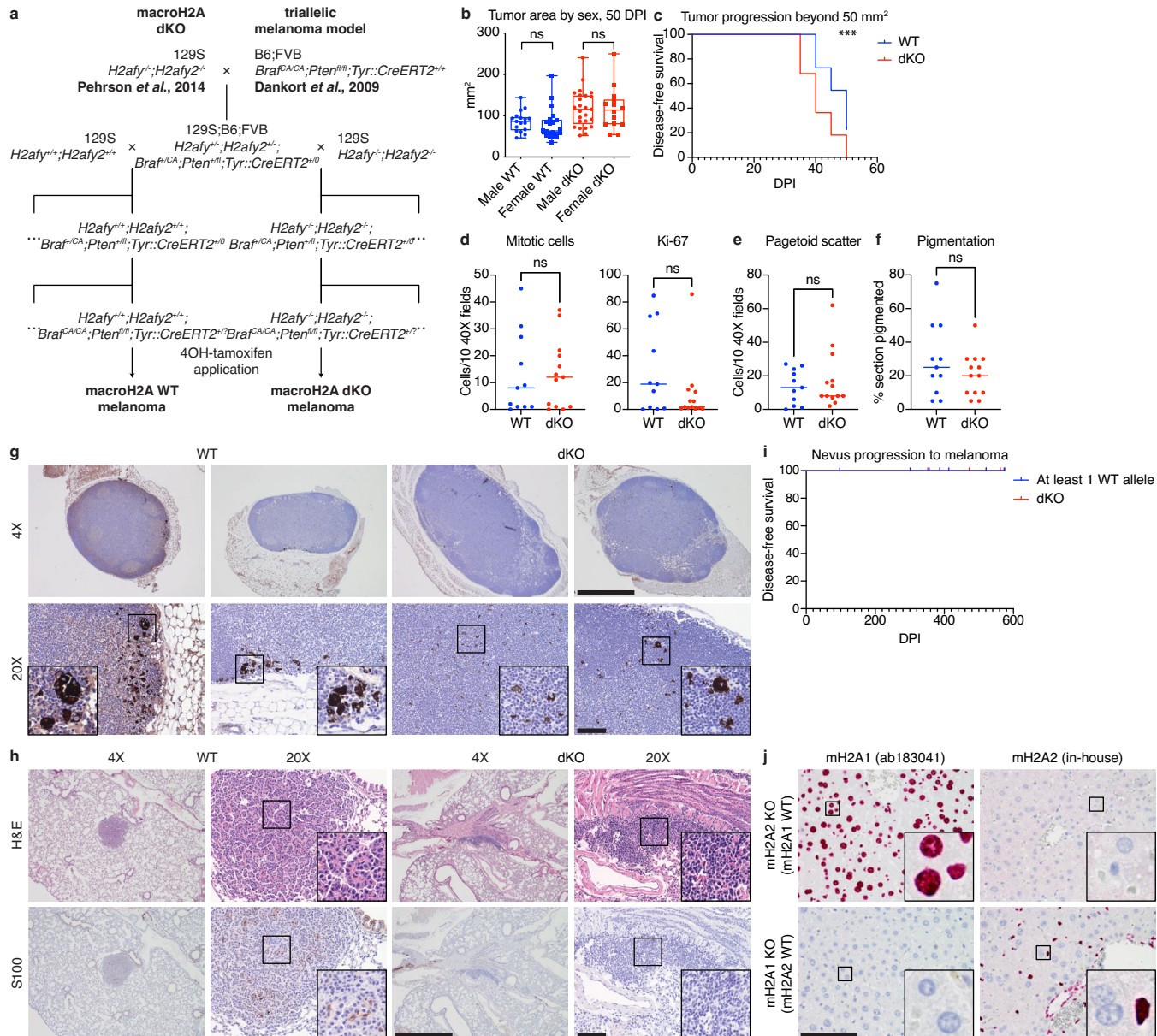

**Extended Data Fig. 1 | Characterization of macroH2A WT and dKO murine melanomas.** a) Breeding strategy to obtain WT and dKO mice used for melanoma induction. b) Tumor area in males and females, $n_{WT-M}$ = 17, $n_{WT-F}$ = 21, $n_{dKO-M}$ = 25, $n_{dKO-F}$ = 14. No significant inter-sex differences were observed within genotypes (Kruskal-Wallis with Dunn's multiple comparisons test). Box plot whiskers represent min-max range, box plot limits – 25th to 75th percentiles, center line – median. c) Kaplan-Meier analysis of disease-free survival. Events represent melanoma growth beyond an area of 50 mm², $n_{WT}$ = 22, $n_{dKO}$ = 22. P-value = 0.0009, log-rank (Mantel-Cox) test. d) Immunohistochemical scoring of H3S10ph and Ki-67 at 50 DPI in ten random fields per tumor. e) Pagetoid spread of melanocytic cells into epidermal structures scored in ten epidermis-containing fields per tumor on H&E-stained sections. f) Relative pigmented area of the tumor, estimated on a 2X magnification ensemble view of H&E-stained sections. d-f, $n_{WT}$ = 11, $n_{dKO}$ = 13, center line represents median,

significance determined using a Mann-Whitney two-tailed test. Exact P-values are provided as numerical source data. g) Tumor-draining axillary lymph nodes stained for S100 (reddish-brown NovaRed substrate color). h) Lung sections of tumor-bearing mice at 50 DPI. g-h, 4X (scale bar = 1 mm) and 20X (scale bar = 100 μm) magnification shown; inserts shown at 2.5X additional magnification. Staining and analysis were performed in 2 animals per genotype with similar results. i) Kaplan-Meier analysis of disease-free survival following nevus induction, $n_{at\ least\ 1\ WT\ allele}$ = 7, $n_{dKO}$ = 6. Progression to melanoma, defined as radial and/or vertical lesion growth, was not observed during the lifespan of the mice. j) Validation of antibody specificity for macroH2A variants in immunohistochemistry, using liver of either macroH2A1 or macroH2A2 single KO mice. Scale bar = 100 μm, inserts shown at 5X additional magnification. Experiment was performed twice with similar results.

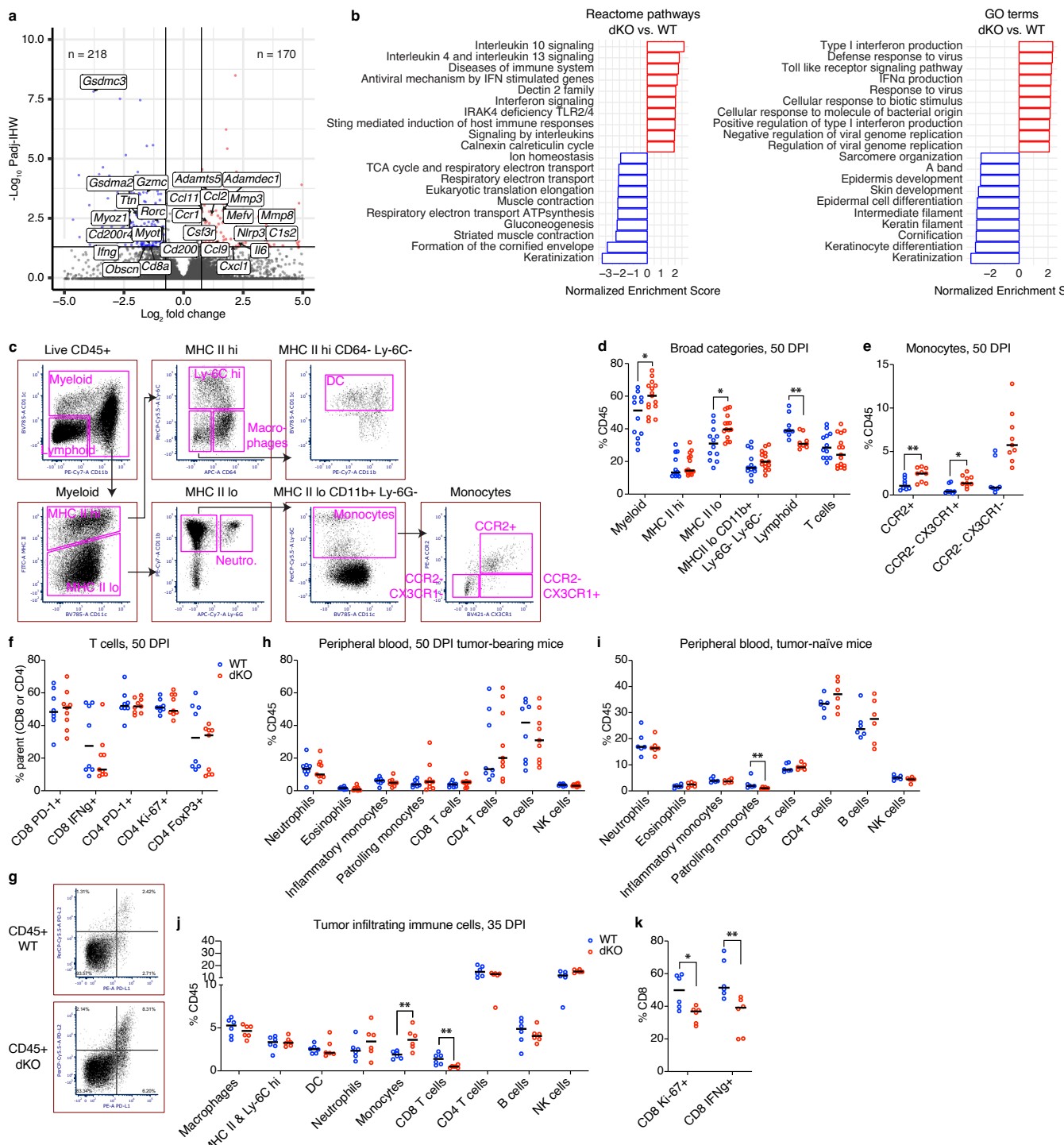

**Extended Data Fig. 2 | Murine macroH2A-deficient melanomas harbor a dysfunctional immune microenvironment. a)** Volcano plot of differential gene expression in dKO vs. WT murine melanomas analyzed by RNA-seq. Significant DEGs are colored: red = up in dKO and blue = down in dKO. X and Y axes are zoomed in to depict genes in Fig. 2b. Independent hypothesis weighted Wald test P-values adjusted for multiple comparisons, computed by DESeq2. **b)** GSEA analysis of Reactome pathways and GO terms in dKO vs. WT murine melanomas. Top 10 significant pathways shown. **c)** Gating strategy used to delineate tumor-infiltrating myeloid cell subtypes by flow cytometry, shown in a WT tumor. **d)** Broad immune cell categories not shown in Fig. 2 at 50 DPI, $n_{WT}$ = 12, $n_{dKO}$ = 15 animals except for lymphoid population (sum of T, B and NK cells) where $n_{WT}$ = 8 and $n_{dKO}$ = 11 animals. **e)** Relative abundance of monocyte subpopulations identified by CCR2 and CX3CR1 expression. $n_{WT}$ = 8

and $n_{dKO}$ = 9 animals. **f)** Analysis of additional markers of T cell functionality. $n_{WT}$ = 8 and $n_{dKO}$ = 9 animals. **g)** PD-L1 and PD-L2 staining of CD45+ cells infiltrating WT and dKO melanomas. **h)** Quantification of indicated populations of non-overlapping peripheral blood immune cell populations at 50 DPI, determined by flow cytometry, $n_{WT}$ = 8, $n_{dKO}$ = 9 animals. **i)** As in (h) in tumor-naïve mice, $n_{WT}$ = 6, $n_{dKO}$ = 6 animals. **j)** Quantification of indicated non-overlapping tumor-infiltrating immune cell populations at 35 DPI. **k)** Proliferative status and anti-tumor activity of CD8 + T cells in (j), assessed as percentage of Ki-67 positivity and interferon-gamma production, respectively. j-k, $n_{WT}$ = 6, $n_{dKO}$ = 6 animals. d-f and h-k, Mann-Whitney two-tailed test P-values: * < 0.05, ** < 0.01, center line represents median. Exact P-values are provided as numerical source data. Non-significant differences are not labeled.

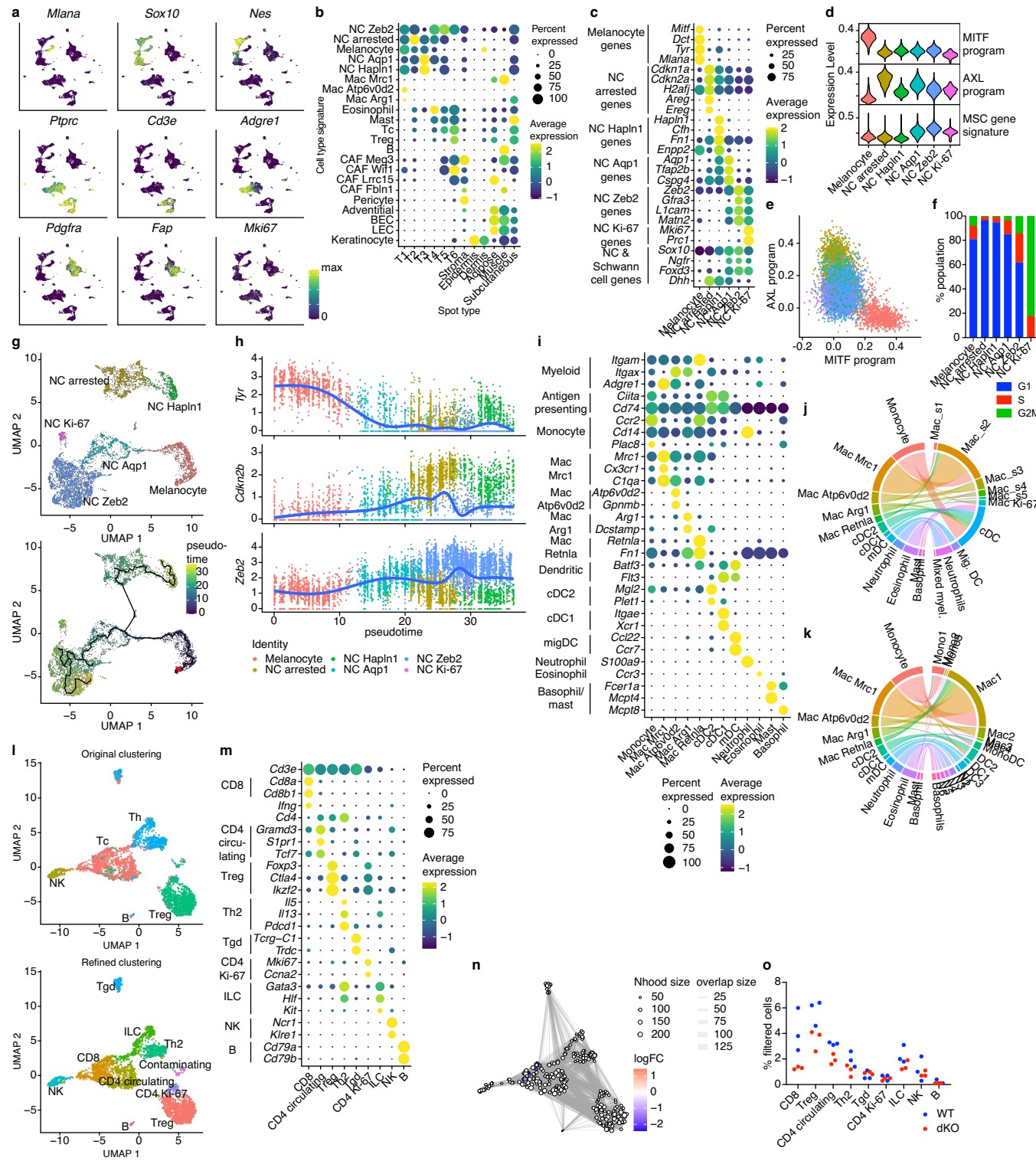

**Extended Data Fig. 3 | See next page for caption.**

**Extended Data Fig. 3 | Details of scRNA-seq analysis to annotate components of the melanoma TME.** a) Expression of marker genes of major cell lineages expected in the TME, shown on UMAP plots. b) Distribution of cell type signatures derived from murine melanoma scRNA-seq in spot types of the murine melanoma ST dataset. c) Selected cluster-specific markers used to annotate neural crest (NC) cell types/states. d) Violin plots of indicated gene expression signatures measured across NC clusters. e) Scatter plot of antagonistic MITF and AXL-driven gene expression signatures identified in human melanoma, across NC clusters in (d). f) Distribution of predicted cell cycle stages across NC clusters in (d). g) Pseudotime analysis of reclustered NC cells. Top, original neural crest clusters after re-integration, dimensionality reduction and UMAP embedding. Bottom, cell trajectories in pseudotime anchored in the melanocyte cluster (red dot). h) Representation of expression profiles of key genes across pseudotime in NC clusters. i) As in (c), for myeloid cells. j) Circos plots showing correspondence between myeloid cell clusters identified and myeloid cell identities in a published scRNA-seq datasets of murine subcutaneous sarcoma[52]. k) As in (j) in murine lung adenocarcinoma[53]. l) UMAP plot of lymphoid cells after re-integration, dimensionality reduction and UMAP embedding, labeled according to their original clusters (top) and annotations generated by reclustering (bottom). m) Selected markers used to annotate lymphoid cell types/states following reclustering in (l). n) UMAP representation of differential abundance analysis of cells in (l) performed with *Milo*. Cells are grouped into overlapping neighborhoods based on their K-nearest neighbor graph position, depicted as circles proportional in size to the number of cells contained, colored by log fold change of abundance between genotypes. Graph edge thickness is proportional to the number of cells shared between adjacent neighborhoods. o) Relative abundance across lymphoid clusters as annotated in (l) and (m), shown as percentage of total viable cells in each tumor.

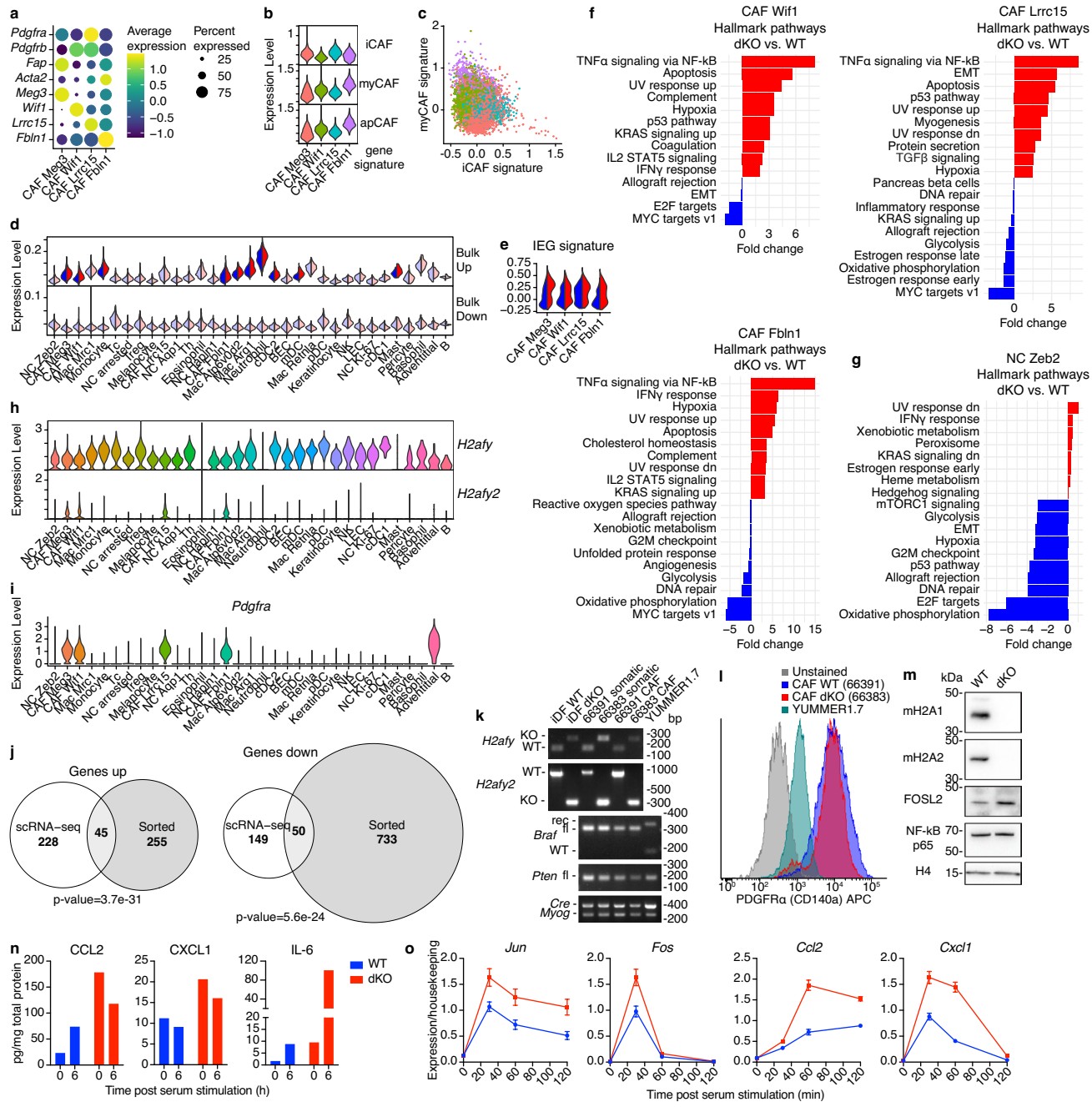

**Extended Data Fig. 4 | Characterization of CAF clusters in the melanoma TME and their *in vitro* counterparts.** a) Dot plot of markers used to annotate CAFs, and CAF cluster specific genes. b) Violin plots of murine pancreatic cancer CAF population signatures measured across murine melanoma CAF clusters. c) Scatter plot of inflammatory and myocyte-like CAF gene expression signatures identified in pancreatic cancers across murine melanoma CAF clusters. d) Gene signatures derived from differentially expressed genes in the bulk RNA-seq dataset, calculated across clusters and genotypes. Significance defined as Wilcoxon rank sum test adjusted P-value < 0.05, cluster-average log2 fold change > 0.02 or < -0.02. Exact P-values are provided in Table 3. e) Violin plot of an immediate early gene (IEG) signature in CAFs. Significant dKO vs. WT differences are present within each cluster (Wilcoxon rank sum test adjusted P-value < 0.05, cluster-average log2 fold change > 0.1). Exact P-values are provided in Table 3. f) Significant Hallmark pathways in GSEA analysis of dKO vs. WT CAF clusters. g) As in f, for the NC Zeb2 cluster. h) Violin plots of macroH2A gene expression in WT melanoma. i) Violin plot of *Pdgfra* expression in the melanoma TME.

j) Intersections of DEGs across single-cell and bulk RNA-seq modalities in CAFs. DEGs in the scRNA-seq dataset were determined by grouping all CAF clusters as one, prior to contrasting by genotype. P-values of Fisher's exact test shown. k) Genotyping of cultured CAFs and immortalized dermal fibroblasts (iDFs) compared to somatic DNA. YUMMER1.7 cells[168] are used as a reference for *Braf* allele recombination. Experiment was performed twice with similar results. l) Flow cytometric detection of PDGFRα on cultured CAFs. YUMMER1.7 is used as a negative control. m) Western blot showing deletion of macroH2A and accumulation of FOSL2 in dKO cultured CAFs. Histone H4 and NF-κB p65 are used as loading controls. Experiment was performed twice with similar results. n) Protein levels of indicated cytokines in serum-starved and stimulated cultured CAFs, measured by multiplexed bead-capture assay. o) Expression normalized to housekeeping controls of indicated immediate-early and cytokine genes determined by RT-qPCR in iDFs from WT and dKO mice at indicated time after serum stimulation. Mean of 3 PCR replicates shown, error bars represent SEM.

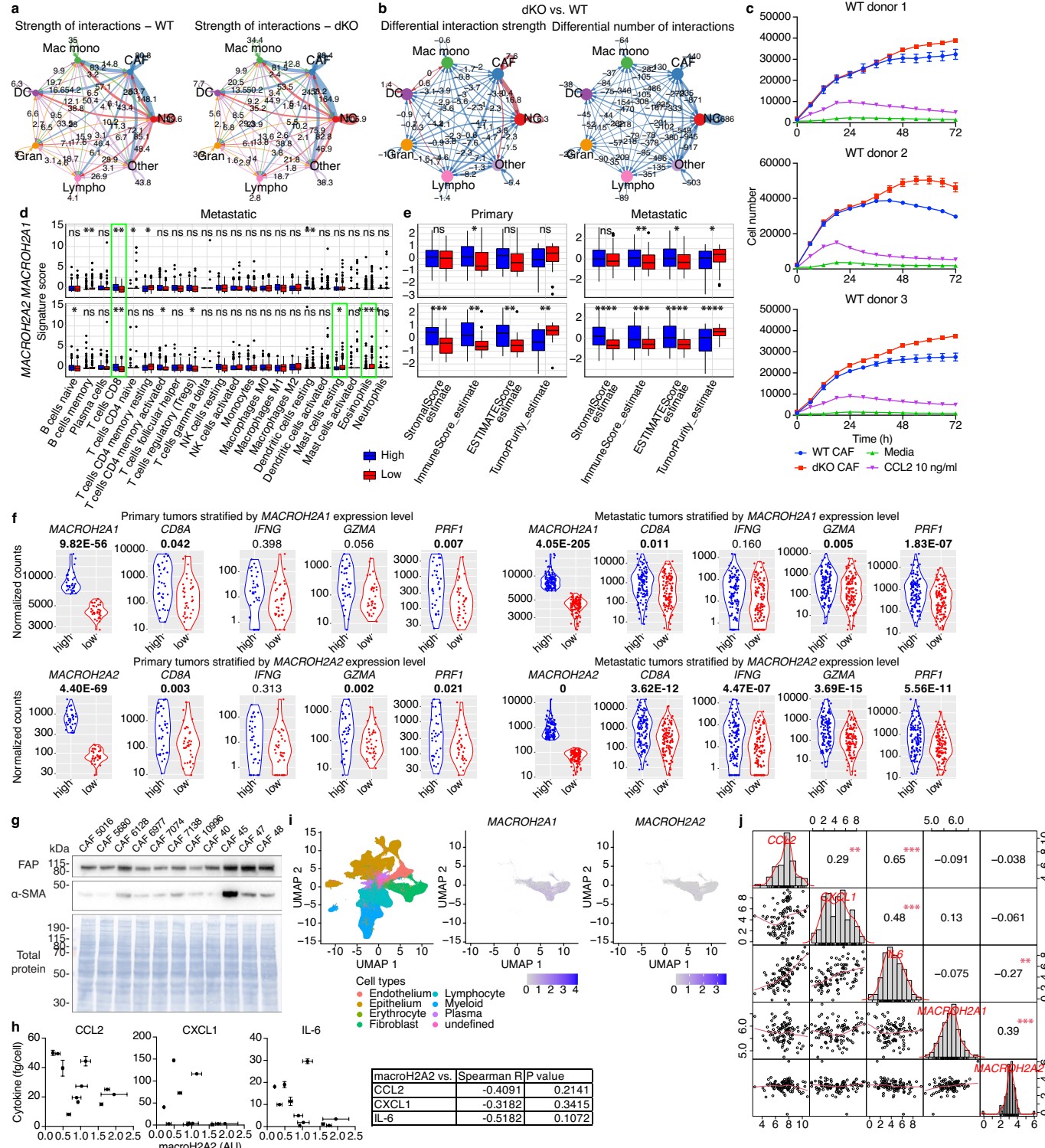

Extended Data Fig. 5 | See next page for caption.

**Extended Data Fig. 5 | MacroH2A regulates pro-inflammatory signals in mouse and human melanoma CAFs.** a) Ligand-receptor analysis of cell signaling performed with *CellChat*[110] summarizing interaction strength between broad cell types. Arrows depicting signaling direction are colored according to emitting cell type. Arrow weight is proportional to interaction strength. b) Differential interaction strength and number between WT and dKO cell type pairs. Arrows are colored according to direction of change (WT – blue, dKO – red) and weight represents amplitude of change. c) Transwell assay measuring the migration of CMFDA-labeled WT bone marrow-derived monocytes towards unlabelled WT or dKO CAFs. Average cell counts of 3 technical replicates and error bars representing SEM are shown for WT and dKO CAF conditions. No replicates are performed for negative and positive controls. Individual experiments using different monocyte donors shown. d) Comparison of deconvoluted immune cell type scores[142] between TCGA metastatic melanoma tumors with *MACROH2A1/2* high and low expression levels. n = 123 biologically independent samples per category. e) As in (d) for estimated immune, stromal and tumor purity scores[143]. $n_{primary} = 35$, $n_{metastatic} = 123$ biologically independent samples per category. d-e, Wilcoxon rank-sum test P-values adjusted for multiple comparisons shown: * < 0.05, ** < 0.01, *** < 0.001, **** < 0.0001. Exact P-values are provided as numerical source data. Box plot center line represents the median, box plot limits – 25th to 75th percentiles, whiskers extend from the box limit to the most extreme

value no further than 1.5 * inter-quartile range from the box limit, any data beyond whiskers is plotted as individual points. f) Expression of genes associated with anti-tumor cytotoxic activity in human primary and metastatic melanoma samples from the TCGA cohort, segregated by *MACROH2A1* or *MACROH2A2* gene expression levels. High and low terciles are compared. Independent hypothesis weighted Wald test P-values adjusted for multiple comparisons, computed by DESeq2, shown. g) Detection of CAF markers by western blot in a panel of 11 human melanoma primary CAF cultures. Membrane stain for total protein shown for loading control. h) Correlation between macroH2A2 protein levels and indicated cytokine secretion in CAF cultures from (g). n = 11 biologically independent samples. Error bars correspond to SEM of 3 technical replicates of blot-based quantification for macroH2A2 and up to 3 dilutions each in 2 technical replicates for cytokine ELISA. Spearman correlation statistics, calculated on the average values without considering individual technical replicates, are shown. i) UMAP plots of a human pan-cancer scRNA-seq dataset[60], together with macroH2A gene expression in fibroblasts. j) Correlation analysis between pseudobulk expression levels of selected cytokines and macroH2A genes in CAFs from (i). Pearson correlation coefficients and significance shown above the diagonal, pairwise scatter plots shown below. Histograms of individual gene expression comprise the diagonal. P-values shown: ** < 0.01, *** < 0.001. Exact P-values are provided as numerical source data.

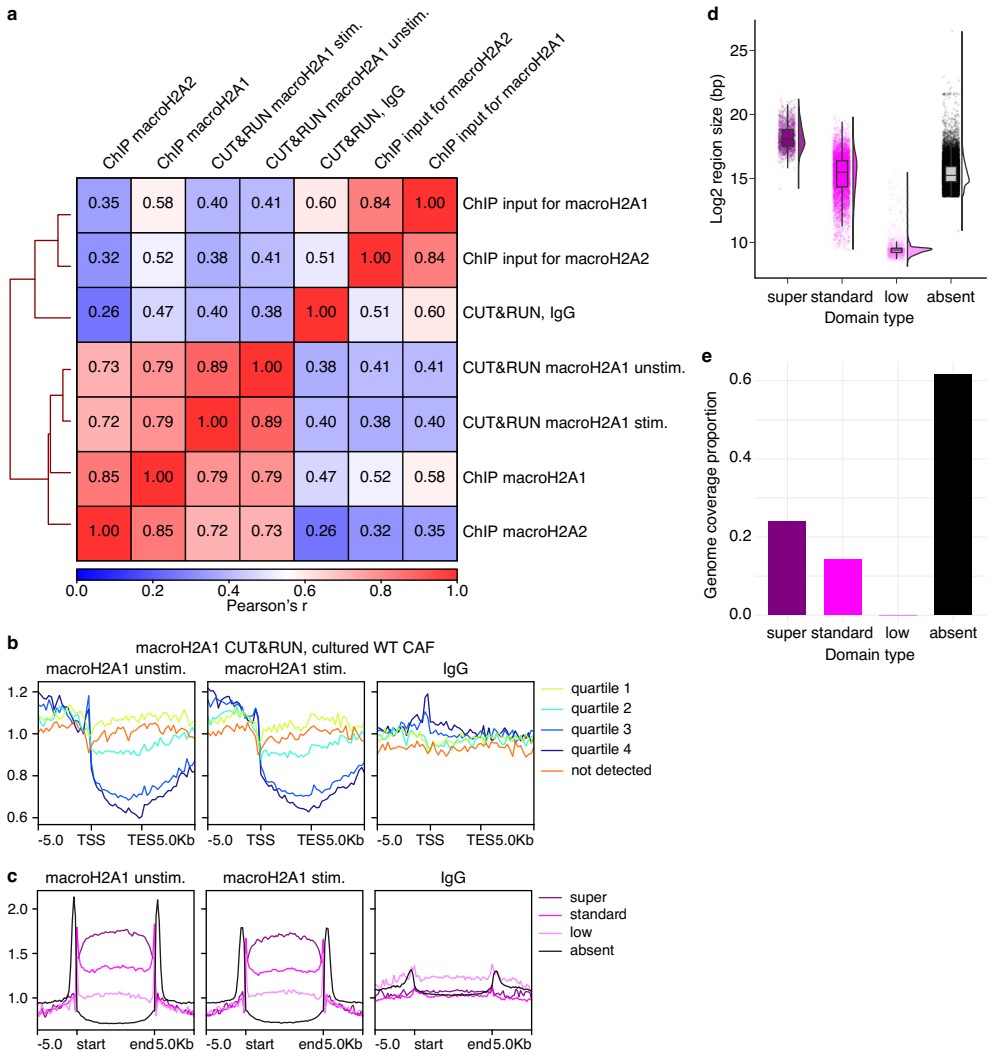

**Extended Data Fig. 6 | Genome-wide macroH2A2 occupancy in cultured WT CAFs.** a) Correlation of genome-wide enrichment of macroH2A between ChIP-seq and CUT&RUN methodologies. ChIP-seq and associated inputs were previously generated in dermal fibroblasts[62], CUT&RUN was performed in serum-starved unstimulated and serum-stimulated WT CAFs, with associated IgG control. Pearson's correlation coefficients shown, conditions are clustered based on Euclidean distance. b) Metagene profile of macroH2A CUT&RUN signal in cultured WT CAFs before and after serum stimulation across genes binned into quartiles according to their expression levels. Genes not detected are either

not expressed or not mappable. n = 3000 randomly selected genes within each group. c) Metagene profile of macroH2A CUT&RUN signal in cultured WT CAFs before and after serum stimulation at different classes of MCDs and genomic regions where MCDs are absent. d) Size distributions of regions in (c). Box plot center line represents the median, box plot limits – 25th to 75th percentiles, whiskers extend from the box limit to the most extreme value no further than 1.5 * inter-quartile range from the box limit. c-d, $n_{super}$ = 1560, $n_{standard}$ = 5440, $n_{low}$ = 1556, $n_{absent}$ = 23615 regions. e) Proportion of the genome comprised within different classes of MCDs.

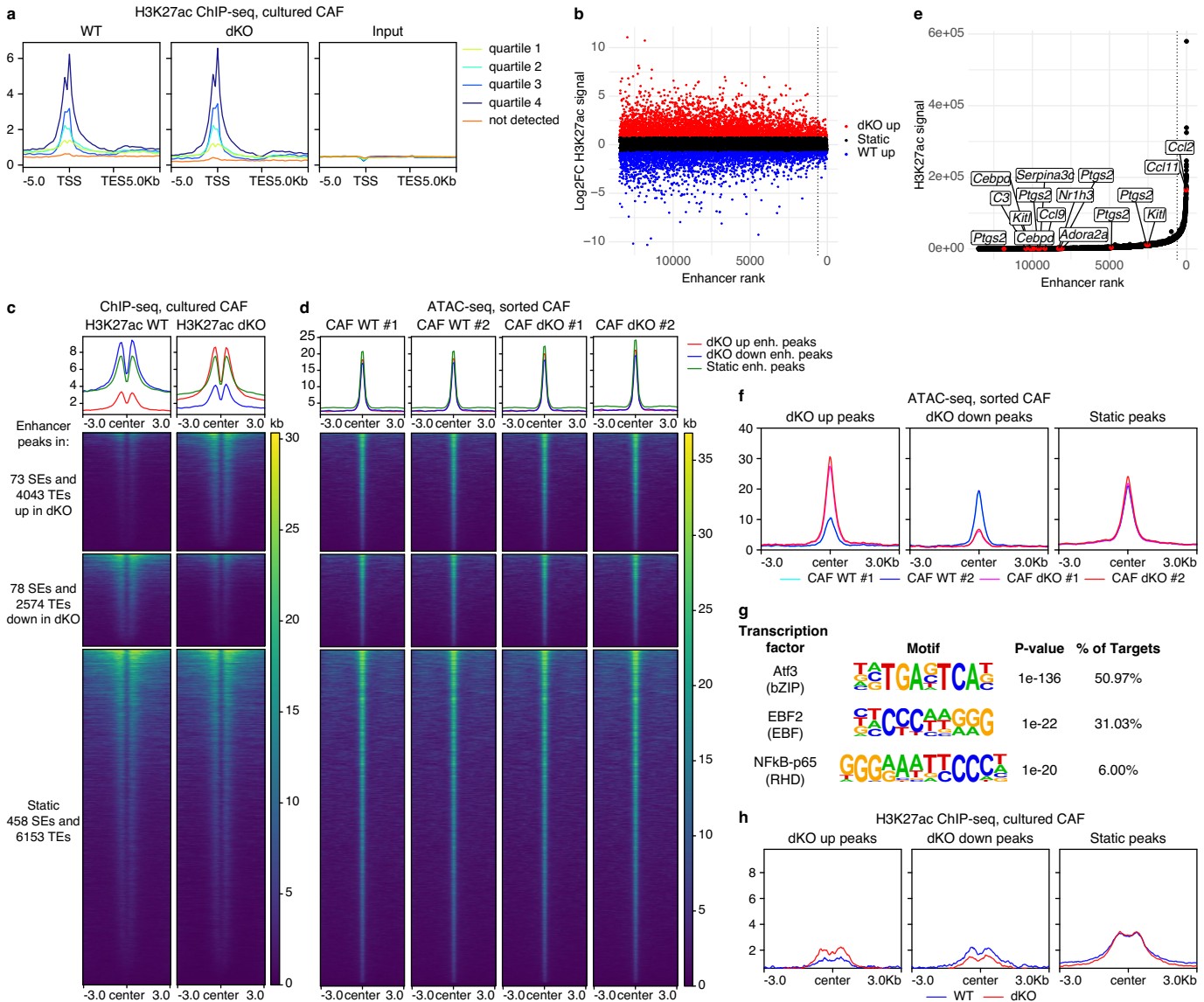

**Extended Data Fig. 7 | Enhancer and chromatin accessibility analysis in the absence of macroH2A in CAFs.** a) Metagene profile of H3K27ac levels in serum-stimulated cultured CAFs across genes binned into quartiles according to their expression levels. Genes not detected are either not expressed or not mappable. n = 3000 randomly selected genes within each group. b) Changes in H3K27ac signal at all detected enhancers in cultured CAFs depicted by MA plot. TEs and SEs are separated by a vertical dashed line. Enhancers with a log2 fold change > 0.75 are shown in color for the respective genotype. c) Heatmap of H3K27ac ChIP signal in serum-stimulated cultured CAFs centered around ATAC peaks located in enhancers that gain, lose or maintain static H3K27ac

levels in dKO vs. WT, $n_{dKO\,up}$ = 6659, $n_{dKO\,down}$ = 5211, $n_{Static}$ = 18961. Number of TEs and SEs noted for each cluster. d) ATAC-seq signals at peaks in (c). e) Hockey plot highlighting TEs and SEs gaining H3K27ac in dKO within 50 kb of upregulated inflammatory genes (red) among all TEs and SEs ranked by H3K27ac signal in cultured CAFs. f) Average profile of ATAC-seq signal at differentially accessible regions in dKO vs. WT sorted CAFs, $n_{dKO\,up}$ = 667, $n_{dKO\,down}$ = 668, $n_{Static}$ = 3000 randomly selected non-changing peaks. g) Top 3 hits of HOMER TF motif analysis of regions of increased accessibility in dKO vs. WT sorted CAFs. Fisher Exact test P-values shown. h) H3K27ac ChIP signal in serum-stimulated cultured CAFs at ATAC-seq regions defined in (j). The same scale was used as in (g) for comparison.

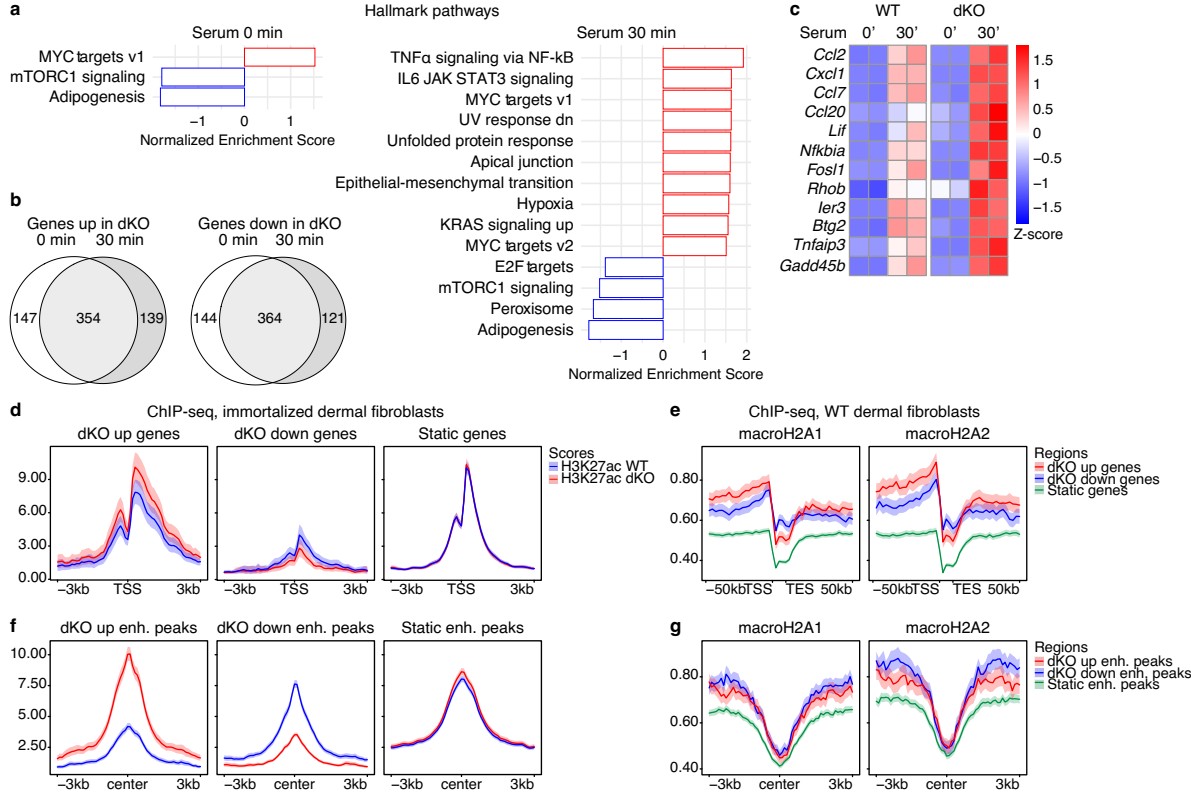

**Extended Data Fig. 8 | Transcriptomic and enhancer analysis in the absence of macroH2A in iDFs.** a) Significant Hallmark pathways in GSEA analysis of dKO vs. WT performed in iDFs prior to (0 min) and post serum stimulation (30 min). b) Venn diagrams representing the extent of overlap between pre- and post-serum stimulation in genes up- and downregulated by macroH2A deficiency in iDFs. c) Heatmap of a subset of inflammatory genes hyper-induced by serum stimulation in the absence of macroH2A. d) Average profile of H3K27ac ChIP signal in serum-stimulated WT vs. dKO iDFs at promoters of all differentially

expressed and static genes of matched expression levels following serum stimulation, $n_{dKO\,up}$ = 494, $n_{dKO\,down}$ = 485, $n_{Static}$ = 2937. e) Average profile of macroH2A1 and macroH2A2 signal[62] in dermal fibroblasts at genes described in (d). f) As in (d), for enhancer H3K27ac peaks differentially enriched in H3K27ac, $n_{dKO\,up}$ = 1438, $n_{dKO\,down}$ = 1764, $n_{Static}$ = 7037. g) Average profile of macroH2A1 and macroH2A2 signal in dermal fibroblasts at regions described in (f). d-g, mean signal value and 95% CI as determined by bootstrap shown.

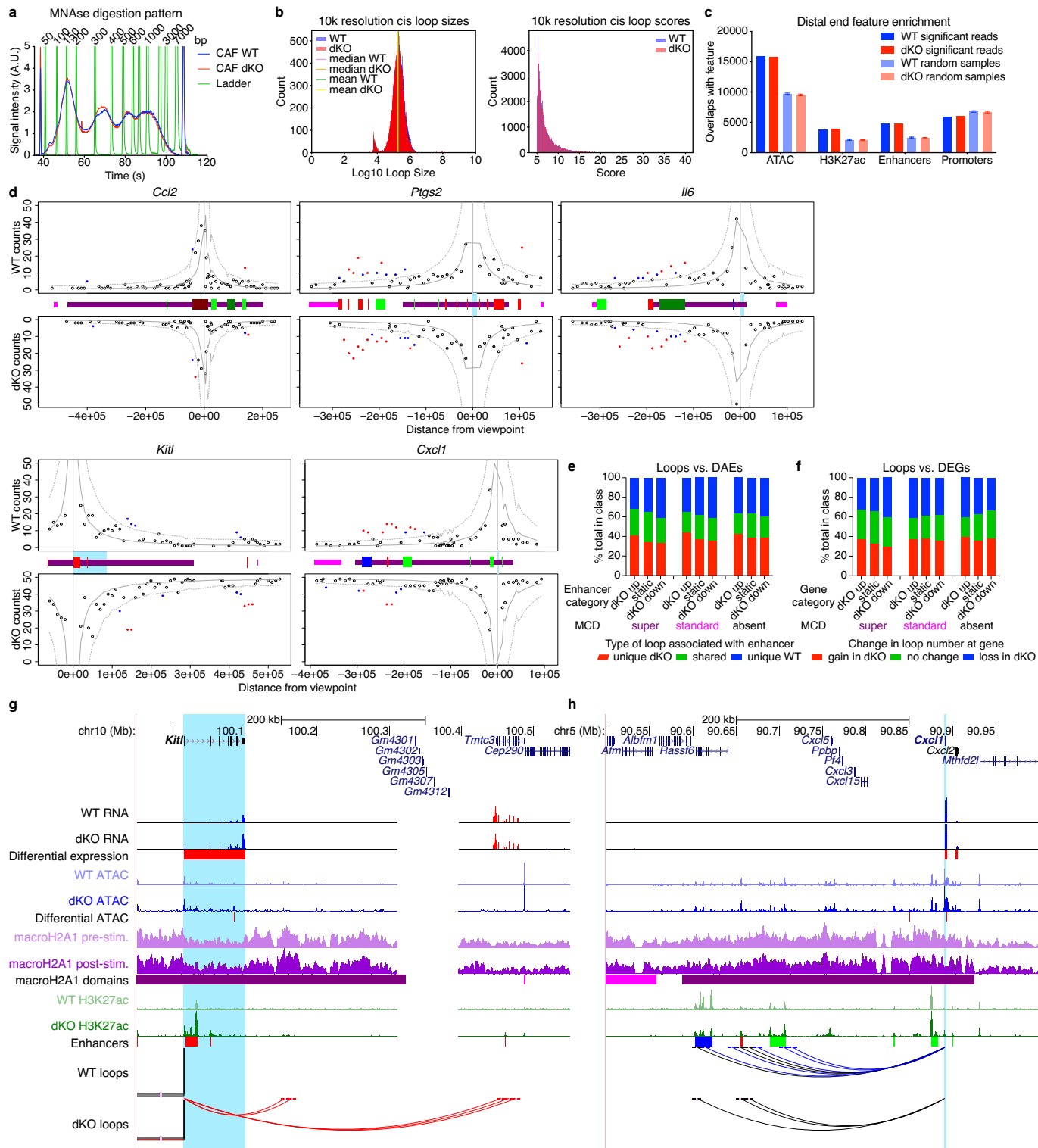

**Extended Data Fig. 9 | See next page for caption.**

**Extended Data Fig. 9 | pcMicro-C analysis in WT and dKO CAFs.** a) Size distribution of MNAse-digested double cross-linked input chromatin for Micro-C. b) Distribution of sizes and scores of chromatin loops called at 10 kb resolution, retained for further analysis. c) Enrichment of indicated functional elements at the distal end of loops, compared to a random distribution. Number of overlaps of significantly called interactions with genomic features are shown in solid color for n = 1 biological sample. Random regions of the same length and count of the significant regions are chosen, and the overlaps with genomic features are counted. The operation is permuted n = 100 times and the average number of overlaps is shown in light color. Error bars represent 95% confidence intervals. d) Bait map plots depicting all called interactions for the bait located at the indicated gene promoter. Dotted lines represent significance thresholds. Only highly significant loops (passing threshold = 5, red dots) are considered. Colored bars represent enhancers and MCDs as in (g). e) Proportions of enhancers associated with changes in chromatin looping, according to changes in their H3K27ac level and location within MCDs. f) Proportions of genes associated with net gain or loss of loops, according to changes in their expression level and location within MCDs. g) UCSC browser screenshots of the *Kitl* locus and its chromatin environment. Bars under RNA-seq and ATAC-seq tracks indicate significantly up- (red) or downregulated (blue) genes or accessible regions in dKO vs. WT sorted CAFs. Bars under macroH2A CUT&RUN tracks indicate 'super' (purple) and "standard" (magenta) macroH2A chromatin domains. Below H3K27ac tracks, bright and dark bars indicate TEs and SEs, respectively; red, blue, and green denote gain, loss, and no change, respectively, of H3K27ac level in dKO vs. WT CAFs. Chromatin loops at 10 kb resolution, originating at the promoter of the highlighted gene, are shown in red if specific for the dKO, blue for the WT, and black if shared. h) as in e, for the *Cxcl1* locus.

# Reporting Summary

## Statistics

For all statistical analyses, confirm that the following items are present in the figure legend, table legend, main text, or Methods section.

| n/a | Confirmed | |
|---|---|---|
| ☐ | ☒ | The exact sample size (*n*) for each experimental group/condition, given as a discrete number and unit of measurement |
| ☐ | ☒ | A statement on whether measurements were taken from distinct samples or whether the same sample was measured repeatedly |
| ☐ | ☒ | The statistical test(s) used AND whether they are one- or two-sided<br>*Only common tests should be described solely by name; describe more complex techniques in the Methods section.* |
| ☒ | ☐ | A description of all covariates tested |
| ☐ | ☒ | A description of any assumptions or corrections, such as tests of normality and adjustment for multiple comparisons |
| ☐ | ☒ | A full description of the statistical parameters including central tendency (e.g. means) or other basic estimates (e.g. regression coefficient) AND variation (e.g. standard deviation) or associated estimates of uncertainty (e.g. confidence intervals) |
| ☐ | ☒ | For null hypothesis testing, the test statistic (e.g. *F*, *t*, *r*) with confidence intervals, effect sizes, degrees of freedom and *P* value noted<br>*Give P values as exact values whenever suitable.* |
| ☒ | ☐ | For Bayesian analysis, information on the choice of priors and Markov chain Monte Carlo settings |
| ☒ | ☐ | For hierarchical and complex designs, identification of the appropriate level for tests and full reporting of outcomes |
| ☐ | ☒ | Estimates of effect sizes (e.g. Cohen's *d*, Pearson's *r*), indicating how they were calculated |

*Our web collection on statistics for biologists contains articles on many of the points above.*

## Software and code

Policy information about availability of computer code

| Data collection | Flow cytometry: FacsDiVa (BD Biosciences) v8.0.2<br>qPCR: CFX Manager (Bio-Rad) v3.1<br>Automated microscopy: Gen5 (Agilent) v3.12 |
|---|---|
| Data analysis | No custom code was generated for this study. All scripts used for bulk RNA seq, scRNA seq, ATAC seq, ChIP seq and CUT&RUN data analysis in this study are available from the corresponding author upon request.<br><br>Flow cytometry:<br>FCS Express (De Novo Software) v7.12<br><br>Statistics:<br>Prism (GraphPad) v9.5.1<br><br>RNA-seq:<br>salmon v1.2.1<br>DESeq2 v1.36.0<br>fgsea v1.22.0<br>HOMER v4.10<br>limma v3.54.0<br><br>scRNA_seq: |

cellranger v7.0.1 (10X Genomics)
Seurat v4.0
SCPA v1.5.1
Monocle3 v1.3.1
singleR v1.10.0
miloR v1.5.0
Augur v1.0.3
CellChat v1.6.1
SeuratWrappers v0.3.0
SingleCellExperiment v1.12
PerformanceAnalytics v2.0.4

Spatial transcriptomics:
spaceranger v2.0.0 (10X Genomics)
psych v2.3.3

TCGA data analysis:
TCGAbiolinks v2.25.3
edgeR v3.40.1
IOBR v0.99.9

ChIP-seq, CUT&RUN, ATAC-seq:
NGmerge v.0.3
bowtie2 v.2.4.1
samtools v.1.9
deepTools v.3.5.1
MACS2 v.2.1.0
bedtools v.2.29.2
Trimmomatic v.0.36
ROSE v0.11
SEACR v.1.3

Micro-C:
BWA v0.7.17-r1188
PairTools v1.0.2
CHiCAGO v1.2.0
UCSC-Utils v2.9

For manuscripts utilizing custom algorithms or software that are central to the research but not yet described in published literature, software must be made available to editors and reviewers. We strongly encourage code deposition in a community repository (e.g. GitHub). See the Nature Portfolio guidelines for submitting code & software for further information.

## Data

Policy information about availability of data

All manuscripts must include a data availability statement. This statement should provide the following information, where applicable:
- Accession codes, unique identifiers, or web links for publicly available datasets
- A description of any restrictions on data availability
- For clinical datasets or third party data, please ensure that the statement adheres to our policy

The transcriptomic and epigenomic datasets including raw and processed sequencing data generated and analyzed during the current study are available in the GEO repository under accession number GSE200751 (https://www.ncbi.nlm.nih.gov/geo/query/acc.cgi?acc=GSE200751) and in this article's table files. TCGA melanoma data is publicly available through the NCI Genomic Data Commons (GDC) data portal under project ID TCGA-SKCM (https://portal.gdc.cancer.gov/projects/TCGA-SKCM). The human pan-cancer scRNA-seq dataset mined in this study is available in the GEO repository under accession number GSE210347 (https://www.ncbi.nlm.nih.gov/geo/query/acc.cgi?acc=GSE210347). The mouse M25 (GRCm38.p6) genome assembly and gene set used for transcriptomic and epigenomic analysis are available at Gencode (https://www.gencodegenes.org/mouse/release_M25.html). All other data supporting the findings of this study is available from the corresponding author upon request.

## Research involving human participants, their data, or biological material

Policy information about studies with human participants or human data. See also policy information about sex, gender (identity/presentation), and sexual orientation and race, ethnicity and racism.

| Reporting on sex and gender | This study does not involve human participants or generate human data. |
| Reporting on race, ethnicity, or other socially relevant groupings | This study does not involve human participants or generate human data. |
| Population characteristics | This study does not involve human participants or generate human data. |
| Recruitment | This study does not involve human participants or generate human data. |

| Ethics oversight | This study does not involve human participants or generate human data. |
|---|---|

Note that full information on the approval of the study protocol must also be provided in the manuscript.

# Field-specific reporting

Please select the one below that is the best fit for your research. If you are not sure, read the appropriate sections before making your selection.

☒ Life sciences ☐ Behavioural & social sciences ☐ Ecological, evolutionary & environmental sciences

For a reference copy of the document with all sections, see nature.com/documents/nr-reporting-summary-flat.pdf

# Life sciences study design

All studies must disclose on these points even when the disclosure is negative.

| Sample size | Based on the nature of these studies, no sample-size calculation was performed. Sample size was chosen based on available number of animals or cell lines, attempting to maximize the number of biological replicates analyzed whenever possible. |
|---|---|
| Data exclusions | No data was excluded. Biological samples were excluded only as a result of sample preparation or data acquisition failure. |
| Replication | All experiments were replicated with the exception of Supplemental Figure 4n and o, H3K27ac ChIP-seq and macroH2A1 CUT&RUN. Multiple cohorts of mice were used as available due to limited breeding output and comparable mouse age required at induction in WT and dKO. Importantly, for tumor analysis experiments, each data point represents a distinct animal and therefore a biological replicate. For tumor immunophenotyping experiments, data shown is a sum of at least 2 experiments, each with at least 4 animals per genotype. Tumor induction – Summary of 8 experiments, WT/dKO mice in each as follows: 7/6, 4/6, 4/5, 5/5, 4/4, 4/4, 5/5, 5/4. RNA-seq – Summary of 3 experiments, 6 mice/genotype (2 in each experiment) Sorted CD8 RNA-seq – 1 experiment, 4 mice/genotype scRNA-seq – 3 experiments, 1/genotype in each experiment Sorted CAFs RNA-seq – 1 experiment, 4 mice/genotype Spatial transcriptomics – 1 experiment, 1 mouse/genotype Human melanoma CAF macroH2A and cytokine quantification – macroH2A western blot 3 technical replicates, ELISA 8 technical replicates for 11 independent biological samples. Sorted CAF ATAC-seq – 1 experiment, 2 mice/genotype Cultured CAF Micro-C – 1 experiment, 2 technical replicates/condition Supplementary Figure 4n was a confirmation at the protein level of extensively replicated transcriptional changes (Figures 4d, f), so we did not consider replicates were necessary. The findings in figure 4o were confirmed by RNA-seq on replicates independent from this material in an experiment currently in Supplemental Figure 8b. In our experience, and standard in the field, the highly reproducible nature of ChIP-seq does not require technical replicates. No additional CAF lines were available as biological replicates. MacroH2A1 cut&run was benchmarked against existing ChIP-seq data (Supplemental Figure 6a) and was highly similar. |
| Randomization | No samples were allocated to experimental groups as no treatments were tested on any type of subject in this study. Comparisons were performed on murine-derived samples according to the genotype of the animal they were derived from. Experiments not involving mice were based on molecular/cellular biology readouts where acquisition and analysis were performed automatically, in the same manner for all samples, and did not require randomization. For TCGA analysis, samples were allocated to primary/metastatic cohorts based on the appropriate recorded metadata, and into macroH2A high and low categories based on the 1st and 3rd tercile of MACROH2A1 or MACROH2A2 expression; no covariates were included in the analysis. For analysis of human melanoma CAFs, the available cell lines were grouped into MACROH2A2 high and low based on MACROH2A2 protein level measured by western blotting; no covariates were included in the analysis. |
| Blinding | For most experiments, blinding was not possible because the same investigator(s) who generated the mice determined their genotype, performed tumor induction, processed samples, acquired and analyzed the data. Histopathologic scoring of murine tumors was performed in a blinded manner, where information on mouse genotype, gender or age was withheld from pathologists. Experiments involving molecular/cellular biology readouts were acquired and analyzed automatically, using identical parameters for all samples, and therefore blinding was not relevant. Knowledge of sample genotype was necessary in order to perform the analysis. |

# Reporting for specific materials, systems and methods

We require information from authors about some types of materials, experimental systems and methods used in many studies. Here, indicate whether each material, system or method listed is relevant to your study. If you are not sure if a list item applies to your research, read the appropriate section before selecting a response.

## Materials & experimental systems

| n/a | Involved in the study |
|-----|----------------------|
| ☐ | ☒ Antibodies |
| ☐ | ☒ Eukaryotic cell lines |
| ☒ | ☐ Palaeontology and archaeology |
| ☐ | ☒ Animals and other organisms |
| ☒ | ☐ Clinical data |
| ☒ | ☐ Dual use research of concern |
| ☒ | ☐ Plants |

## Methods

| n/a | Involved in the study |
|-----|----------------------|
| ☐ | ☒ ChIP-seq |
| ☐ | ☒ Flow cytometry |
| ☒ | ☐ MRI-based neuroimaging |

# Antibodies

| | |
|---|---|
| Antibodies used | Antibody catalogue numbers, manufacturer websites and their dilutions are provided in Table 8 for all commercial antibodies, and a publication reference for one non-commercial antibody. In addition to Table 8, the following antibodies were used for epigenomic profiling by ChIP and CUT&RUN, respectively: H3K27ac: 13-0045, Epicypher, lot # 20120001-28, 4 µg/reaction macroH2A1: ab37264, Abcam, lot # GR278020-1, 1 µg/reaction |
| Validation | Antibodies were validated by manufacturer, by publications cited by manufacturer, and in the case of the one non-commercial antibody, as detailed in our study for its relevant application by using macroH2A-deficient tissue. Links and species/application-relevant validation statements are provided in Table 8. In addition to Table 8: Epicypher H3K27ac 13-0045 validation data is provided by manufacturer here https://www.epicypher.com/products/antibodies/ snap-chip-certified-antibodies/histone-h3k27ac-antibody-snap-chip-certified ; antibody is validated for ChIP-Seq in mouse, among others. Abcam macroH2A1 ab37264 validation data is provided for CUT&RUN in Supplemental Figure 6a. |

# Eukaryotic cell lines

Policy information about cell lines and Sex and Gender in Research

| | |
|---|---|
| Cell line source(s) | Murine cells used in this study were derived directly from our mouse colony. Human melanoma CAF primary cultures were obtained from the NCI patient-derived models repository, and the laboratory of Dr. Andrew Aplin. Male and female CAFs (mouse and human) were used in this study. |
| Authentication | For murine CAFs, genotyping was performed for engineered genes of interest. Human CAFs were identified by positive detection of CAF-specific markers and non-detectable levels of melanoma markers. |
| Mycoplasma contamination | All cell lines were routinely tested for mycoplasma by PCR and were negative. |
| Commonly misidentified lines (See ICLAC register) | No such cell lines were used. |

# Animals and other research organisms

Policy information about studies involving animals; ARRIVE guidelines recommended for reporting animal research, and Sex and Gender in Research

| | |
|---|---|
| Laboratory animals | The strain used was generated in-house by breeding B6;FVB-Tg(Tyr-cre/ERT2)13Bos Braf<tm1Mmcm> Pten<tm1Rdp> and 129S6.Cg-Macroh2a2<tm1.1Peh> Macroh2a1<tm1Peh> or 129S6/SvEvTac mice. Mice were housed in a facility with specified pathogen-free (SPF) health status, in individually ventilated cages at 21-22ºC and 39-50% relative humidity, on a 7AM-7PM light cycle, with free access to food and water. |
| Wild animals | No wild animals are used in this study. |
| Reporting on sex | Animals of both sexes were included in the (immuno)phenotypic characterization of murine melanomas. Differences in tumor growth between sexes were tested for, and were found to be non-significant (Supplemental Figure 1b). Females were used for scRNA-seq and spatial transcriptomics, to avoid the potential impact of sex-specific transcripts on integration. |
| Field-collected samples | No field-collected samples are used in this study. |
| Ethics oversight | The Icahn School of Medicine at Mount Sinai's Institutional Animal Care and Use Committee approved the study protocol (Protocol # LA11-00122). |

Note that full information on the approval of the study protocol must also be provided in the manuscript.

# Plants

| Seed stocks | No plants are used in this study |
|---|---|
| Novel plant genotypes | No plants are used in this study |
| Authentication | No plants are used in this study |

# ChIP-seq

## Data deposition

☒ Confirm that both raw and final processed data have been deposited in a public database such as GEO.

☒ Confirm that you have deposited or provided access to graph files (e.g. BED files) for the called peaks.

**Data access links**
*May remain private before publication.*

https://www.ncbi.nlm.nih.gov/geo/query/acc.cgi?acc=GSE200751

**Files in database submission**

```
Accession      Title              Release date  Status   Supplemen-
                                                         tary files
--------------------------------------------------------------------------
GSE200751      MacroH2A restricts melanoma progression via inhibition of chemokine expression in cancer-associated
fibroblasts.  Aug 01, 2023   approved  None
--------------------------------------------------------------------------
GSE200725      MacroH2A restricts melanoma progression via inhibition of chemokine expression in cancer-associated
fibroblasts [RNA-seq]  Aug 01, 2023   approved  None
GSM6042770     Mouse # 64727 melanoma RNA-seq  Aug 01, 2023   approved  SF
GSM6042771     Mouse # 64747 melanoma RNA-seq  Aug 01, 2023   approved  SF
GSM6042772     Mouse # 64799 melanoma RNA-seq  Aug 01, 2023   approved  SF
GSM6042773     Mouse # 64803 melanoma RNA-seq  Aug 01, 2023   approved  SF
GSM6042774     Mouse # 64805 melanoma RNA-seq  Aug 01, 2023   approved  SF
GSM6042775     Mouse # 64808 melanoma RNA-seq  Aug 01, 2023   approved  SF
GSM6042776     Mouse # 64660 melanoma RNA-seq  Aug 01, 2023   approved  SF
GSM6042777     Mouse # 64661 melanoma RNA-seq  Aug 01, 2023   approved  SF
GSM6042778     Mouse # 64982 melanoma RNA-seq  Aug 01, 2023   approved  SF
GSM6042779     Mouse # 65018 melanoma RNA-seq  Aug 01, 2023   approved  SF
GSM6042780     Mouse # 65020 melanoma RNA-seq  Aug 01, 2023   approved  SF
GSM6042781     Mouse # 65041 melanoma RNA-seq  Aug 01, 2023   approved  SF
GSM6042782     Mouse # 65662 tumor-infiltrating CD8 RNA-seq  Aug 01, 2023   approved  SF
GSM6042783     Mouse # 65664 tumor-infiltrating CD8 RNA-seq  Aug 01, 2023   approved  SF
GSM6042784     Mouse # 65666 tumor-infiltrating CD8 RNA-seq  Aug 01, 2023   approved  SF
GSM6042785     Mouse # 65668 tumor-infiltrating CD8 RNA-seq  Aug 01, 2023   approved  SF
GSM6042786     Mouse # 65674 tumor-infiltrating CD8 RNA-seq  Aug 01, 2023   approved  SF
GSM6042787     Mouse # 65675 tumor-infiltrating CD8 RNA-seq  Aug 01, 2023   approved  SF
GSM6042788     Mouse # 65677 tumor-infiltrating CD8 RNA-seq  Aug 01, 2023   approved  SF
GSM6042789     Mouse # 65678 tumor-infiltrating CD8 RNA-seq  Aug 01, 2023   approved  SF
GSM6042790     Mouse # 66561 CAF RNA-seq  Aug 01, 2023   approved  SF
GSM6042791     Mouse # 66562 CAF RNA-seq  Aug 01, 2023   approved  SF
GSM6042792     Mouse # 66565 CAF RNA-seq  Aug 01, 2023   approved  SF
GSM6042793     Mouse # 66566 CAF RNA-seq  Aug 01, 2023   approved  SF
GSM6042794     Mouse # 66554 CAF RNA-seq  Aug 01, 2023   approved  SF
GSM6042795     Mouse # 66555 CAF RNA-seq  Aug 01, 2023   approved  SF
GSM6042796     Mouse # 66558 CAF RNA-seq  Aug 01, 2023   approved  SF
GSM6042797     Mouse # 66559 CAF RNA-seq  Aug 01, 2023   approved  SF
GSM6042798     iDF WT serum 0 min replicate 1 RNA-seq  Aug 01, 2023   approved  SF
GSM6042799     iDF WT serum 0 min replicate 2 RNA-seq  Aug 01, 2023   approved  SF
GSM6042800     iDF WT serum 30 min replicate 1 RNA-seq  Aug 01, 2023   approved  SF
GSM6042801     iDF WT serum 30 min replicate 2 RNA-seq  Aug 01, 2023   approved  SF
GSM6042802     iDF dKO serum 0 min replicate 1 RNA-seq  Aug 01, 2023   approved  SF
GSM6042803     iDF dKO serum 0 min replicate 2 RNA-seq  Aug 01, 2023   approved  SF
GSM6042804     iDF dKO serum 30 min replicate 1 RNA-seq  Aug 01, 2023   approved  SF
GSM6042805     iDF dKO serum 30 min replicate 2 RNA-seq  Aug 01, 2023   approved  SF
--------------------------------------------------------------------------
GSE200734      MacroH2A restricts melanoma progression via inhibition of chemokine expression in cancer-associated
fibroblasts [epigenomics]  Aug 01, 2023   approved  None
GSM6042890     Mouse # 66620 CAF ATAC-seq  Aug 01, 2023   approved  BW
GSM6042891     Mouse # 66626 CAF ATAC-seq  Aug 01, 2023   approved  BW
GSM6042892     Mouse # 66623 CAF ATAC-seq  Aug 01, 2023   approved  BW
GSM6042893     Mouse # 66624 CAF ATAC-seq  Aug 01, 2023   approved  BW
GSM6042894     CAF culture from mouse # 66391 H3K27ac ChIP-seq  Aug 01, 2023   approved  BW
GSM6042895     CAF culture from mouse # 66383 H3K27ac ChIP-seq  Aug 01, 2023   approved  BW
GSM6042896     CAF culture genomic DNA input for ChIP-seq  Aug 01, 2023   approved  BW
```

GSM6042897    CAF culture from mouse # 66391 macroH2A1 CUT&RUN prior to serum stimulation  Aug 01, 2023   approved  BW

GSM6042898    CAF culture from mouse # 66391 macroH2A1 CUT&RUN 30 minutes after serum stimulation  Aug 01, 2023   approved  BW

GSM6042899    CAF culture from mouse # 66391 IgG background for CUT&RUN  Aug 01, 2023   approved  BW

GSM6042900    H3K27ac ChIP-seq in iDF dKO cells  Aug 01, 2023   approved  BW

GSM6042901    H3K27ac ChIP-seq in iDF WT cells  Aug 01, 2023   approved  BW

GSM6042902    iDF culture genomic DNA input for ChIP-seq  Aug 01, 2023   approved  BW

--------------------------------------------------------------------

GSE229530    MacroH2A restricts melanoma progression via inhibition of chemokine expression in cancer-associated fibroblasts [scRNA-seq]  Aug 01, 2023   approved  RDATA RDATA

GSM7164976    Mouse #66193 scRNA-seq  Aug 01, 2023   approved  TSV TSV MTX

GSM7164977    Mouse #66208 scRNA-seq  Aug 01, 2023   approved  TSV TSV MTX

GSM7164978    Mouse #66691 scRNA-seq  Aug 01, 2023   approved  TSV TSV MTX

GSM7164979    Mouse #66675 scRNA-seq  Aug 01, 2023   approved  TSV TSV MTX

GSM7164980    Mouse #19332 scRNA-seq  Aug 01, 2023   approved  TSV TSV MTX

GSM7164981    Mouse #19323 scRNA-seq  Aug 01, 2023   approved  TSV TSV MTX

--------------------------------------------------------------------

GSE229531    MacroH2A restricts melanoma progression via inhibition of chemokine expression in cancer-associated fibroblasts [3D chromatin]  Aug 01, 2023   approved  BIGINTERACT BIGINTERACT

GSM7164982    CAF culture from mouse # 66391 pcMicro-C replicate 1  Aug 01, 2023   approved  None

GSM7164983    CAF culture from mouse # 66391 pcMicro-C replicate 2  Aug 01, 2023   approved  None

GSM7164984    CAF culture from mouse # 66383 pcMicro-C replicate 1  Aug 01, 2023   approved  None

GSM7164985    CAF culture from mouse # 66383 pcMicro-C replicate 2  Aug 01, 2023   approved  None

--------------------------------------------------------------------

GSE229532    MacroH2A restricts melanoma progression via inhibition of chemokine expression in cancer-associated fibroblasts [spatial transcriptomics]  Aug 01, 2023   approved  RDATA RDATA

GSM7164986    Mouse #19333 spatial transcriptomics  Aug 01, 2023   approved  H5 JSON PNG CSV

GSM7164987    Mouse #19331 spatial transcriptomics  Aug 01, 2023   approved  H5 JSON PNG CSV

--------------------------------------------------------------------

**Genome browser session**
(e.g. UCSC)

https://genome.ucsc.edu/s/filipd02/CAF%20Filipescu%20v2
 https://genome.ucsc.edu/s/filipd02/iDF%20Filipescu

## Methodology

**Replicates**

In our experience, and standard in the field, the highly reproducible nature of ChIP-seq does not require technical replicates. No additional CAF lines were available as biological replicates. MacroH2A1 cut&run was benchmarked against existing ChIP-seq data (Supplemental Figure 6a) and was highly similar.

**Sequencing depth**

CAF H3K27ac ChIP - single end
Sample 66391_H3K27ac 66383_H3K27ac Inputs_combined
Total reads 39383904 51206219 70931993
Aligned to mouse 37083059 43981970 69796720
Filtered 30290128 39456067 46240441
Deduplicated 13464611 10736503 34167706

CAF macroH2A1 CUT&RUN - paired end
Sample 66591-U-m1 66591-S-m1 66391-IgG
Total reads 29912046 35920306 35371080
Aligned to mouse 26191146 31614828 26824800
Filtered 19569422 24532310 19528262
Deduplicated 19079270 22770678 17738696

iDF H3K27ac ChIP  - single end
Sample iDF WT H3K27ac iDF dKO H3K27ac input mixed
Total reads 53295577 54864962 86801179
Aligned to mouse 36864021 39644575 86190317
Filtered 30730014 34341625 63956202
Deduplicated 24792362 26175630 61629179

**Antibodies**

H3K27ac: 13-0045, Epicypher, lot # 20120001-28, 4 µg/reaction

**Peak calling parameters**

```
#Read mapping:
bowtie2 -p ${PPN} -x $INDEXDIR --local --very-sensitive-local --no-unal -U ${sample}_out.fastq.gz 2> ${OUTDIR}${sample}_bowtie2.log -S ${sample}.sam --un ${sample}_unaligned.fastq
samtools sort -@ ${PPN}-1 ${sample}.sam -O bam -o ${sample}_sorted.bam
samtools index -@ ${PPN} ${sample}_sorted.bam
samtools view -@ ${PPN}-1 ${sample}_sorted.bam chr{1..19} chrX chrY -q 30 -b -o ${sample}_filtered.bam
samtools index -@ ${PPN} ${sample}_filtered.bam
samtools markdup -@ ${PPN}-1 -r -S ${sample}_filtered.bam ${sample}_dedup.bam
samtools index -@ ${PPN} ${sample}_dedup.bam

#Peak calling CAF:
macs2 callpeak -t H3K27ac-merge.bam -c Input.bam --name H3K27ac-1e-7 --extsize 300 --to-large --nomodel -f BAM -g 3.0e9 -s 75 -p
```

```
1e-7 --outdir /output/ --verbose 2

#Peak calling iDF:
macs2 callpeak -t iDF-master-WT_dKO-H3K27ac_dedup.bam -f BAM -c iDF-mixed-input_dedup.bam --outdir /output/ -n iDF-master-
WT_dKO-H3K27ac_q1e-9-g mm --extsize 300 -q 1e-9
```

| Data quality | After removal of low-quality and duplicated reads, and peak calling on master bam files, ChIP-seq and input reads were normalized for sequencing depth using counts per million (CPM). CPM fold enrichment (FE) was calculated for peaks in each sample over input.<br><br>Peaks Above 5 FE Below 5 FE % above 5 fe<br>CAF WT 33700 6832 83.14%<br>CAF dKO 39162 1371 96.62%<br>IDF WT 30857 2386 92.82%<br>IDF DKO 30096 3146 90.54% |
|---|---|
| Software | Adapters were trimmed with Trimmomatic v.0.36, followed by alignment to the mm10 assembly with bowtie2 v.2.4.1. Low-quality (MAPQ < 30), mitochondrial genome and duplicated reads were removed using samtools v.1.9 and genome coverage calculation for visualization purposes on the UCSC genome browser was performed using deepTools v.3.5.1 excluding blacklisted regions. The bam files of WT and dKO samples were concatenated into a master bam file, used to call significant peaks with matching input controls using MACS2 v.2.1.0. Significance q-value cut-offs were determined post-hoc, testing several q-values based on signal to background ratio. Quantification of reads in significant peak for all samples was preformed using bedtools v.2.29.2 multicov. Traditional (TEs) and super-enhancers (SEs) were called based on H3K27ac enrichment using the ROSE v0.11 algorithm (Rank Ordering of Super-Enhancers) with stitching distance 12.5kb and TSS exclusion zone size 2.5kb. The ROSE algorithm was also used to extract H3K27ac levels at TEs and SEs for WT and dKO samples individually. Average H3K27ac signal across all elements was calculated for each sample and further used for normalization between samples. Log2 fold change ratio of normalized signal (-0.75 < log2FC < 0.75) was used to call differential TEs and SEs. All other enhancers were considered static. |

# Flow Cytometry

## Plots

Confirm that:

☒ The axis labels state the marker and fluorochrome used (e.g. CD4-FITC).

☒ The axis scales are clearly visible. Include numbers along axes only for bottom left plot of group (a 'group' is an analysis of identical markers).

☒ All plots are contour plots with outliers or pseudocolor plots.

☒ A numerical value for number of cells or percentage (with statistics) is provided.

## Methodology

| Sample preparation | Resected BRAFV600E/PTEN-deficient melanomas were cut into 1-2mm fragments. For immunophenotyping and CD8+ T cell sorting, tumor fragments were digested in RPMI containing 400 U/ml collagenase IV (Gibco), 100 U/ml hyaluronidase (Sigma) and 100 μg/ml DNAse (Roche) at 37ºC for 1 hour, aspirated 5 times through a 14G needle, and filtered through a 70 μm cell strainer. Immune cells were enriched by centrifugation through a discontinuous 40/90 Percoll (GE Healthcare Life Sciences) gradient. For sCAF sorting, tumor fragments were digested using the Tumor Dissociation Kit, mouse (Miltenyi) in DMEM using the soft/medium protocol according to manufacturer's instructions. |
|---|---|
| Instrument | Analysis: LSRFortessa (BD Biosciences); sorting: FACSAria III SORP (BD Biosciences) |
| Software | FacsDiVa (BD Biosciences) v8.0.2 |
| Cell population abundance | Sorting was performed in "purity" mode and sort performance was determined to be >95% using Accudrop beads.<br>Cell numbers were insufficient to determine purity for sorted CD8 T cells. Transcriptomic analysis performed on the sorted T cells suggests these are pure populations based on expression of key cell type-specific genes.<br>Similarly, all sorted CAFs were used for downstream studies. Sorted CAFs that went into culture were confirmed to be free of tumor cell contamination and expressed CAF marker PDGFRa (Supplemental Figure 4k, l). |
| Gating strategy | Single events were gated on FSC-A/FSC-W and SSC-A/SSC-W. Cells were gated on FSC-A/SSC-A. Live cells were gated on viability dye (DAPI for sorting, LIVE/DEAD Fixable Blue Dead Cell Stain for analysis) negativity. Immune cells were gated on CD45 positivity. Further gating is illustrated in Supplemental Figure 2c and follows the strategy used in Salmon et al., 2016 (https://pubmed.ncbi.nlm.nih.gov/27096321/). Boundaries between "positive" and "negative" staining cell populations were defined based on the histogram valley of the respective channel. When sorting CAFs from whole tumor single cell suspensions, immune cells present in each sample, expected to be positive for CD45 and negative for PDGFRa, were used to define PDGFRa gate boundaries. |

☒ Tick this box to confirm that a figure exemplifying the gating strategy is provided in the Supplementary Information.

