## [Peer Review File · Nature Cell Biology]

Peer Review Information

Journal: Nature Cell Biology

Manuscript Title: MacroH2A restricts inflammatory gene expression in melanoma cancer-associated fibroblasts by coordinating chromatin looping

Corresponding author name(s): Dan Filipescu, Emily Bernstein

Editorial Notes:

Reviewer Comments & Decisions:

Decision Letter, initial version:
--

*Please delete the link to your author homepage if you wish to forward this email to co-authors.

Dear Dr Bernstein,

Your manuscript, "MacroH2A restricts melanoma progression via inhibition of inflammatory gene expression in cancer-associated fibroblasts", has now been seen by 3 referees, who are experts in histone variant biology (referee 1); CAFs (referee 2); and melanoma and immunology (referee 3). As you will see from their comments (attached below) they find this work of potential interest, but have raised substantial concerns, which in our view would need to be addressed with considerable revisions before we can consider publication in Nature Cell Biology.

Nature Cell Biology editors discuss the referee reports in detail within the editorial team, including the chief editor, to identify key referee points that should be addressed with priority, and requests that are overruled as being beyond the scope of the current study. To guide the scope of the revisions, I have listed these points below. We are committed to providing a fair and constructive peer-review process, so please feel free to contact me if you would like to discuss any of the referee comments further.

In particular, it would be essential to:

A) Address the concerns related to the proposed mechanism as questioned by Reviewer 1:

"The attempt to mechanistically explain how macroH2A ablation affects inflammatory genes falls short."

"...but fails to explain how macroH2A actually affects gene transcription."

"It should be clarified if the genes upregulated in CAFs are actually mediating the phenotype (tumor growth or immunosuppression). Can this be shown in vivo? If not, could a co-culture experiment help? If the signals are secreted, media transfer experiments can be helpful."

"It should be clarified if the loss of macroH2A in tumor cells contribute to the phenotyp and CAFs respond. Again, do CAFs respond to a secreted or a cell-boud signal. Is this signal the same in WT and dKO cancer cells? Do WT and dKO CAFs respond differently to WT cancer cells. Do WT CAFs respond differently to"

B) It would be a plus if you could add data to strengthen the human relevance as suggested by Reviewer 2 and 3;

C) All other referee concerns pertaining to strengthening existing data, providing controls, methodological details, clarifications and textual changes, as applicable should also be addressed.

D) Finally please pay close attention to our guidelines on statistical and methodological reporting (listed below) as failure to do so may delay the reconsideration of the revised manuscript. In particular please provide:

We would be happy to consider a revised manuscript that would satisfactorily address these points, unless a similar paper is published elsewhere, or is accepted for publication in Nature Cell Biology in the meantime.

- ensure that it conforms to our format instructions and publication policies (see below and <https://www.nature.com/nature/for-authors>).

- provide a point-by-point rebuttal to the full referee reports verbatim, as provided at the end of this letter.

- provide the completed Reporting Summary (found here <https://www.nature.com/documents/nr-reporting-summary.pdf>). This is essential for reconsideration of the manuscript will be available to editors and referees in the event of peer review. For more information see <http://www.nature.com/authors/policies/availability.html> or contact me.

When submitting the revised version of your manuscript, please pay close attention to our [href="https://www.nature.com/nature-portfolio/editorial-policies/image-integrity">Digital Image Integrity Guidelines](https://www.nature.com/nature-portfolio/editorial-policies/image-integrity). and to the following points below:

Nature Cell Biology is committed to improving transparency in authorship. As part of our efforts in this direction, we are now requesting that all authors identified as 'corresponding author' on published papers create and link their Open Researcher and Contributor Identifier (ORCID) with their account on the Manuscript Tracking System (MTS), prior to acceptance. ORCID helps the scientific community achieve unambiguous attribution of all scholarly contributions. You can create and link your ORCID from the home page of the MTS by clicking on 'Modify my Springer Nature account'. For more information please visit www.springernature.com/orcid.

This journal strongly supports public availability of data. Please place the data used in your paper into a public data repository, or alternatively, present the data as Supplementary Information. If data can only be shared on request, please explain why in your Data Availability Statement, and also in the correspondence with your editor. Please note that for some data types, deposition in a public repository is mandatory - more information on our data deposition policies and available repositories appears below.

[REDACTED]

We would like to receive a revised submission within six months.

We hope that you will find our referees' comments, and editorial guidance helpful. Please do not hesitate to contact me if there is anything you would like to discuss.

Best wishes,
Zhe Wang

Zhe Wang, PhD

Senior Editor
Nature Cell Biology

Tel: +44 (0) 207 843 4924
email: zhe.wang@nature.com

Reviewers' Comments:

Reviewer #1:

Remarks to the Author:

MacroH2A histone variants have tumor suppressive roles in many cancer types. This was pioneered by the last author's lab showing that loss of macroH2A increased the aggressiveness of melanoma cells reflected in increased metastasis in a xenograft model. In their current manuscript by Filipescu et al. they present the logical and long-awaited follow-up describing how the systemic absence of all macroH2A impinges on melanoma growth in a fully competent mouse model. Increased tumor growth in macroH2A-deficient mice is associated with a change in tumor-infiltrating immune cells and an inflammatory expression signature in bulk tissue. Specifically, results suggest a reduction in cytotoxic T cells and an increase in tumor suppressive myeloid cells. Single-cell RNA-seq indicates that differential abundances in two major cell populations that are a subpopulation of tumor cells with a stemness signature (NC-Rxr) and cancer-associated fibroblast (CAF). CAFs are shown to be the main source of upregulated cytokines and chemokines likely to trigger the immunophenotype. Up and downregulated genes and their enhancers are enriched in macroH2A. Upregulation is associated with increased enhancer activity.

Strength:

- First study assessing the impact of macroH2As on cancer in a fully immunocompetent mouse model.
- Dissection of changes in a complex cancer tissue type. Identification of cell types and populations affected by the loss of macroH2A.
- Demonstration that macroH2A has a repressive role in inflammatory genes in CAF.

Weakness:

- Figures are difficult to read (overloaded, non-intuitive abbreviations, missing labels and graphical legends).
- The attempt to mechanistically explain how macroH2A ablation affects inflammatory genes falls short.

The strengths of the study clearly lies on the cell biology of a complex tumor and how the loss of macroH2A histone variants affects the tumor microenvironment. This is an important and interesting aspect that will warrant publication if further strengthened. As of now the study is descriptive with a very high level of quality and quantity.

In addition, the authors provide a wealth of genomic data including from isolated primary cells that will be a valuable resource for the field. As in many studies before the genomic data indicates previously known associations but fails to explain how macroH2A actually affects gene transcription. Elucidating the exact mechanism is beyond scope of the presented study.

MAJOR COMMENTS:

1. It should be clarified if the genes upregulated in CAFs are actually mediating the phenotype (tumor growth or immunosuppression). Can this be shown in vivo? If not, could a co-culture experiment help? If the signals are secreted, media transfer experiments can be helpful.
2. It should be clarified if the loss of macroH2A in tumor cells contribute to the phenotyp and CAFs respond. Again, do CAFs respond to a secreted or a cell-boud signal. Is this signal the same in WT and dKO cancer cells? Do WT and dKO CAFs respond differently to WT cancer cells. Do WT CAFs respond differently to

OTHER GENERAL COMMENTS:

Given our expertise in genomic studies, we provide detailed suggestions on how to improve the data presentation (see further below). I consider that no additional data is needed on this side.

Statistical tests are missing in many figure panels. It has to be clear without reading the figure legend or the results text if differences are significant or not.+

Legends are missing in several panels (eg 5c and related).

Please reduce the number of panels to the most informative ones.

Label the panels with the cells / tissues analyzed e.g. 'sorted CAFs' in 5e.

In my humble opinion the authors could consider a different flow of their results. The single-cell RNA-seq is most informative and could be presented earlier (maybe right after presenting the phenotype) to introduce the two main results: 1. the change in tumor-composition and 2. changes in inflammatory gene expression. All other results could then be built on this.

The presentation of the immunophenotype is confusing and scattered. Consider compiling all data on immune cells in one coherent figure.

It should be clarified if there is anything special to the repressed 39 inflammatory genes compared to other macroH2A-regulated genes.

SUGGESTIONS FOR SPECIFIC FIGURES:

Figures 4-5

The NC Rxrg cluster has a high contribution to the dKO bulk gene expression changes, comparable to the CAF Meg3 which are the focus of further analysis. I am missing a description of genes differentially expressed in melanoma, in particular NC Rxrg cells. What was the rationale for excluding this tumor cell population from further analysis? See my major comment.

Figure 5c could perhaps be converted to a heatmap summarizing the expression differences to make it more visual and equivalent to other representations.

Figure 5g lacks statistical testing results.

Could the Milo cell abundance method be applied to the re-clustered immune cell data shown in Figure S4i to assess changes in composition of immune subtypes?

Figure 6 and S6

In these figures the authors present results obtained in CAFs and iDFs, but it is not always labelled in a way that allows easy interpretation without resorting to the legend or main text. For example, panels c to e versus panels h to j could be more readily interpreted by indicating the cells analyzed. This applies to the Supplementary Figure 6 and also to other parts of the manuscript, I think it would improve readability and faster interpretation of the results.

As far as I know, this is the first time a profiling of macroH2A by CUT&RUN is reported in a manuscript. The authors show a depletion from the gene bodies of highly expressed genes and some example loci, and in both cases the results are highly reminiscent of previous published results for ChIP-seq. Given the novelty of the experimental technique applied to this histone variant, I would appreciate a better generic description of the CUT&RUN results and a comparison to previously performed ChIP-seq (eg dermal fibroblast data) as benchmark, at least: the total number of enriched regions called ("MCDs"), size distribution of the identified MCDs, total % of genome coverage by all MCDs.

The results in Figure 6c and d show an increased H3K27Ac enrichment at enhancers and promoters of the selected 39 upregulated inflammatory genes, however the difference seems mild when compared to the genome-wide effect shown in Figure S6a and the shaded area of the profiles (as I understand, a confidence interval) are largely overlapping. Could the authors perform a statistic test on the area around the enhancer centers or TSS to show that this is statistically significant? Perhaps a Fisher exact test on the read counts?

MacroH2A1 has a higher enrichment in the 39 inflammatory genes than on thousands of static genes. This is also true for all dKO up and down genes (Figure 6k). The current presentation is misleading. I think a more logical description would be to first show the genome-wide analysis and then present that this holds true for the subset of genes that the authors propose are linked to the phenotype. Although the authors put the spotlight on this set of genes, I think it is important to highlight that the findings regarding mH2A occupancy and H3K27Ac changes are not exclusive of these genes but general for all mH2A-sensitive genes.

Given the importance placed on these 39 DEGs, it would be important to know how many of them are detected as upregulated in both scRNA-Seq and sorted-bulk RNA-Seq datasets. Moreover, are they also deregulated in the NC clusters or exclusively in the CAFs?

Similarly, the analysis of H3K27Ac in Figure 6c and d could also be performed for all enhancer peaks within 50kb and the TSS of all the dKO up and dKO down genes, providing a general analysis of how the changes in H3K27Ac enrichment relate to the expression changes in the dKO CAFs.

Which test was performed in Figure S6 L and N?

Reviewer #2:

Remarks to the Author:

A. Summary of the key results

MacroH2A as a key epigenetic suppressor of tumour growth and progression in melanoma. The mechanism is shown to be likely caused by alteration in immunoregulation in the tumour mediated by the production of inflammatory cytokines by cancer activated fibroblasts. This was shown by use of a murine double-KO for macroH2A (H2afy and H2afy2) and autochthonous immunocompetent model for melanoma. The results are largely based on transcriptomic (both bulk and single cell) analyses of melanoma induced in this mouse model compared to mice that do not have the double-KO for macroH2A. The role of macroH2A in the epigenetic landscape in various cell types in the tumour microenvironment is also explored

B. Originality and significance: if not novel, please include reference

This study uncovers several key findings in the context of melanoma:

- Firstly, the heterogeneity of the melanoma tumour microenvironment is delineated on a cellular transcriptional level herein. This has already been done for an orthotopic transplantation model of melanoma (Davidson et al., 2020; <https://pubmed.ncbi.nlm.nih.gov/32433953/>). However, to my knowledge this is first time this has been performed in this autochthonous murine model of melanoma.
- Secondly, to my knowledge this is the first study to perform single cell RNA sequencing on macroH2A deficient murine melanoma models.
- Thirdly, a link between the function between cancer activated fibroblasts and immunoregulation in melanoma is re-enforced by this study. This hypothesis/discovery is not novel in and of itself (Papaccio et al., 2021; <https://pubmed.ncbi.nlm.nih.gov/34298873/>), however this study adds valuable high-resolution data, previously unexplored, that adds to this body of evidence.
- Finally, a link between the function of macroH2A and cancer activated fibroblast behaviour is linked and underpinned as the consequential difference in melanoma progression with or without macroH2A. Namely, melanomas with cancer activated fibroblasts with dysfunctional macroH2A have increased pro-inflammatory cytokine expression by means of an altered epigenetic landscape in absence of macroH2A. The consequence of increased pro-inflammatory cytokine expression is an immunosuppressive tumour microenvironment, that promotes melanoma growth and progression.

C. Data & methodology: validity of approach, quality of data, quality of presentation

Validity of approach:

- A fine approach and line of inquiry for the issue of delineating macroH2A function in the melanoma microenvironment.

Quality of data:

- Data is generally of high quality.
- Concern over number of replicates in the scRNAseq data.

Data presentation:

- Data is generally presented well, with little issue for legibility or font size.

D. Appropriate use of statistics and treatment of uncertainties

The approaches used are generally appropriate and well performed.

E. Conclusions: robustness, validity, reliability

The final line states that "the convergence of these mechanisms would predict poor response of

macroH2A-low tumors to immunotherapy, suggesting a possible biomarker application to stratify melanoma patients.” However this is speculative and not sufficiently evidenced in this study – how does the mouse model respond to immunotherapy based upon macroH2A status, for example?

F. Suggested improvements: experiments, data for possible revision

As noted in E., treating dKO and WT mice by immunotherapy to observe difference in response would strengthen this conclusion. Additionally/alternatively, a human tissue microarray (TMA) demonstrating variable macroH2A protein levels and correlating this with differences in immune cell infiltration as observed in the mouse model would be of merit.

Only two tumours per condition are profiled by scRNAseq which is insufficient given the prominence these data have in the study. At least one more replicate per condition should be included which would also permit interrogation in the differences of tumour composition.

CIBERSORT should be used to also determine whether there is a difference in myeloid cell populations when stratifying patients based on MACROH2A1 and MACROH2A2 expression.

When stratifying tumours by MACROH2A2 expression (Fig 2c-d) there is a significant reduction in CD8 T cell markers but no difference in the inferred prevalence of CD8 T cells from CIBERSORT. How is this accounted for?

G. References: appropriate credit to previous work?

Manuscript is well referenced.

H. Clarity and context: lucidity of abstract/summary, appropriateness of abstract, introduction and conclusions.

The manuscript is delightfully written and easy to follow. The abstract is an appropriate summary of the findings. The introduction is similarly concise and appropriate. The conclusions also follow and are written well. However, as noted earlier, the final sentence of the discussion is not directly supported by the evidence presented and so must be supplied as an optimistic speculation. If, however, there were direct evidence to support this claim, the manuscript would be greatly elevated from identification of biologically relevant and interesting mechanism to also that of clinical and therapeutic relevance.

Reviewer #3:

Remarks to the Author:

In this manuscript, the authors tested macroH2A knockout in a BRAFV600E;PTEN-deficient melanoma model. They found by that macroH2A accelerated tumor growth but exhibited insignificant differences in other metrics by histology. The authors then performed bulk RNA-sequencing on the KO and WT tumors and discovered upregulation of genes involved in myeloid cell recruitment and downregulation of genes involved in myeloid cell inhibition and cytotoxic T cell activation. Flow cytometry of the murine model exhibited an increase in monocytes in the dKO and a decrease in CD8 T cells. scRNA-seq revealed an enrichment of de-differentiated neural crest cells and immunosuppressive Mrc1+ macrophages in dKO tumors. The authors identify 3 CAF clusters, which upregulated the gene signature associated with the dKO bulk RNA-seq and overexpressed the myeloid chemoattractant genes upregulated in the bulk RNA-seq. The authors also see H2afy2 expression is specific to CAF cells in the scRNA-seq. Cultured CAF cells exhibited higher expression of the chemokines in the dKO with serum stimulation, suggesting CAF cell-intrinsic mechanisms of immunosuppression with macroH2A

KO. ChIP-seq for H3K27ac showed a distinct landscape between dKO and WT CAFs but a minimal change in chromatin accessibility by ATAC-seq, and macroH2A1 CUT&RUN showed enrichment proximal to the dKO upregulated inflammatory genes. Collectively, the authors conclude that macroH2A KO relieves repression at inflammatory genes, eliciting CAF-induced promotion of an immunosuppressive tumor microenvironment. The primary finding of the paper is identification of a previously undescribed role for macroH2A in suppressing inflammatory signaling in fibroblasts, as well as evidence for macroH2A suppression of melanoma cell de-differentiation. These findings were largely convincing and are interesting contributions to the literature but require some further connection to human tumors to establish relevance to human biology.

Major Comments:

- The authors' primary finding relates to the effect of macroH2A KO in the CAFs and suggest a cell-intrinsic mechanism by in vitro studies. This provides new information about the role of macroH2A in CAFs but is limited in data to support relevance in human samples. Do human melanomas with downregulated macroH2A exhibit increased infiltration of the myeloid compartment and decreased infiltration of lymphocytes (e.g., by histology)? In probing published human scRNA-seq datasets, does the stromal compartment exhibit a macroH2A-high and -low state? If so, do these states correlate with a distinct immune landscape?
- When comparing the GSEA data from the murine model versus the human samples, several of the pathways upregulated in the murine dKO model are downregulated in the human MACROH2A1 low samples (TNF α signaling via NFKB, IL6 Jak Stat3 signaling, IFN α /g response). What would explain this signature inversion? In looking at the myeloid-related genes (as in Fig 2B) in the human datasets, do they have a similar expression pattern? Does testing a gene signature for myeloid cell activation, cytotoxic T cell activation, etc. show a parallel between the murine and human samples?

Minor Comments:

- Interesting that the H2A2-low tumors exhibit a significant decrease in the cytotoxic genes (Fig 2C) but an insignificant difference in CD8 T cell abundance (Fig 2D). What explains why H2A1 would significantly decrease CD8 T cells but not H2A2? If the samples are stratified that are low in both H2A1 and H2A2, is the difference even more pronounced?
- Is there an increase in myeloid cells by histology in the dKO? Where do these cells localize? Localization to the stroma would further support a CAF-centric effect.
- Please include a description of the acronyms used in Fig 4A in the figure, legend, or text.
- After filtering the cells, are there equivalent numbers of cells for the WT and dKO samples in Fig 4? Normalizing the values in Fig 4C to the total number of cells per tumor (% filtered cells of parent tumor) would correct for sampling bias.
- CAF Tnc drive the dKO transcriptional profile most significantly, but CAF Meg3 shows the biggest change in abundance in dKO. Do the authors believe all 3 CAF subpopulations contribute to this dKO phenotype?
- Fig S5g shows H2afy2 expression in CAFs and H2afy expression in all clusters. This is assumedly in the WT tumors but isn't specified in the figure or corresponding legend. Please show or describe the sample origin of this expression.
- In looking at the CAF ATAC-seq for the inflammatory genes differentially expressed in the CAF RNA-seq and exhibiting differential H3K27ac, are these genes in the newly open peaks or unchanged but open in both dKO and WT?
- Line 347 misspelled word "previously" should be "previously."
- Since macroH2A2 was shown to be specific to CAFs, would it be expected that CUT&RUN for H2A2 would exhibit distinct enrichment localization?

- Figure S4i isn't referenced in the text, and lines 352-353 should reference Fig 6f and 6g.

FINANCIAL AND NON-FINANCIAL COMPETING INTERESTS – the authors must include one of three

declarations: (1) that they have no financial and non-financial competing interests; (2) that they have financial and non-financial competing interests; or (3) that they decline to respond, after the Author Contributions section. This statement will be published with the article, and in cases where financial and non-financial competing interests are declared, these will be itemized in a web supplement to the article. For further details please see <https://www.nature.com/licenceforms/nrg/competing-interests.pdf>.

Methods should be written concisely, but should contain all elements necessary to allow interpretation and replication of the results. As a guideline, Methods sections typically do not exceed 3,000 words. The Methods should be divided into subsections listing reagents and techniques. When citing previous methods, accurate references should be provided and any alterations should be noted. Information must be provided about: antibody dilutions, company names, catalogue numbers and clone numbers for monoclonal antibodies; sequences of RNAi and cDNA probes/primers or company names and catalogue numbers if reagents are commercial; cell line names, sources and information on cell line identity and authentication. Animal studies and experiments involving human subjects must be reported in detail, identifying the committees approving the protocols. For studies involving human subjects/samples, a statement must be included confirming that informed consent was obtained. Statistical analyses and information on the reproducibility of experimental results should be provided in a section titled "Statistics and Reproducibility".

All Nature Cell Biology manuscripts submitted on or after March 21 2016 must include a Data availability statement as a separate section after Methods but before references, under the heading "Data Availability". For Springer Nature policies on data availability see <http://www.nature.com/authors/policies/availability.html>; for more information on this particular policy see <http://www.nature.com/authors/policies/data/data-availability-statements-data-citations.pdf>. The Data availability statement should include:

- Accession codes for primary datasets (generated during the study under consideration and designated as "primary accessions") and secondary datasets (published datasets reanalysed during the study under consideration, designated as "referenced accessions"). For primary accessions data should be made public to coincide with publication of the manuscript. A list of data types for which submission to community-endorsed public repositories is mandated (including sequence, structure, microarray, deep sequencing data) can be found here <http://www.nature.com/authors/policies/availability.html#data>.

- Unique identifiers (accession codes, DOIs or other unique persistent identifier) and hyperlinks for datasets deposited in an approved repository, but for which data deposition is not mandated (see here for details <http://www.nature.com/sdata/data-policies/repositories>).
- At a minimum, please include a statement confirming that all relevant data are available from the authors, and/or are included with the manuscript (e.g. as source data or supplementary information), listing which data are included (e.g. by figure panels and data types) and mentioning any restrictions on availability.
- If a dataset has a Digital Object Identifier (DOI) as its unique identifier, we strongly encourage including this in the Reference list and citing the dataset in the Methods.

We recommend that you upload the step-by-step protocols used in this manuscript to the Protocol Exchange. More details can found at www.nature.com/protocolexchange/about.

All imaging data should be accompanied by scale bars, which should be defined in the legend. Cropped images of gels/blots are acceptable, but need to be accompanied by size markers, and to retain visible background signal within the linear range (i.e. should not be saturated). The boundaries of panels with low background have to be demarked with black lines. Splicing of panels should only be considered if unavoidable, and must be clearly marked on the figure, and noted in the legend with a statement on whether the samples were obtained and processed simultaneously. Quantitative comparisons between samples on different gels/blots are discouraged; if this is unavoidable, it should only be performed for samples derived from the same experiment with gels/blots were processed in parallel, which needs to be stated in the legend.

- For line art, graphs, charts and schematics we prefer Adobe Illustrator (.AI), Encapsulated PostScript (.EPS) or Portable Document Format (.PDF). Files should be saved or exported as such directly from the application in which they were made, to allow us to restyle them according to our journal house style.
- We accept PowerPoint (.PPT) files if they are fully editable. However, please refrain from adding PowerPoint graphical effects to objects, as this results in them outputting poor quality raster art. Text used for PowerPoint figures should be Helvetica (preferred) or Arial.
- We do not recommend using Adobe Photoshop for designing figures, but we can accept Photoshop generated (.PSD or .TIFF) files only if each element included in the figure (text, labels, pictures, graphs, arrows and scale bars) are on separate layers. All text should be editable in 'type layers' and line-art such as graphs and other simple schematics should be preserved and embedded within 'vector smart objects' - not flattened raster/bitmap graphics.
- Some programs can generate Postscript by 'printing to file' (found in the Print dialogue). If using an application not listed above, save the file in PostScript format or email our Art Editor, Allen Beattie for advice (a.beattie@nature.com).

The total number of Supplementary Figures (not including the “unprocessed scans” Supplementary Figure) should not exceed the number of main display items (figures and/or tables (see our Guide to Authors and March 2012 editorial <http://www.nature.com/ncb/authors/submit/index.html#suppinfo>; <http://www.nature.com/ncb/journal/v14/n3/index.html#ed>). No restrictions apply to Supplementary Tables or Videos, but we advise authors to be selective in including supplemental data.

GUIDELINES FOR EXPERIMENTAL AND STATISTICAL REPORTING

REPORTING REQUIREMENTS – We are trying to improve the quality of methods and statistics reporting in our papers. To that end, we are now asking authors to complete a reporting summary that collects information on experimental design and reagents. The Reporting Summary can be found here <https://www.nature.com/documents/nr-reporting-summary.pdf>) If you would like to reference the guidance text as you complete the template, please access these flattened versions at <http://www.nature.com/authors/policies/availability.html>.

Information on how many times each experiment was repeated independently with similar results

needs to be provided in the legends and/or Methods for all experiments, and in particular wherever representative experiments are shown.

Author Rebuttal to Initial comments

Dear Dr. Zhe and Reviewers,

We thank the reviewers for their time and effort in reviewing our manuscript now titled "*MacroH2A restricts inflammatory gene expression in melanoma cancer-associated fibroblasts by coordinating chromatin interactions*". The reviewers made favorable comments about our study, including:

- "the strength of the study clearly lies on the cell biology of a complex tumor and how the loss of macroH2A histone variants affects the tumor microenvironment"
- "the manuscript is delightfully written and easy to follow"
- "these findings were largely convincing and are interesting contributions to the literature"

The reviewers also made helpful suggestions and provided constructive critiques, which we believe have **significantly improved the quality and impact** of our manuscript and enhanced our mechanistic understanding of macroH2A-deficiency in melanoma, with a focus on the CAF compartment. Our **substantially revised manuscript** contains two heavily amended Main figures (**Figs. 5 & 7** and their corresponding Supplementary figures), consisting of almost entirely new data. Here, we highlight some of the key new data:

1. To mechanistically explain how macroH2A ablation affects inflammatory gene expression in CAFs, we performed **promoter-capture Micro-C** (pcMicro-C, a chromosome conformation capture technique that identifies chromatin loops with nucleosomal precision, here using promoters as the bait) to investigate the role of macroH2A in 3D chromatin architecture. Notably, this analysis has been computationally integrated with extensive transcriptomics and epigenomics data. We find the following:
 - I. loss of macroH2A is associated with global changes in chromatin contacts (**Fig. 7a**)

- II. a significant correlation between dKO activated enhancers (gain of H3K27ac) with dKO-specific promoter-enhancer loops (**Fig. 7b**)
- III. enrichment of inflammatory genes among those that gain promoter-enhancer contacts with increased enhancer activity in the dKO setting (**Fig. 7d**).

Overall, this new data suggests that macroH2A variants regulate the 3D chromatin landscape, and furthers our understanding of the regulatory elements behind those genes that drive the dKO CAF inflammatory phenotype.

2. Second, as requested by reviewer 2, we carried out an additional **single cell RNA-sequencing** (scRNA-seq) experiment of BRAF^{V600E}/PTEN-deficient melanomas for a total of three WT and three dKO mice, generating a dataset of *~24,000 high-quality cells* (**Fig. 3a**). While this additional replicate has not changed our overall conclusions, attesting to the robustness of our original scRNA-seq analysis, it has allowed us to call cell clusters in more detail, as well as to add statistical significance to differences in the proportions of cells (**Fig. 3d**). This newly analyzed data continues to demonstrate significant changes in the NC and CAF populations, as well as various immune cell types (immunosuppressive myeloid, and cytotoxic T cells) in the dKO vs. WT melanomas (**Fig. 3d**). This data further maintains the role of CAFs as the drivers of inflammatory mediators in the TME (**Fig. 4a-c**). Moreover, it has now allowed us to robustly call receptor-ligand interactions, which revealed the predominant role of CAFs in communicating to other cell types (**Fig. S5a, b**), and the increased communication along the CCL2, CXCL1 and IL-6 pathways from dKO CAFs to myeloid cells (**Fig. 5a**).
3. To complement the scRNA-seq data, and to identify interactions between transformed melanocytes, CAFs, and the immune compartment in their native morphological context, we performed **spatial transcriptomics** using the 10X Visium spatial gene expression assay on WT and dKO melanomas. This approach allowed us to infer functional relationships between cell types that we identified in scRNA-seq data. As suggested by reviewers, we now present evidence of spatial colocalization between CAFs and immunosuppressive myeloid cells, and their exclusion of cytotoxic T cells from their tissue niches (**Fig. 5b, c**). Moreover, genetically engineered melanoma models have limited spatial transcriptomics data; thus we hereby also provide a key resource for the melanoma community.
4. Finally, as requested by reviewers 2 and 3, we also translated our murine findings to human biology through the use of **patient-derived melanoma CAFs**, as well as analyzing a published **pan-cancer human CAF scRNA-seq dataset**. Interestingly, we find that human melanoma CAFs vary in their levels of macroH2A2, and via ELISA assays, we demonstrate that macroH2A2^{low} CAFs secrete more CCL2, CXCL1 and IL-6 upon stimulation (**Fig. 5f, g**). Moreover, by analyzing scRNA-seq data from over 56,000 CAFs across 98 tumor samples, we find *CCL2*, *CXCL1* and *IL6* are positively correlated with each other, while *IL6* and *MACROH2A2* show a negative correlation (**Fig. S5j**).

Overall, we would like to point out that our study is the first to: 1) report a cancer-associated phenotype in macroH2A-deficient animals, and 2) associate a histone variant with the non-immune tumor stroma, a compartment of increasing relevance to cancer biology. We demonstrate that CAF activation in the absence of macroH2A shapes tumor initiation and the ensuing dampened immune response by producing inflammatory cytokines that recruit pro-tumorigenic monocytes (also known as MDSCs) that can suppress T cell function. Moreover, we have utilized multiple innovative approaches, including scRNA-seq, spatial transcriptomics, and 3D chromatin structural analyses. The latter two approaches were performed during the revision process. Moreover, the use of patient-derived melanoma CAFs and mining of publicly available CAF datasets supports our findings in the human setting.

We highlighted sections of the manuscript and figure legends referring to this new data and revised analysis by using underlined text.

Please see our point-by-point responses to reviewers' comments below in blue:

Reviewer #1:

Remarks to the Author:

MacroH2A histone variants have tumor suppressive roles in many cancer types. This was pioneered by the last author's lab showing that loss of macroH2A increased the aggressiveness of melanoma cells reflected in increased metastasis in a xenograft model. In their current manuscript by Filipescu et al. they present the logical and long-awaited follow-up describing how the systemic absence of all macroH2A impinges on melanoma growth in a fully competent mouse model. Increased tumor growth in macroH2A-deficient mice is associated with a change in tumor-infiltrating immune cells and an inflammatory expression signature in bulk tissue. Specifically, results suggest a reduction in cytotoxic T cells and an increase in tumor suppressive myeloid cells. Single-cell RNA-seq indicates that differential abundances in two major cell populations that are a subpopulation of tumor cells with a stemness signature (NC-Rxr) and cancer-associated fibroblast (CAF). CAFs are shown to be the main source of upregulated cytokines and chemokines likely to trigger the immunophenotype. Up and downregulated genes and their enhancers are enriched in macroH2A. Upregulation is associated with increased enhancer activity.

Strength:

- First study assessing the impact of macroH2As on cancer in a fully immunocompetent mouse model.
- Dissection of changes in a complex cancer tissue type. Identification of cell types and populations affected by the loss of macroH2A.
- Demonstration that macroH2A has a repressive role in inflammatory genes in CAF.

Weakness:

- Figures are difficult to read (overloaded, non-intuitive abbreviations, missing labels and graphical legends).
- The attempt to mechanistically explain how macroH2A ablation affects inflammatory genes falls short.

The strengths of the study clearly lie on the cell biology of a complex tumor and how the loss of macroH2A histone variants affects the tumor microenvironment. This is an important and interesting aspect that will warrant publication if further strengthened. As of now the study is descriptive with a very high level of quality and quantity. In addition, the authors provide a wealth of genomic data

including from isolated primary cells that will be a valuable resource for the field. As in many studies before the genomic data indicates previously known associations but fails to explain how macroH2A actually affects gene transcription. Elucidating the exact mechanism is beyond scope of the presented study.

We thank the reviewer for acknowledging the significance of our study and recognizing its value. While the reviewer suggests that we do not fully address the mechanism by which macroH2A deficiency affects inflammatory gene expression, they also mention that it is “beyond the scope of the present study”. Nonetheless, we are pleased to provide new data implicating macroH2A in the regulation of 3D chromatin interactions, particularly at inflammatory gene loci in CAFs. Specific points are addressed below.

MAJOR COMMENTS:

1. It should be clarified if the genes upregulated in CAFs are actually mediating the phenotype (tumor growth or immunosuppression). Can this be shown in vivo? If not, could a co-culture experiment help? If the signals are secreted, media transfer experiments can be helpful.

We thank the reviewer for raising this important point. We have now edited the manuscript to ensure the clarity of our message regarding the contribution of inflammatory gene expression in CAFs to the phenotype, as well as experimental/computational steps to further validate this hypothesis.

First, following the reviewer’s suggestion of a co-culture experiment, we demonstrate that serum-stimulated dKO CAFs induce increased migration of WT monocytes compared to WT CAFs via transwell assays (**Fig. 5d**). Second, using our scRNA-seq dataset, which is now expanded through the addition of a third replicate per genotype, we performed ligand-receptor analysis to dissect the extent of cellular communication between CAFs and myeloid cells, and confirmed increased signaling from the former to the latter along the signaling pathways comprising the upregulated cytokines (**Fig. 5a**). Furthermore, by leveraging a new spatial transcriptomics dataset we generated in WT and dKO melanomas, we confirmed accumulation of Mac Mrc1 cells (myeloid cells with a predicted immunosuppressive profile) in the vicinity of CAFs, coupled to the exclusion of cytotoxic T cells (**Fig. 5b, c**). Taken together, these data suggest that macroH2A deficiency in CAFs is sufficient to drive increased monocyte/monocyte-derived cell accumulation in the TME.

Finally, we have now emphasized throughout the text that CCL2, CXCL1 and IL-6 are widely accepted as signals secreted by CAFs to attract myeloid cells and polarize them toward an immunosuppressive phenotype across several cancer types. The revised manuscript cites a review summarizing these studies: <https://pubmed.ncbi.nlm.nih.gov/35331673/>

2. It should be clarified if the loss of macroH2A in tumor cells contribute to the phenotype and CAFs respond. Again, do CAFs respond to a secreted or a cell-bound signal. Is this signal the same in WT and dKO cancer cells? Do WT and dKO CAFs respond differently to WT cancer cells. Do WT CAFs respond differently to (*please note this sentence was not completed by the reviewer*)

The reviewer raises a valid point regarding communication from the tumor compartment towards CAFs. Signals from tumor cells (in our case, the NC compartment) can reprogram normal tissue mesenchymal cells into CAFs, and provide one of the pathways for CAF accumulation in the TME (reviewed in <https://pubmed.ncbi.nlm.nih.gov/34670861/>). While we cannot formally rule out extrinsic factors, several lines of evidence strongly suggest the inflammatory phenotype is CAF-intrinsic.

First, while we would have been keen to perform an experiment involving stimulation of WT CAFs with conditioned medium from WT or dKO melanoma/NC cells, followed by measurement of cytokine expression in CAFs, unfortunately it is not feasible. Despite considerable efforts, we have not been able to purify NC-derived cells from our melanoma model, or establish culture conditions that select for and allow propagation of tumor cells out of the mixed primary cell culture. However, our stimulation experiments in cultured CAFs, in the absence of any active signaling from tumor cells (given their absence from these cultures), show dKO CAFs can intrinsically sustain increased inflammatory signaling (**Fig. 4f, S4n**) and functionally, lead to increased recruitment of WT monocytes *in vitro* (**Fig. 5d, S5c**). Second, iDF stimulation experiments (**Fig. S4o, S6m-o**) show that this same phenotype occurs outside of any influence of the TME, since these WT and dKO fibroblasts were derived from normal skin. Third, ligand-receptor interaction analysis shows the NC compartment actually communicates less towards CAFs in the dKO, while CAFs have a stronger strength of interaction towards NC in the dKO (**Fig. S5b**).

OTHER GENERAL COMMENTS:

Given our expertise in genomic studies, we provide detailed suggestions on how to improve the data presentation (see further below). I consider that no additional data is needed on this side.

Statistical tests are missing in many figure panels. It has to be clear without reading the figure legend or the results text if differences are significant or not.+

Details of statistical tests performed are now present in all figures, and we state in the legend when non-significant differences are not labeled. We deliberately omitted marking non-significant differences with “ns” when significant differences are also marked, in order to maintain figure clarity. For example, adding “ns” in panels such as **Fig. 2c** would render them less legible.

Legends are missing in several panels (eg 5c and related).

We apologize for any omissions and have modified the current version.

Please reduce the number of panels to the most informative ones.

We reorganized figures in the interest of clarity and moved multiple panels to the Supplement. The reviewer will now appreciate that former Figures 2 and 3 were combined and former Figures 3 and 4 have been simplified.

Label the panels with the cells / tissues analyzed e.g. 'sorted CAFs' in 5e.

We thank the reviewer for this suggestion. This is now the case for figures where data from multiple sources is shown.

In my humble opinion the authors could consider a different flow of their results. The single-cell RNA-seq is most informative and could be presented earlier (maybe right after presenting the phenotype) to introduce the two main results: 1. the change in tumor-composition and 2. changes in inflammatory gene expression. All other results could then be built on this.

We thank the reviewer for the suggestion. Accordingly, we tried to move this data forward as much as possible, especially considering the additional data added during revision. However, we believe the current flow of the paper allows the scRNA-seq data to link more readily to the aspects presented immediately after, and the data presented before provides strong evidence justifying its application, which we present in new **Fig. 2** (a merge of previous Figures 2 and 3). We hope the flow of the revised manuscript showcases the scRNA-seq and continues to build out from there.

The presentation of the immunophenotype is confusing and scattered. Consider compiling all data on immune cells in one coherent figure.

We thank the reviewer for the suggestion. We have now simplified the immunophenotypic analysis and grouped it with the bulk tumor transcriptomic analysis into revised **Fig. 2**.

It should be clarified if there is anything special to the repressed 39 inflammatory genes compared to other macroH2A-regulated genes.

While this is somewhat open to speculation, as included in the Discussion, the promoter-capture Micro-C looping data generated during revision provides some insight into this question. By intersecting genes upregulated by macroH2A loss with the *bona fide* target genes of enhancers gaining H3K27ac in the dKO (determined by pcMicro-C looping data), 7 out of 20 genes were part of inflammatory signaling pathways (**Fig. 7d**), with significant enrichment for NF- κ B targets (**Table 7**). In addition, 17 out of these 20 genes also had a net increase in the number of loops in the dKO (**Fig. 7d**). Therefore, several inflammatory genes are at the

intersection of expression, looping and enhancer activity perturbations induced by macroH2A loss. While this does not encompass all genes, it is known that IL-6 can signal in a paracrine manner, activating its own targets, and therefore some gene expression changes might be secondary to that.

SUGGESTIONS FOR SPECIFIC FIGURES:

Figures 4-5

The NC Rxrg cluster has a high contribution to the dKO bulk gene expression changes, comparable to the CAF Meg3 which are the focus of further analysis. I am missing a description of genes differentially expressed in melanoma, in particular NC Rxrg cells. What was the rationale for excluding this tumor cell population from further analysis? See my major comment.

The reviewer correctly points out that the NC Zeb2 cluster (previously annotated as Nc Rxrg but has been renamed after the additional scRNA-seq data set requested by reviewer 2) has proven difficult to analyze. The significant accumulation of NC Zeb2 cells in the dKO (**Fig. 3d**) and their de-differentiated nature (**Fig. S3g-h**) are consistent with macroH2A loss driving tumor progression, as we showed previously (Kapoor et al., 2010). However, NC Zeb2 cells displayed minimal upregulation of any Hallmark pathway in the dKO, compared to much more significant and pronounced (in terms of fold change) downregulation of pathways (**Fig. S4g**). Furthermore, among the downregulated pathways, some were seemingly antagonistic (G2M checkpoint/p53 vs. E2F targets; oxidative phosphorylation vs. hypoxia), which did not aid in informing what processes are being affected in this cluster. More importantly, because of a lack of specific markers suited for FACS, we were unable to isolate NC Zeb2 (or any other NC cells) from tumors, precluding any validation or molecular dissection of macroH2A's role in this cell type.

Figure 5c could perhaps be converted to a heatmap summarizing the expression differences to make it more visual and equivalent to other representations.

We maintained this standard representation of scRNA-seq data (now **Fig. 4b**), as in addition to differences between WT and dKO, the violin plot shape also displays the proportion of expressing cells and maximum expression value, showing which cluster is the main source of each highlighted cytokine/gene. This second point gets lost in heatmaps, where data is normalized as Z-scores for each gene to emphasize difference over absolute value.

Figure 5g lacks statistical testing results.

Statistical testing results and individual replicate values have been added to what is now **Fig. 4f**.

Could the Milo cell abundance method be applied to the re-clustered immune cell data shown in Figure S4i to assess changes in composition of immune subtypes?

We performed this analysis as suggested, and it confirmed local changes in CD8 clusters. This has been included as Fig. S3n.

Figure 6 and S6

In these figures the authors present results obtained in CAFs and iDFs, but it is not always labelled in a way that allows easy interpretation without resorting to the legend or main text. For example, panels c to e versus panels h to j could be more readily interpreted by indicating the cells analyzed. This applies to the Supplementary Figure 6 and also to other parts of the manuscript, I think it would improve readability and faster interpretation of the results.

We agree that comparing multiple cell types within the same figure can lead to confusion and have labeled these panels accordingly.

As far as I know, this is the first time a profiling of macroH2A by CUT&RUN is reported in a manuscript. The authors show a depletion from the gene bodies of highly expressed genes and some example loci, and in both cases the results are highly reminiscent of previous published results for ChIP-seq. Given the novelty of the experimental technique applied to this histone variant, I would appreciate a better generic description of the CUT&RUN results and a comparison to previously performed ChIP-seq (eg dermal fibroblast data) as benchmark, at least: the total number of enriched regions called (“MCDs”), size distribution of the identified MCDs, total % of genome coverage by all MCDs.

We thank the reviewer for the suggestions and have now included the total number of MCDs (Fig. S6b legend), panels on size distribution of the identified MCDs (Fig. S6c) and total % of

Figure R1. Comparison of MCDs called with *epic2* on macroH2A1 CUT&RUN vs. ChIP-seq data.

genome coverage by all MCDs (Fig. S6d). We observed MCDs varied greatly in size, as well as in the level of macroH2A enrichment, prompting us to stratify them into super vs. standard, using an approach adapted for enhancer analysis. While we make no claim that these categories parallel the differences between traditional and superenhancers, we observe a more pronounced link with gene/enhancer deregulation at regions highly enriched in macroH2A such as the super MCDs.

As suggested, we attempted to compare our MCDs called on CUT&RUN data in CAFs with previously called MCDs on ChIP-seq in DFs (Sun et al., 2018). However, the two had different size distributions and patterns, which was likely due to the different methods used to call them. Therefore, we attempted to use the same methodology in the hope of obtaining a more comparable result, based on a recent publication analyzing macroH2A ChIP-seq. Corujo et al. (2022) used *epic2* (Stovner and Sætrum, 2019), a peak caller tailored to broad enrichment regions, which we applied to our macroH2A1 ChIP-seq and CUT&RUN datasets. After testing multiple parameters to optimize the identification of MCDs (e.g., gap size, bin size, q-value, background), we observed that *epic2* performed well for ChIP-seq, identifying large macro domains that corresponded to highly enriched regions, while it did not perform well for CUT&RUN, missing multiple large macroH2A enriched domains previously identified by the

peak caller *Seacr*. Thus, that suitable CUT&RUN caller (Fig. R1). we believe *epic2* is not for

peak caller
Seacr
Thus, that suitable

Figure R2. Correlation of genome-wide enrichment of macroH2A between ChIP-seq and CUT&RUN methodologies. ChIP-seq and associated inputs were previously generated in dermal fibroblasts (“DF” samples), CUT&RUN was performed in serum-starved unstimulated (“U”) and serum-stimulated (“S”) WT CAFs, with associated IgG control.

identification of large domains from CUT&RUN data, and were not able to perform a domain-based comparison.

Nevertheless, we compared actual enrichment signal in the form of bigwig files, for their genome-wide correlation. Using this approach, we found a high degree of correlation ($r = 0.79$) between macroH2A1 ChIP-seq and CUT&RUN (**Fig. R2**). We unfortunately do not have space in the Supplemental figures to add this data; however, we now mention our benchmarking of macroH2A CUT&RUN vs. ChIP-seq in the Methods section: “*MacroH2A1 enrichment determined by CUT&RUN was benchmarked by correlation analysis with published macroH2A1 ChIP-seq in dermal fibroblasts (Sun 2018). Enrichment at the level of read pileups had a correlation coefficient of 0.79 (data not shown).*”

The results in Figure 6c and d show an increased H3K27Ac enrichment at enhancers and promoters of the selected 39 upregulated inflammatory genes, however the difference seems mild when compared to the genome-wide effect shown in Figure S6a and the shaded area of the profiles (as I understand, a confidence interval) are largely overlapping. Could the authors perform a statistic test on the area around the enhancer centers or TSS to show that this is statistically significant? Perhaps a Fisher exact test on the read counts?

We performed a test after binning the signal upstream, on and downstream of the elements and indeed, it is not statistically significant (**Fig. R3**). This prompted us to examine other mechanisms of gene regulation, such as chromatin looping. We used looping data to delineate promoter-enhancer associations based on interactions in 3D, moving away from proximity-based associations (which are predictions), and have now removed the data the reviewer refers to.

Figure R3. Binned H3K27ac signal at promoters and proximity-associated enhancers of 39 inflammatory genes upregulated in sorted dKO CAFs.

MacroH2A1 has a higher enrichment in the 39 inflammatory genes than on thousands of static genes. This is also true for all dKO up and down genes (Figure 6k). The current presentation is misleading. I think a more logical description would be to first show the genome-wide analysis and then present that

this holds true for the subset of genes that the authors propose are linked to the phenotype. Although the authors put the spotlight on this set of genes, I think it is important to highlight that the findings regarding mH2A occupancy and H3K27Ac changes are not exclusive of these genes but general for all mH2A-sensitive genes.

We took the reviewer's suggestion and organized **Fig. 6** accordingly, emphasizing statistically significant genome-wide associations between macroH2A occupancy and gene/enhancer dysfunction.

Given the importance placed on these 39 DEGs, it would be important to know how many of them are detected as upregulated in both scRNA-Seq and sorted-bulk RNA-Seq datasets. Moreover, are they also deregulated in the NC clusters or exclusively in the CAFs?

We include **Table R1** below to show 17 of the 39 genes derived from sorted CAFs are also significantly upregulated in CAF Meg3 of the scRNA-seq dataset, the most abundant CAF cluster that shows a 3-fold frequency increase in the dKO. Furthermore, CAF Wif1, CAF Fbn1 and CAF Lrrc15 follow in terms of most genes upregulated within that list. Among NC clusters, only NC Aqp1 displays upregulation of 2 such genes. This confirms the bulk of inflammatory gene upregulation in the dKO originates in CAFs.

Similarly, the analysis of H3K27Ac in Figure 6c and d could also be performed for all enhancer peaks within 50kb and the TSS of all the dKO up and dKO down genes, providing a general analysis of how the changes in H3K27Ac enrichment relate to the expression changes in the dKO CAFs.

As above, we revised our analysis to move away from proximity-based associations, which were not significant in the way suggested by the reviewer. We include the data (**Fig. R4**), where on average, H3K27ac at enhancers within 50 kb of TSS of all DEGs does not change in the direction of expression changes. Accordingly, we have also emphasized in the discussion that enhancer activity changes are not sufficient to explain all gene expression changes we observe.

Figure R4. H3K27ac at enhancer peaks within 50 kb of DEG promoters and control static genes with matched expression levels.

Which test was performed in Figure S6 L and N?

A chi-square test was performed, now presented in **Fig. 6b** and **6e**, with appropriate description in the legend.

Table R1. Upregulation status of inflammatory genes identified in sorted CAFs, across clusters defined by scRNA-seq in murine melanoma. A value of T (true) represents significant upregulation ($P\text{-adj} < 0.05$, $\log_2fc > 0$) of indicated gene in indicated cluster. Genes and clusters with no T values are not shown.

	CAF Meg3	CAF Wif1	Mac Mrc1	Mono-cyte	Mela-nocyte	CAF Lrrc15	NC Aqp1	CAF Fbln1	Mac Arg1	BEC	Adven-titial
Apoe	T	T	F	F	T	T	T	T	F	F	T
Hmgb1	T	F	F	F	F	F	T	T	F	F	F
Ntrk2	T	T	F	F	F	F	F	T	F	F	F
Ccl11	T	T	F	F	F	F	F	F	F	F	F
Tnfsf9	T	F	F	F	F	F	F	F	F	F	F
Thbs1	T	F	F	T	F	F	F	F	T	F	F
Cebpd	T	T	F	F	F	F	F	F	F	F	F
Tac1	T	F	F	F	F	F	F	F	F	F	F
Ccl2	T	T	F	F	F	F	F	F	F	F	F
Cp	F	T	T	F	F	T	F	T	F	F	F
Serping1	T	T	F	F	F	F	F	T	F	F	F
Cxcl1	T	T	F	T	F	T	F	T	T	T	F
Il6	T	F	F	F	F	F	F	F	F	F	F
Fosl2	T	F	F	F	F	F	F	F	F	F	F
Kitl	T	F	F	F	F	F	F	F	F	F	F
Agtr1a	F	F	F	F	F	F	F	T	F	F	F
Cfh	T	T	F	F	F	F	F	T	F	F	F
Dock10	T	T	F	F	F	F	F	T	F	F	F
Tslp	F	F	T	F	F	F	F	F	F	F	F
Apod	F	T	F	F	F	F	F	F	F	F	F
Il6st	T	F	F	F	F	F	F	F	F	F	F

Reviewer #2:

Remarks to the Author:

A. Summary of the key results

MacroH2A as a key epigenetic suppressor of tumour growth and progression in melanoma. The mechanism is shown to be likely caused by alteration in immunoregulation in the tumour mediated by the production of inflammatory cytokines by cancer activated fibroblasts. This was shown by use of a murine double-KO for macroH2A (H2afy and H2afy2) and autochthonous immunocompetent model for melanoma. The results are largely based on transcriptomic (both bulk and single cell) analyses of melanoma induced in this mouse model compared to mice that do not have the double-KO for macroH2A. The role of macroH2A in the epigenetic landscape in various cell types in the tumour microenvironment is also explored

B. Originality and significance: if not novel, please include reference

This study uncovers several key findings in the context of melanoma:

- Firstly, the heterogeneity of the melanoma tumour microenvironment is delineated on a cellular transcriptional level herein. This has already been done for an orthotopic transplantation model of melanoma (Davidson et al., 2020; <https://pubmed.ncbi.nlm.nih.gov/32433953/>). However, to my knowledge this is first time this has been performed in this autochthonous murine model of melanoma.
- Secondly, to my knowledge this is the first study to perform single cell RNA sequencing on macroH2A deficient murine melanoma models.
- Thirdly, a link between the function between cancer activated fibroblasts and immunoregulation in melanoma is re-enforced by this study. This hypothesis/discovery is not novel in and of itself (Papaccio et al., 2021; <https://pubmed.ncbi.nlm.nih.gov/34298873/>), however this study adds valuable high-resolution data, previously unexplored, that adds to this body of evidence.
- Finally, a link between the function of macroH2A and cancer activated fibroblast behaviour is linked and underpinned as the consequential difference in melanoma progression with or without macroH2A. Namely, melanomas with cancer activated fibroblasts with dysfunctional macroH2A have increased pro-inflammatory cytokine expression by means of an altered epigenetic landscape in absence of macroH2A. The consequence of increased pro-inflammatory cytokine expression is an immunosuppressive tumour microenvironment, that promotes melanoma growth and progression.

We thank the reviewer for recognizing the value of our study and emphasizing its novel findings. We hope they will appreciate the additional insight of the macroH2A-deficient TME provided by the spatial transcriptomics, 3D chromatin conformation and human data analysis included in the revised manuscript.

C. Data & methodology: validity of approach, quality of data, quality of presentation

Validity of approach:

- A fine approach and line of inquiry for the issue of delineating macroH2A function in the melanoma microenvironment.

Quality of data:

- Data is generally of high quality.
- Concern over number of replicates in the scRNAseq data.

Data presentation:

- Data is generally presented well, with little issue for legibility or font size.

D. Appropriate use of statistics and treatment of uncertainties

The approaches used are generally appropriate and well performed.

E. Conclusions: robustness, validity, reliability

The final line states that “the convergence of these mechanisms would predict poor response of macroH2A-low tumors to immunotherapy, suggesting a possible biomarker application to stratify melanoma patients.” However this is speculative and not sufficiently evidenced in this study – how does the mouse model respond to immunotherapy based upon macroH2A status, for example?

We thank the reviewer for bringing up this point. We reserved this statement for the discussion section, given its speculative nature, and now state the following: “*We speculate that the convergence of these mechanisms would predict poor response of macroH2A^{low} tumors to immunotherapy, suggesting a possible biomarker application to stratify melanoma patients*”.

We agree that testing response to immunotherapy may complement our data, but we believe this is beyond the scope of our study, in which we focus on the mechanistic understanding of macroH2A to inflammatory gene regulation in CAFs. While we considered performing immune checkpoint blockade, anti PD-1 or PD-L1 monotherapy has been attempted in this melanoma model and shows minimal effects (<https://www.ncbi.nlm.nih.gov/pmc/articles/PMC4097121/>)

see Fig. 5). Furthermore, these BRAF^{V600E}/Pten^{null} tumors respond only partially to anti-CTLA-4 + anti-PD-L1 combination therapy (<https://www.nature.com/articles/nature14404> in Figure 4), which is not surprising given the lack of neoantigens in these tumors. Given these drawbacks, and the considerable expense of the drug regimen for a cohort of 20 mice over a period of 25 days, we did not pursue this approach.

F. Suggested improvements: experiments, data for possible revision

As noted in E., treating dKO and WT mice by immunotherapy to observe difference in response would strengthen this conclusion. Additionally/alternatively, a human tissue microarray (TMA) demonstrating variable macroH2A protein levels and correlating this with differences in immune cell infiltration as observed in the mouse model would be of merit.

We have addressed point E above.

We thank the reviewer for the suggestion to consider TMAs, however, unfortunately, the small cores of tissue present in these arrays do not allow one to assess all the cell types of interest in the TME. Further, the impact of macroH2A loss on the tumor immune microenvironment appears to stem primarily from the CAF population, and thus the IHC analysis would need to target this population specifically. We note that IHC does not have the necessary resolution to evaluate macroH2A levels in CAFs vs. rest of the tumor, and would require simultaneous detection of at least one CAF marker, macroH2A1/2, and normalization controls such as histone H3. However, we now include two new sources of data that confirm our observations in human CAFs.

First, we obtained human melanoma CAF cultures from the NCI PDMR and Andrew Aplin's team at Thomas Jefferson University, who has published multiple studies using these CAFs (<https://pubmed.ncbi.nlm.nih.gov/26269601/>; <https://pubmed.ncbi.nlm.nih.gov/30115691/>). Our studies revealed an anticorrelation between endogenous macroH2A2 levels and CCL2, CXCL1, and IL-6 cytokine production in these human CAFs (**Fig. 5f, g, S5h**). While this analysis did not reach statistical significance (likely due to heterogeneous source material and/or human biological variation), it indeed paralleled our observations from the mouse model.

Second, we analyzed a published scRNA-seq dataset containing over 56,000 CAFs across 98 samples comprising multiple cancer types. By calculating pseudobulk values, we found a significant pan-cancer anticorrelation between macroH2A2 and IL-6 expression (**Fig. S5j**).

Only two tumours per condition are profiled by scRNAseq which is insufficient given the prominence these data have in the study. At least one more replicate per condition should be included which would also permit interrogation in the differences of tumour composition.

We followed the reviewer's suggestion and added a third biological replicate per genotype. While this additional replicate has not changed our overall conclusions, attesting to the robustness of our original scRNA-seq analysis, it has allowed us to call cell clusters in more detail, as well as to add statistical significance to the differences in proportions of cells (**Fig. 3d**). This newly analyzed data continues to show significant changes in the NC populations, CAFs, and various immune cell types (immunosuppressive myeloid, and cytotoxic T cells) in the dKO vs. WT melanomas (**Fig. 3d**). This data maintained the role of CAFs as drivers of the accumulation of inflammatory mediators in the TME (**Fig. 4a-c**). Furthermore, it allowed us to robustly call receptor-ligand interactions, which revealed the predominant role of CAFs in communicating to other cell types (**Fig. S5a, b**), and the increase of communication along the CCL2, CXCL1 and IL-6 pathways from dKO CAFs to myeloid cells (**Fig. 5a**).

CIBERSORT should be used to also determine whether there is a difference in myeloid cell populations when stratifying patients based on MACROH2A1 and MACROH2A2 expression.

We agree with the reviewer regarding the potential of CIBERSORT to interrogate accumulation of myeloid cells in macroH2A^{low} tumors. Therefore, we *completely overhauled* this analysis by implementing a recent analysis package, *IOBR*, which contains the CIBERSORT functionality. In contrast to our previous approach, this allowed us to use data normalized in the same manner via the robust TMM method, both for stratifying samples according to macroH2A expression, and for calculating immune population estimates. We no longer relied on CIBERSORT scores calculated by others; we compared samples in the 1st and 3rd macroH2A1 or macroH2A2 expression terciles, eliminating samples with intermediate expression levels which may have reduced the amplitude of differences observed.

The updated CIBERSORT analysis shows M2 (pro-tumor) macrophages are significantly more abundant in macroH2A2^{low} primary tumors and near significance in macroH2A1^{low} tumors (**Fig. 5e**). Of note, some myeloid subtypes are negatively correlated with macroH2A2 in metastases (**Fig. S5d**), although macroH2A^{low} tumors appear overall depleted of immune cells and conversely display increased tumor purity (**Fig. S5e**), which likely affects our ability to detect relative increases of immune subtypes via CIBERSORT. We also emphasize that the TCGA SKCM metastatic samples mainly comprise lymph node lesions, where the TME is different – both through a presence of immune populations residing in the lymph node, and a lack of skin-derived CAFs – therefore it is difficult to translate findings from our mouse model to these samples.

When stratifying tumours by MACROH2A2 expression (Fig 2c-d) there is a significant reduction in CD8 T cell markers but no difference in the inferred prevalence of CD8 T cells from CIBERSORT. How is this accounted for?

This was a shortcoming of the approach we used previously (see above). *MACROH2A2* RNA-seq counts are low and likely affected the accuracy of stratification using the previous FPKM normalization method. Our updated CIBERSORT approach now shows significant CD8 T cell depletion in macroH2A1 and 2 low tumors both in the primary and metastatic setting (**Fig. 5e, S5d**).

G. References: appropriate credit to previous work?

Manuscript is well referenced.

H. Clarity and context: lucidity of abstract/summary, appropriateness of abstract, introduction and conclusions.

The manuscript is delightfully written and easy to follow. The abstract is an appropriate summary of the findings. The introduction is similarly concise and appropriate. The conclusions also follow and are written well. However, as noted earlier, the final sentence of the discussion is not directly supported by the evidence presented and so must be supplied as an optimistic speculation. If, however, there were direct evidence to support this claim, the manuscript would be greatly elevated from identification of biologically relevant and interesting mechanism to also that of clinical and therapeutic relevance.

We thank the reviewer for acknowledging the clarity and context of our manuscript.

Reviewer #3:

Remarks to the Author:

In this manuscript, the authors tested macroH2A knockout in a BRAFV600E;PTEN-deficient melanoma model. They found by that macroH2A accelerated tumor growth but exhibited insignificant differences in other metrics by histology. The authors then performed bulk RNA-sequencing on the KO and WT tumors and discovered upregulation of genes involved in myeloid cell recruitment and downregulation of genes involved in myeloid cell inhibition and cytotoxic T cell activation. Flow cytometry of the murine model exhibited an increase in monocytes in the dKO and a decrease in CD8 T cells. scRNA-seq revealed an enrichment of de-differentiated neural crest cells and immunosuppressive Mrc1+ macrophages in dKO tumors. The authors identify 3 CAF clusters, which upregulated the gene signature associated with the dKO bulk RNA-seq and overexpressed the myeloid chemoattractant genes upregulated in the bulk RNA-seq. The authors also see H2afy2 expression is specific to CAF cells in the scRNA-seq. Cultured CAF cells exhibited higher expression of the chemokines in the dKO with serum stimulation, suggesting CAF cell-intrinsic mechanisms of immunosuppression with macroH2A KO. ChIP-seq for H3K27ac showed a distinct landscape between dKO and WT CAFs but a minimal change in chromatin accessibility by ATAC-seq, and macroH2A1 CUT&RUN showed enrichment proximal to the dKO upregulated inflammatory genes. Collectively, the authors conclude that macroH2A KO relieves repression at inflammatory genes, eliciting CAF-induced promotion of an immunosuppressive tumor microenvironment. The primary finding of the paper is identification of a previously undescribed role for macroH2A in suppressing inflammatory signaling in fibroblasts, as well as evidence for macroH2A suppression of melanoma cell de-differentiation. These findings were largely convincing and are interesting contributions to the literature but require some further connection to human tumors to establish relevance to human biology.

We thank the reviewer for highlighting the interesting contributions of our study and convincing nature of our data. We agree that addressing the consequences of macroH2A deregulation in human CAFs would extend the relevance of our findings. We hope the reviewer will appreciate our efforts to translate our findings to human melanoma: 1) we investigated macroH2A biology in patient-derived melanoma CAFs and 2) we analyzed a published a pan-cancer scRNA-seq dataset containing over 56,000 CAFs across 98 samples. Please see below for further details.

Major Comments:

The authors' primary finding relates to the effect of macroH2A KO in the CAFs and suggest a cell-intrinsic mechanism by in vitro studies. This provides new information about the role of macroH2A in CAFs but is limited in data to support relevance in human samples. Do human melanomas with downregulated

macroH2A exhibit increased infiltration of the myeloid compartment and decreased infiltration of lymphocytes (e.g., by histology)? In probing published human scRNA-seq datasets, does the stromal compartment exhibit a macroH2A-high and -low state? If so, do these states correlate with a distinct immune landscape?

We thank the reviewer for these suggestions. Our updated CIBERSORT analysis shows M2 (pro-tumor) macrophages are significantly more abundant in macroH2A2^{low} primary tumors and near significance in macroH2A1^{low} tumors, accompanied by significant CD8 T cell depletion in macroH2A1 and 2 low tumors (**Fig. 5e**).

We agree that further confirmatory assays such as IHC in human samples would supplement our data. As the reviewer remarks, the impact of macroH2A loss on the TME appears to stem from the CAF population, therefore, this analysis should target the CAF population specifically. We note that IHC does not have the necessary resolution to evaluate macroH2A levels in CAFs vs. rest of the tumor, and would require simultaneous detection of at least one CAF marker, macroH2A1/2, and normalization controls such as histone H3. However, we now include two new sources of data that confirm our observations in human CAFs.

We obtained melanoma CAF cultures from the NCI Patient-Derived Models Repository (note: application proposal approval was required), as well as additional cultures from Dr. Andrew Aplin who successfully used them to demonstrate the role of stromal ligands in promoting melanoma resistance to targeted therapy (<https://pubmed.ncbi.nlm.nih.gov/26269601/> , <https://pubmed.ncbi.nlm.nih.gov/30115691/>). Our studies revealed an anticorrelation between endogenous macroH2A2 levels and CCL2, CXCL1, and IL-6 cytokine production in these human CAFs (**Fig. 5f, g, S5h**). While this analysis did not reach significance, not surprising given the heterogeneous source material and possible confounding variables, it paralleled the direction of change observed in our mouse model.

Following the reviewer's suggestion, we analyzed a large scRNA-seq dataset containing over 56,000 CAFs across 98 samples comprising multiple cancer types. We found that macroH2A2 expression is below the detection threshold of 10X Visium scRNA-seq (**Fig. S5i**), therefore we could not carry out correlation analysis at the single cell level. Instead, we calculated pseudobulk values for the entire CAF population of each tumor and found a subset of tumors were populated with CAFs in a macroH2A2^{low} state. Importantly, a significant anticorrelation between macroH2A2 and IL-6 expression emerged (**Fig. S5j**).

- When comparing the GSEA data from the murine model versus the human samples, several of the pathways upregulated in the murine dKO model are downregulated in the human MACROH2A1 low samples (TNF α signaling via NFKB, IL6 Jak Stat3 signaling, IFN α /g response). What would explain this signature inversion? In looking at the myeloid-related genes (as in Fig 2B) in the human datasets, do they have a similar expression pattern? Does testing a gene signature for myeloid cell activation, cytotoxic T cell activation, etc. show a parallel between the murine and human samples?

MACROH2A2^{low} primary tumors (**Fig. R5**), as the reviewer expected. Regarding myeloid cell activation, the updated CIBERSORT analysis shows the macrophage alternative activation signature (M2 macrophages) is significantly higher in *macroH2A2*^{low} primary tumors and near significance in *macroH2A1*^{low} tumors (**Fig. 5e**), altogether showing a parallel between the murine model and human samples.

Minor Comments:

- Interesting that the H2A2-low tumors exhibit a significant decrease in the cytotoxic genes (Fig 2C) but an insignificant difference in CD8 T cell abundance (Fig 2D). What explains why H2A1 would significantly decrease CD8 T cells but not H2A2? If the samples are stratified that are low in both H2A1 and H2A2, is the difference even more pronounced?

We completely overhauled the TCGA analysis by implementing a recent analysis package, IOBR, which contains the CIBERSORT functionality. In contrast to our previous approach, this allowed us to use data normalized in the same manner via the robust TMM method, both for stratifying samples according to macroH2A expression, and for calculating immune population estimates; we no longer relied on CIBERSORT scores calculated by others; we compared samples in the 1st and 3rd macroH2A1 or macroH2A2 expression terciles, eliminating samples with intermediate expression levels which may have reduced the amplitude of differences observed.

The discrepancy highlighted by the reviewer was a shortcoming of the approach we used previously. *MACROH2A2* RNA-seq counts are low and likely affected the accuracy of stratification using the previous FPKM normalization method. Our updated CIBERSORT approach now shows significant CD8 T cell depletion in *macroH2A1* and 2 low tumors both in the primary and metastatic setting (**Fig. 5e, S5d**).

As suggested by the reviewer, we also compared samples with combined low levels of both macroH2A1 and macroH2A2. For this, we stratified each gene based on the median instead of terciles (otherwise, there would be too few samples in each of the possible 9 categories), and the results are similar (Fig. R6).

Figure R6. IOBR analysis of CIBERSORT immune cell type scores in TCGA SKCM primary tumors that have expression values of both macroH2A genes above or below the median.

- Is there an increase in myeloid cells by histology in the dKO? Where do these cells localize?

Localization to the stroma would further support a CAF-centric effect.

Regarding the increase in myeloid cells, we believe that flow cytometric analysis of $n_{WT} = 12$ and $n_{dKO} = 15$ mice in 3 independent flow cytometry experiments is robust evidence, complemented by an orthogonal technique, scRNA-seq, where now a 3rd replicate also shows the Mac Mrc1 cluster is more abundant in the dKO. Both flow cytometry and scRNA-seq approaches sample a single cell suspension of the entire tumor and is likely more accurate/representative than histology, which would assess a single section per tumor. Furthermore, distinguishing the Mac Mrc1 cells (or cells annotated as monocytes by flow cytometry) from other myeloid cells (i.e., DCs with a different functional impact on tumor immunity) by IHC would be very challenging as it relies on co-staining for multiple markers.

To address cell type co-localization as requested by the reviewer, we instead performed spatial transcriptomics on WT and dKO tumors at the same time point during melanoma development, and projected cell identities from the scRNA-seq dataset onto the spatial dataset. This showed a distinct peritumoral localization of both the significantly dKO enriched Mac Mrc1 cells and 3

out of 4 CAF clusters (**Fig. 5b**). Furthermore, correlation analysis of these projected cell type scores revealed a significant, positive association between Mac Mrc1 and the 3 CAF clusters at the spot level (**Fig. 5c**). Importantly, we also observed a significant negative association between Tc (cytotoxic T cells) and both Mac Mrc1 and the 3 CAF clusters (Meg3, Lrrc15, Fbln1), supporting our hypothesis that CAFs and the myeloid cells they attract exclude cytotoxic T cells.

Furthermore, we demonstrate experimentally that serum-stimulated dKO CAFs induce increased migration and/or proliferation of WT monocytes compared to WT CAFs via a transwell assay (**Fig. 5d, S5c**). This functional readout supports our assumption that more monocyte-derived cells would be recruited to the TME niche where dKO CAFs reside.

- Please include a description of the acronyms used in Fig 4A in the figure, legend, or text.

We apologize for the oversight and have updated the corresponding figure legend (**now Fig. 3a**) to point the reader to **Table 3**, which describes these acronyms.

- After filtering the cells, are there equivalent numbers of cells for the WT and dKO samples in Fig 4? Normalizing the values in Fig 4C to the total number of cells per tumor (% filtered cells of parent tumor) would correct for sampling bias.

Values in this panel, now located in **Fig. 3d**, are presented as normalized to the total number of cells per tumor passing quality control filters. This is now emphasized in the figure legend.

- CAF Tnc drive the dKO transcriptional profile most significantly, but CAF Meg3 shows the biggest change in abundance in dKO. Do the authors believe all 3 CAF subpopulations contribute to this dKO phenotype?

The Augur analysis, performed on a dataset now including a 3rd replicate and updated clustering of cell populations including CAFs, now shows CAF Meg3 as the top driver of the transcriptional profile, besides its change in abundance in the dKO. However, we observe significant increase in the TNF α signaling via NF- κ B pathway and immediate-early gene induction in all CAF clusters (**Fig. 4c, S4e, f**), as well as upregulation of individual inflammatory genes we associate with macroH2A-driven epigenetic alterations (**Table R1**; see above in response to reviewer 1).

- Fig S5g shows H2afy2 expression in CAFs and H2afy expression in all clusters. This is assumedly in the WT tumors but isn't specified in the figure or corresponding legend. Please show or describe the sample origin of this expression.

We apologize for the oversight and have updated the panel, now **Fig. S6h**, to state expression is shown in WT samples.

- In looking at the CAF ATAC-seq for the inflammatory genes differentially expressed in the CAF RNA-seq and exhibiting differential H3K27ac, are these genes in the newly open peaks or unchanged but open in both dKO and WT?

ATAC-seq changes are minimal and show minimal overlap with gene expression changes. To our knowledge, other groups have also found a lack of ATAC changes upon macroH2A depletion, but their data remains unpublished. Inflammatory genes, examples of which are shown in **Fig. 7e-g** and **Fig. S7g-h**, display open chromatin in both dKO and WT. Only *Cxcl1* gains accessibility in an intragenic ATAC peak.

- Line 347 misspelled word “prevoiusly” should be “previously.”

This mistake was corrected.

- Since macroH2A2 was shown to be specific to CAFs, would it be expected that CUT&RUN for H2A2 would exhibit distinct enrichment localization?

We attempted to perform macroH2A2 CUT&RUN but failed for reasons that remain unclear, but may be due to much lower abundance of this variant vs. macroH2A1 (~5-fold at RNA level). In dermal fibroblasts, which express both variants, they are distributed in a similar manner, with correlated enrichment levels (**Fig. R2, R7**). We therefore expect this similar distribution to be maintained in CAFs; however, we cannot rule out distinct functions for macroH2A2 in the CAF compartment.

Figure R7. *macroH2A1* and *macroH2A2* ChIP signal around the *Cxcl1* locus in dermal fibroblasts.

- Figure S4i isn't referenced in the text, and lines 352-353 should reference Fig 6f and 6g.

We thank the reviewer for noticing, and this omission was corrected.

Decision Letter, first revision:

Our ref: NCB-A48296A

11th May 2023

Dear Dr. Bernstein,

Thank you for submitting your revised manuscript "MacroH2A restricts inflammatory gene expression in melanoma cancer-associated fibroblasts by coordinating chromatin interactions" (NCB-A48296A). It has now been seen by the original referees and their comments are below. The reviewers find that the paper has improved in revision, and therefore we'll be happy in principle to publish it in Nature Cell Biology, pending minor revisions to satisfy the referees' final requests and to comply with our editorial and formatting guidelines.

Thank you again for your interest in Nature Cell Biology Please do not hesitate to contact me if you have any questions.

Sincerely,

Zhe Wang, PhD
Senior Editor
Nature Cell Biology

Tel: +44 (0) 207 843 4924
email: zhe.wang@nature.com

Reviewer #1 (Remarks to the Author):

Filipescu et al. present the revised version of the manuscript showing that loss of macroH2A contributes to melanoma progression by changing the tumor microenvironment. In particular, they convincingly demonstrate that cancer-associated fibroblasts are highly macroH2A-sensitive and that its loss results in the upregulation of cytokine-encoding genes and immunomodulation consisting of increased attraction of immunosuppressive monocytes and a reduction of cytotoxic T-cells.

This is a great study which advances the fields of cancer, chromatin biology and inflammation. For the histone variant community these findings are a milestone and exemplify the importance of histone variants in inflammatory gene expression and immunomodulation.

While I was supportive of the study from the very beginning, I had many comments and suggestions that the authors have addressed in an extensive revision. The revised version and in particular the new figures are easy to read and data and results are of high quality. I have left a few very minor

comments that I believe the authors will be able to address without the need of another revision from my side. I am looking forward to seeing the study published.

Minor comments:

Figure 3c and 5b: Consider showing 'zoom ins' to illustrate the points you wish to make.

Figure 4b: Add a legend showing what is blue, red, light red and light blue.

Figure 5b: Add a legend showing what bullet size encodes and add a label to the color scale.

Figure 7b: Consider adding a figure showing how super and standard domains were defined.

Figure 7d: Consider changing the top label to 'Genes gaining loops in dKO' to clarify that this analysis is on the gene level.

I conclude by congratulating the authors to their work and this excellent revision.

Marcus Buschbeck with the support of David Corujo

Reviewer #2 (Remarks to the Author):

After going through your point by point response and evaluated the revised manuscript I think you have addressed my previous concerns sufficiently.

Reviewer #3 (Remarks to the Author):

In this revised manuscript, the authors have made substantial edits and added additional datasets in response to our comments, strengthening human melanoma relevance and adding additional mechanistic insight. These revisions have sufficiently addressed our concerns. The manuscript will be of benefit to the readers of Nature Cell Biology.

Author Rebuttal, first revision:

Reviewer #1 (Remarks to the Author):

Filipescu et al. present the revised version of the manuscript showing that loss of macroH2A contributes to melanoma progression by changing the tumor microenvironment. In particular, they convincingly demonstrate that cancer-associated fibroblasts are highly macroH2A-sensitive and that its loss results in the upregulation of cytokine-encoding genes and immunomodulation consisting of increased attraction of immunosuppressive monocytes and a reduction of cytotoxic T-cells.

This is a great study which advances the fields of cancer, chromatin biology and inflammation. For the histone variant community these findings are a milestone and exemplify the importance of histone variants in inflammatory gene expression and immunomodulation.

While I was supportive of the study from the very beginning, I had many comments and

suggestions that the authors have addressed in an extensive revision. The revised version and in particular the new figures are easy to read and data and results are of high quality. I have left a few very minor comments that I believe the authors will be able to address without the need of another revision from my side. I am looking forward to seeing the study published.

We thank the Reviewer(s) for their overall enthusiasm and support of our study that demonstrates *'the importance of histone variants in inflammatory gene expression and immunomodulation'*. We have addressed these final minor comments below:

Minor comments:

Figure 3c and 5b: Consider showing 'zoom ins' to illustrate the points you wish to make.

Response: We thank the Reviewer for this very helpful suggestion and have added the requested zoomed inserts.

Figure 4b: Add a legend showing what is blue, red, light red and light blue.

Response: A legend was added to this panel.

Figure 5b: Add a legend showing what bullet size encodes and add a label to the color scale.

Response: We believe the Reviewer is referring to Figure 5c; we have added a statement to the legend explaining dot size is proportional to the r coefficient.

Figure 7b: Consider adding a figure showing how super and standard domains were defined.

Response: The approach used to define different macroH2A domain classes is described in detail in the Methods section and accompanied by plots illustrating the features of these domains in Supplemental Figure 6c-e.

Figure 7d: Consider changing the top label to 'Genes gaining loops in dKO' to clarify that this analysis is on the gene level.

Response: To clarify this point, we added the label "Classes of genes:", which applies to all categories overlapped in Figure 7d.

I conclude by congratulating the authors to their work and this excellent revision.

We thank the Reviewer(s) once again for their continued support of our manuscript, in-depth review, and valuable recommendations during the revision process.

Marcus Buschbeck with the support of David Corujo

Reviewer #2 (Remarks to the Author):

After going through your point by point response and evaluated the revised manuscript I think you have addressed my previous concerns sufficiently.

Response: We thank the Reviewer for their careful review and support of our study.

Reviewer #3 (Remarks to the Author):

In this revised manuscript, the authors have made substantial edits and added additional datasets in response to our comments, strengthening human melanoma relevance and adding additional mechanistic insight. These revisions have sufficiently addressed our concerns. The manuscript will be of benefit to the readers of Nature Cell Biology.

Response: We are pleased to have strengthened the human melanoma relevance and thank the Reviewer for their support of publication in *Nature Cell Biology*.

Final Decision Letter:

Dear Dr Bernstein,

I am pleased to inform you that your manuscript, "MacroH2A restricts inflammatory gene expression in melanoma cancer-associated fibroblasts by coordinating chromatin looping", has now been accepted for publication in Nature Cell Biology.

Thank you for sending us the final manuscript files to be processed for print and online production,

and for returning the manuscript checklists and other forms. Your manuscript will now be passed to our production team who will be in contact with you if there are any questions with the production quality of supplied figures and text.

Please note that *Nature Cell Biology* is a Transformative Journal (TJ). Authors may publish their research with us through the traditional subscription access route or make their paper immediately open access through payment of an article-processing charge (APC). Authors will not be required to make a final decision about access to their article until it has been accepted. Find out more about Transformative Journals

To assist our authors in disseminating their research to the broader community, our SharedIt initiative provides you with a unique shareable link that will allow anyone (with or without a subscription) to read the published article. Recipients of the link with a subscription will also be able to download and

print the PDF.

If you have not already done so, we strongly recommend that you upload the step-by-step protocols used in this manuscript to the Protocol Exchange (www.nature.com/protocolexchange), an open online resource established by Nature Protocols that allows researchers to share their detailed experimental know-how. All uploaded protocols are made freely available, assigned DOIs for ease of citation and are fully searchable through nature.com. Protocols and Nature Portfolio journal papers in which they are used can be linked to one another, and this link is clearly and prominently visible in the online versions of both papers. Authors who performed the specific experiments can act as primary authors for the Protocol as they will be best placed to share the methodology details, but the Corresponding Author of the present research paper should be included as one of the authors. By uploading your Protocols to Protocol Exchange, you are enabling researchers to more readily reproduce or adapt the methodology you use, as well as increasing the visibility of your protocols and papers. You can also establish a dedicated page to collect your lab Protocols. Further information can be found at www.nature.com/protocolexchange/about

With kind regards,

Zhe Wang, PhD
Senior Editor
Nature Cell Biology

Tel: +44 (0) 207 843 4924
email: zhe.wang@nature.com

** Visit the Springer Nature Editorial and Publishing website at www.springernature.com/editorial-and-publishing-jobs for more information about our career opportunities. If you have any questions please click here.**